



# A simple dynamical model linking radiative-convective instability, convective aggregation and large-scale dynamics

Matthew Davison[1,*] and Peter Haynes[1,*]

[1]Department of Applied Mathematics and Theoretical Physics, University of Cambridge
[*]These authors contributed equally to this work

**Correspondence:** Matt Davison (md752@cam.ac.uk)

**Abstract.** A simple model is used to analyse the relation between the phenomenon of convective aggregation at small scales and larger scale variability, including MJO-like behaviour, that results from coupling between dynamics and moisture in the tropical atmosphere. The model is based on the single-layer dynamical equations coupled to a moisture equation to represent the dynamical effects of latent heating and radiative heating. The moisture variable $q$ evolves through the effect of horizontal convergence, nonlinear horizontal advection and diffusion. Following previous work, the coupling between moisture and dynamics is included in such a way that a horizontally homogeneous state may be unstable to inhomogeneous disturbances and, as a result, localised regions evolve towards either dry or moist states, with respectively divergence or convergence in the horizontal flow. The behaviour of the model system is investigated using a combination of theory and numerical simulation. The spatial organisation of the moist and dry regions demonstrates a spatial coarsening that, if moist regions and dry regions are interpreted respectively as convecting and non-convecting, represents a form of convective aggregation. When the weak temperature gradient (WTG) approximation (i.e. a local balance between heating and convergence) applies and horizontal advection is neglected the system reduces to a nonlinear reaction-diffusion equation for $q$ and the coarsening is a well-know aspect of such systems. When nonlinear advection of moisture is included the large-scale flow that arises from the spatial pattern of divergence and convergence leads to a distinctly different coarsening process. When thermal and frictional damping and $f$-plane rotation are included in the dynamics, there is a dynamical length scale $L_{\mathrm{dyn}}$ that sets an upper limit for the spatial coarsening of the moist and dry regions. The $f$-plane results provide a basis for interpreting the behaviour of the system on an equatorial $\beta$-plane, where the dynamics implies a displacement in the zonal direction of the divergence relative to $q$ and hence to coherent equatorially confined zonally propagating disturbances, comprising separate moist and dry regions. In many cases the propagation speed and direction depend on the equatorial wave response to the moist heating, with the relative strength of the Rossby wave response to the Kelvin wave response determining whether the propagation is eastward or westward. The key overall properties of the propagating disturbances, the spatial scale and the phase speed, depend on nonlinearity in the coupling between moisture and dynamics and any linear theory for such disturbances therefore has limited usefulness. The model described here, in which the moisture and dynamical fields vary in two spatial dimensions and important aspects of nonlinearity are captured, provides an intermediate model between theoretical models based on linearisation and one spatial dimension and GCMs or convection-resolving models.



# 1 Introduction

Much theoretical and modelling work over the past few decades has focused on the coupling between dynamics and moisture in the tropical atmosphere, which it is clear must be taken into account at leading-order to explain many tropical phenomena. Two topics that continue to attract significant attention are the Madden-Julian Oscillation, first identified in observations as a pattern of behaviour of the real tropical atmosphere, and convective aggregation, identified as a behaviour in numerical simulations in convection-representing models. The Madden-Julian Oscillation (hereafter MJO) is the dominant mode of intraseasonal variability of the tropical atmosphere and, until recently, has not been reliably simulated in global models. Recent reviews (e.g. Jiang et al., 2020; Zhang et al., 2020) have emphasised that there still is no universally accepted theory for the oscillation, which reduces confidence in modelling capability. Convective aggregation (e.g Wing et al., 2017; Muller et al., 2022) has been identified as a pattern of behaviour of a hypothetical tropical atmosphere in which there is no imposed spatial inhomogeneity, but which exhibits spontaneous organisation of the circulation and convection into regions of two types, one type with active convection and large-scale ascent, and the other with convection suppressed by large-scale subsidence. The relevance of convective aggregation to the behaviour of the real atmosphere remains a topic of debate, but the study of aggregation in numerical models has provided a great deal of insight into the physics of the tropical atmosphere, particularly the interactions between convecting and non-convecting regions, and the way in which this physics is represented in models.

Furthermore it has also been suggested that the same physics that is responsible for convective aggregation in numerical simulations is also part of the mechanism for the MJO (Bretherton et al., 2005; Arnold and Randall, 2015). Investigating this possibility in numerical simulations has been challenging because the numerical resolution for convective-representing simulations is a few km or less and the spatial structure of the MJO has scales of several thousand km. A small number of papers (e.g Arnold and Randall, 2015; Khairoutdinov and Emanuel, 2018) have described simulations that bridge this gap and have given insight into the relation between convective aggregation and the spontaneous generation of large-scale MJO-like disturbances. Nonetheless, since such simulations are at the very edge of current computational capacity and the scope for thorough examination of parameter space is limited, it is desirable to find a simpler theoretical and modelling framework within which the link between processes such as aggregation on the mesoscale and larger scale organisation of dynamics can be investigated further. The focus of this paper is the formulation and study of a simple model for such investigation.

Several different physical processes have been proposed as important for aggregation, often supported by the results of mechanism-denial experiments in CRMs in which the effects of particular processes have been altered or omitted altogether. One suggested description of the route to aggregation is via an instability of a spatially homogeneous radiative-convective equilibrium state resulting from feedbacks between moisture and radiation (Raymond, 2000; Emanuel et al., 2014), primarily in the free troposphere, though the boundary layer (Yang, 2018) may play a crucial role in the dynamics of these feedbacks. Instability, generally described as radiative instability or radiative-convective instability, may result if these feedbacks can overcome the dynamical stability of the convecting atmosphere. The competition between these processes is typically represented



by the gross moist stability (e.g Raymond et al., 2009), though changes in convective moisture and heat transport should be
taken properly into account (Beucler et al., 2018). The growing instability is expected ultimately to saturate at finite ampli-
tude with the result that locally the system tends towards a moist or a dry state. Such behaviour has been demonstrated using
single-column radiation-convection calculations using various standard radiative and convective parametrizations and adopting
the weak temperature gradient (WTG) assumption where the environmental temperature is specified and the vertical mass flux
is allowed to be non-zero (Sobel et al., 2007; Emanuel et al., 2014). Under suitable conditions the radiative-convective (RCE)
state is unstable and the system evolves towards a moist state or a dry state, depending on the imposed initial perturbation.
Another distinct family of descriptions of the route to aggregation invokes the spatio-temporal propagation of triggering of
convection through different effects, either through the formation and propagation of cold pools (e.g. Hirt et al., 2020), or
through the propagation of gravity waves within the boundary layer (Yang, 2021). As emphasised by Muller et al. (2022),
there remains considerable uncertainty over which of these various descriptions of the aggregation process is most relevant to
convective aggregation and, within each of them, the relative importance of different physical mechanisms.

A more generic and fundamental approach to the study of convective aggregation, starting with instability as the initial
mechanism for the growth of moisture inhomogeneities on the homogeneous state, but then seeking a description of the spa-
tial evolution of the resulting moist and dry regions, has been undertaken by Craig and Mack (2013) and Windmiller and
Craig (2019) (hereafter CMWC). The model system considered in this work is an evolution equation for a time-evolving two-
dimensional moisture concentration field $q$, incorporating a source-sink term that is a nonlinear function of $q$, $G(q)$ say, with
three zeros, each corresponding to possible steady states. The form of the function $G(q)$ is such that the large-$q$ (moist) and
small-$q$ (dry) states are stable and the intermediate-$q$ state is unstable. Transport of moisture is assumed to be diffusive. This
system is equivalent to a reaction-diffusion equation with bistable reaction, sometimes known as the Allen-Cahn equation,
which has been much studied using theoretical and numerical approaches. Such systems exhibit *coarsening* where, after the
initial separation into high-$q$ and low-$q$ regions, typically on small scales, the scale of these regions increases monotonically
with time, until constrained by the large-scale geometry imposed on the system. This is presented by CMWC as a mechanism
for aggregation. The two essential ingredients required for this mechanism to operate are the $q$-dependence, and hence the
'bistable' nature, of the source-sink term, which CMWC argue results from the dependence of subsidence drying and convec-
tive moistening on free-tropospheric moisture, and the diffusive transport. Windmiller and Craig (2019) argue that diffusive
transport can be justified on the basis of a simple model in which a stochastic convective cloud moistens the environment of a
moist region and provide an estimate for the resulting diffusivity as $4 \times 10^2 \mathrm{m}^2\mathrm{s}^{-1}$. A larger value of the diffusivity, $10^5 \mathrm{m}^2\mathrm{s}^{-1}$,
is used in Craig and Mack (2013) envisaged as based on the typical horizontal velocity and length scales of convective systems.
The latter justify this as an eddy diffusivity based on the typical horizontal velocity and length scales of convective motions.
The appropriate value of the diffusivity therefore depends significantly on what the diffusivity is intended to represent. This
reactive-diffusive description of aggregation is interesting, but it has limited usefulness in investigating a link between con-
vective aggregation and the MJO because a link to the large-scale dynamics that is likely to play some role in the MJO is
missing.



Current theories for the MJO have recently been reviewed by Zhang et al. (2020) who focus on four different theories, setting out clearly the important differences between them, in the physics that they incorporate and in the behaviour that they predict. The focus in this paper will be on one of these four – the 'moisture-mode' theory – of which a recent example is Adames and Kim (2016), building on previous work by Raymond (2001); Sobel et al. (2001); Sobel and Maloney (2012, 2013). The key feature of moisture modes (Sobel et al., 2001; Fuchs and Raymond, 2005; Adames et al., 2019) is that there is two-way coupling between dynamics and moisture and furthermore that the dynamical variables are 'slaved' to the moisture variable, i.e. the moisture field is the only independently time-evolving field and the dynamical fields are determined instantaneously from the moisture field. It is this latter aspect that distinguished moisture modes from other phenomena that rely on moisture-dynamics coupling, such as moist gravity waves (e.g. Gill, 1982). In some studies moisture modes emerge from a system that supports a broader class of phenomena; in others (Sugiyama, 2009b; Adames and Kim, 2016) the dynamical equations are reduced to their steady-state form with a moisture dependent forcing and the moisture variable is the only independently time-evolving field, meaning that moisture modes are the only allowed evolving structures. Much of the previous application of moisture-mode theory to the MJO reduces the governing equations to a linear form and then solves for a dispersion relation to determine whether how the possible growth and propagation of disturbances depends on the spatial scale and on the external parameters defining the system. Where or not there is growth is typically determined by the gross moist stability, sometimes suitably generalised to include effects beyond latent heating, such as radiative effects. One exception is the work by Sugiyama (2009a, b) which identifies MJO-like behaviour that is fundamentally nonlinear, with the nonlinearity arising both from the nonlinear dependence of heating on moisture concentration and from nonlinear horizontal advection of moisture. The nonlinear dependence of heating and moistening on moisture concentration effectively combine provide a bistable moisture forcing. Some of the results and discussion we present below are in effect re-visiting Sugiyama (2009a, b) but with a stronger emphasis on the relation to convective aggregation and to the bistable nature of the system.

The structure of this paper is as follows. In Section 2 we define the mathematical model to be studied, following previous approaches in using the shallow-water equations augmented by a prognostic equation for moisture, with the moisture coupling to the shallow-water dynamics. We then set out the behaviour expected of the model on the basis of previous work together with various scaling arguments. In Section 3 we present results from numerical simulations of the system on a doubly periodic domain. In particular we verify that aggregation occurs and that the mechanism for aggregation can be dominated by horizonal diffusion of moisture or by horizontal advection of moisture, depending on the external parameters defining the system. In Section 4 we then show that rotation, thermal damping and frictional damping can each, or in combination, lead to a finite upper limit on the aggregation scale. This is both with theoretical arguments and results from numerical simulations on a doubly periodic $f$-plane (including the zero-rotation case $f = 0$). Then in Section 5 we consider the system on an equatorial $\beta$-plane and show that the process of aggregation is then confined to a low-latitude region with the result of aggregation being the formation of coherent propagating disturbances with scale and propagation speed. These depend on the external parameters, in particular, in some regimes, the relative strength of the equatorial Kelvin and Rossby wave responses to the moist heating. In Section 6 we discuss the results and present overall conclusions.



## 2 Model system to be studied

### 2.1 Model equations

The model to be studied in the remainder of the paper is based on the dynamical equations for a rotating shallow-water system,
describing the evolution of horizontal velocity $\boldsymbol{u}$ and free-surface displacement $h$, augmented by a moisture variable $q$ which
is transported by the horizontal velocity. The dynamical equations are linearised about a state of rest with fluid depth $H$ and
therefore take the following form:

$$\boldsymbol{u}_t = -f\boldsymbol{k} \times \boldsymbol{u} - g\nabla h - \alpha\boldsymbol{u} \tag{1}$$

and

$$h_t + H\nabla.\boldsymbol{u} = F_h(q) - \lambda h \tag{2}$$

where $g$ is the gravitational acceleration and $f$ is the Coriolis parameter which will either be taken to be constant $f = f_0$
corresponding the the $f$-plane, or to be linearly dependent on the $y$ coordinate, $f = \beta y$, corresponding to the equatorial $\beta$-plane.
$\boldsymbol{k}$ is the unit vector in the vertical. $\alpha$ is a linear friction coefficient and $\lambda$ a thermal damping rate. Note that the displacement $h$
is a surrogate for temperature in a three-dimensional atmosphere, with temperature increasing as $h$ decreases. The term $F_h(q)$
included on the right-hand side of (2) represents a moisture-dependent cooling term (cooling because of the relation between
$h$ and temperature), potentially including both latent heating and radiative heating. If cooling decreases with moisture, as is
physically plausible, then $F_h(q)$ will be a decreasing function of $q$.

The equation for the moisture variable $q$ is assumed to take the form

$$q_t + Q\nabla \cdot \boldsymbol{u} + \nabla \cdot (q\boldsymbol{u}) - \kappa\nabla^2 q = F_q(q). \tag{3}$$

including both horizontal advective transport and horizonal diffusive transport, the latter with diffusivity $\kappa$, assumed constant. The term $F_q(q)$ on the right-hand side represents the combined effects of evaporation and precipitation. $Q$ is a suitably
chosen constant, so that $q$ is the perturbation away from a background state where the 'total' moisture variable is $Q$, and it is
convenient to choose $Q$ such that $F_q(0) = 0$, i.e. such that the in the background state there is a balance between evaporation
and precipitation. Equations of the above form have been derived and studied in many previous papers on tropical dynamics,
(e.g Sugiyama, 2009b; Adames and Kim, 2016), and references therein, with $q$ regarded as a measure of moisture in the lower
troposphere. The dependence on $q$ of the part of $F_q(q)$ representing precipitation is justified in these papers on the basis of the
observed correlation between precipitation and moisture in the free troposphere (e.g. Holloway and Neelin, 2009). It is well
known that the above single-layer equations can be interpreted as an approximate representation of a first baroclinic mode in a
stratified 3-D atmosphere and there has been important previous work on systematic derivation of the appropriate form of such
equations from the 3-D equations, such as Neelin and Zeng (2000). Papers by Sugiyama (2009a, b); Adames and Kim (2016)
and others exploit this framework to give detailed justification of the appropriate choice of parameters for the single-layer
model, including possible choices for the functions $F_h(q)$ and $F_q(q)$.





In our formulation of the precipitation, incorporated into the moisture coupling terms, $F_h$ and $F_q$, we do not include any $h$-dependence (i.e. temperature dependence). The focus of this study is the slow behaviour of the moisture variable, analogous to

the moisture mode in a linear setting. In this context, the temperature dependence of precipitation is neglected in many previous papers (Adames and Kim, 2016; Sobel and Maloney, 2012, 2013). Note that Sugiyama (2009a, b) does include $h$-dependence of the moisture coupling terms and we will comment further on the effect of such dependence in Sect. 6. (See also Appendix E.)

Within the constraints of the very simple model specified above, we may identify the possibility of choosing $H$ and $Q$

such that $h = q = 0$ corresponds to a spatially homogeneous radiative-convective equilibrium (RCE) state with $\boldsymbol{u} = \boldsymbol{0}$, and $F_h$ and $F_q$ satisfying the conditions $F_h(0) = F_q(0) = 0$. We may restrict the forms of $F_h(q)$ and $F_q(q)$ to those for which the spatially homogeneous system is stable, which holds if $\partial F_q / \partial q < 0$, with the partial derivative being evaluated at $q = 0$. Then a key question is whether, within this restriction, the radiative-convective equilibrium (RCE) state is unstable to spatially inhomogenous disturbances. It is straightforward to solve the linear stability problem and to demonstrate that such instability is

possible. Without presenting further details at this stage, we simply note that we will interpret this instability as the analogue, within this simple model, of 'radiative-convective instability' that has previously been identified and described in several papers, as discussed above.

It will be helpful later to distinguish between linear and nonlinear terms on the left-hand side of (3). We therefore introduce the parameter $\epsilon$ into (3) to give

$$q_t + Q\nabla \cdot \boldsymbol{u} + \epsilon \nabla \cdot (\boldsymbol{u}q) - \kappa \nabla^2 q = F_q(q). \tag{4}$$

This is a convenient way to allow advective nonlinearity to be varied independently of the nonlinearity also present in the forms of the functions $F_h$ and $F_q$. The linear instability analysis of the RCE state, for example, takes $\epsilon = 0$ in the above equation. Note that in the form of the equations considered by Sugiyama (2009b), derived from the Neelin and Zeng (2000) QTCM equations, there is a distinct constant multiplying the nonlinear advective term. This constant is determined in principle in the

derivation of the single-layer equations by the projection of a horizontal moisture advection term that is varying in height on to the single basis function used to represent the moisture field, but can also be conveniently be varied as a independent parameter, and the $\epsilon$ being introduced here plays the same role as that parameter. We later illustrate the role of advective nonlinearity by comparing $\epsilon = 1$ behaviour with $\epsilon = 0$ behaviour, but note that, for the reasons just given, $\epsilon = 1$ may not be the 'correct' choice for including advective nonlinearity.

The key dimensional quantities that define the above system include $g$ and $H$, which determine the dry gravity-wave speed $c = \sqrt{gH}$, the horizonal diffusivity $\kappa$, the Coriolis parameter $f$, the thermal and frictional damping rates, respectively $\lambda$ and $\alpha$, a typical background value of moisture, $Q$, say, and $\mu$, an inverse timescale for the moist processes represented by $F_h$ and $F_q$. It is convenient to take the dimensions of the moisture $Q$ to be the same as those of the thickness $H$, and indeed this corresponds to a simple re-scaling of the parameters in $F_h$. To assess the importance of the advective term in (3) an additional dimensional

quantity is needed which sets the magnitude of the spatial inhomogeneous part of the moisture field and the magnitude of the corresponding horizontal flow. These magnitudes are set by the nonlinear dependence of $F_q$ and $F_h$ on $q$, but it is convenient




to choose the magnitude $D$, say, of the divergence $\nabla.\boldsymbol{u}$, as the relevant dimensional quantity. The reason for this choice will become clear from the discussion below. The relation between $D$ and spatial variations in $q$ is determined by the leading-order balances operating in (2) and (3).

## 2.2 WTG and the relation to the CMWC reaction-diffusion system

A standard approach, particularly at low latitudes where $f$ is small, to analysing the system defined by (1)-(3) is to make the *weak temperature gradient approximation* (WTG) (e.g. Sobel et al., 2001). This neglects horizontal variation of $h$ and can be justified provided that the horizontal length scale $L$ satisfies $L/c \ll T_q$ i.e. that the time scale for gravity-wave propagation through $L$ is much less than the time scale $T_q$ for moist processes. $T_q$ could either be a timescale $\mu^{-1}$ set by an appropriate combination of $F_q$ and $F_h$ (see below) or an emergent property of the system. Additionally, when damping and rotation are included, it must be the case that $L \ll L_{\text{dyn}}$, where $L_{\text{dyn}}$ is a dynamical length scale that is typically determined by $c$ together with some combination of $f$, $\alpha$ and $\lambda$. We will focus on the zero damping case in this section and return to the dynamical effects of damping and rotation in §4.

Whilst under WTG $h$ is constant in space, it may not be constant in time. Taking the spatial average of (2), using the notation $\overline{\phantom{x}}$ to denote the spatial average. It follows that

$$\frac{d\overline{h}}{dt} = \overline{F_h(q)} - \lambda\overline{h}. \tag{5}$$

The spatially varying part of (2) then has the form

$$H\nabla.\boldsymbol{u} = F_h(q) - \overline{F_h(q)}. \tag{6}$$

implying that $\nabla \cdot \boldsymbol{u}$ and hence the irrotational part of the velocity field, is determined instantaneously by the moisture field $q$. Under the assumption $f = \alpha = 0$ in this section, the rotational part of $\boldsymbol{u}$ is constant in time. When provided with this initial rotational part of the flow, assumed to be zero for the purposes of this section, (4) becomes a self-contained equation for the evolution of the $q$ field with the form:

$$q_t + \epsilon\nabla \cdot (\boldsymbol{u}[q]q) - \kappa\nabla^2 q = F_q(q) - \frac{Q}{H}(F_h(q) - \overline{F_h(q)}) = G_{hq}(q; \overline{F_h(q)}). \tag{7}$$

where the second equality defines the function $G_{hq}$. Note that whilst evaluation of $\overline{F_h(q)}$ requires knowledge of the $q$ field, for the purposes of expressing the right-hand side of the equation as a function of $q$, $\overline{F_h(q)}$ is simply a parameter that appears in the definition of that function. The notation $\boldsymbol{u}[q]$ simply expresses the fact that at each instant $\boldsymbol{u}$ is determined completely, but non-locally, by the $q$ field, through (6).

Neglecting for the moment the advection term $\epsilon\boldsymbol{u}[q].\nabla q$, this may be recognised as a reaction-diffusion equation of the type studied by CMWC. The difference is that, whereas the nonlinear 'reaction' term on the right-hand side of (7) was in CMWC's case entirely motivated by the $q$-dependence of precipitation and evaporation, in this case the reaction term is a combination of the moisture driven heating/cooling $F_h(q)$ and the moisture source/sink $F_q(q)$. A further structural difference from the system considered by CMWC is the evolving quantity $\overline{h}(t)$. The effect of this is felt by the system through the corresponding $\overline{F_h(q)}$



appearing in the definition of the reaction term. The reaction term is therefore not completely specified in advance as a function of $q$ but contains the spatially constant term $\overline{F_h(q)}$ which also drives changes in $\overline{h}$, as specified by (5). The CMWC model, on the other hand, defines the reaction term as $G(q) - \overline{G(q)}$, where $G(q)$, as $F_h(q)$ and $F_q(q)$ in the model being presented in this study, is a function that is specified in advance. Again this means that the complete reaction term requires knowledge of the spatial distribution of $q$.

Simple theory of the reaction-diffusion system with specified reaction term $G(q)$ is that (i) homogenous steady states are possible with $q$ equal to the constant value $q_s$, if $G(q_s) = 0$ and (ii) those homogeneous states are stable if $G'(q_s) < 0$ and unstable if $G'(q_s) > 0$. CMWC consider a 'bistable' system with three possible values for $q_s$, $q_- < q_0 < q_+$, such that $G(q_-) = G(q_0) = G(q_+) = 0$, and $G(q_-) < 0$, $G'(q_0) > 0$, $G'(q_+) < 0$. $G'(q_0)$ provides a useful definition of a reaction inverse timescale $\mu$. The generic behaviour for a non-linear reaction diffusion equation of this type is that locally $q$ tends to one of the stable values, partitioning the domain into two regions one with $q = q_+$ and the other with $q = q_-$, separated by interfaces of thickness $(\kappa/\mu)^{1/2}$. In the absence of rotation and damping WTG will break down on length scales of order $cT_q$, so we require $cT_q \gg (\kappa/\mu)^{1/2}$, i.e., if the reaction timescale $\mu^{-1}$ is such that $\kappa\mu \ll c^2$. The initial geometry of these two regions is set by the initial conditions. A useful simple solution is a 1-dimensional propagating reactive-diffusive wave solution with $q = q_+$ on one side of the wave and $q = q_-$ on the other. The speed of propagation of the wave, $c_{RD} \sim (\kappa\mu)^{1/2}$ is determined by the form of the reaction function $G(q)$. Defining $V(q)$ by $dV/dq = G$, so that $V(q)$ has turning points where $G(q)$ has zeros, then if $V(q_+) > V(q_-)$ the region with $q = q_+$ propagates into the region with $q = q_-$. The corresponding result for the initial value problem, in one or more space dimensions, is that $q$ tends everywhere to $q_+$. Similarly if $V(q_+) < V(q_-)$ then $q$ eventually tends everywhere to $q_-$. Only in the case $V(q_+) = V(q_-)$, which applies in particular to the Allen-Cahn equation, do both regions $q = q_+$ and $q = q_-$ persist. Note that $V(q)$ represents the area under the graph of $G(q)$. To be precise, if the choice $V(q_0) = 0$ is made, then $V(q_+)$ is the area under the graph of $V(q)$ in the interval $[q_0, q_+]$ and $V(q_-)$ is the corresponding (positive) area in the interval $[q_-, q_0]$.

An important effect in two dimensions is that the reaction-diffusion velocity $c_{RD}$ becomes a local property of each point on each interface, depending not only on the form of $G(q)$ but also on curvature of the interface. The reaction velocity $c_{RD}$ should be replaced by $c_{RD} + \kappa/R$, where $R$ is the (signed) radius of curvature of the interface (such that that the propagation is towards the interior of the curve). If $|c_{RD}| < \kappa/|R|$ the velocity speed of the boundary may even change sign. Therefore, $c_{RD}$ decreases as the curvature of the interface increases ($R$ decreases) (Rubinstein et al., 1989). This tends to smooth out the boundary between moist and dry regions, as small -scale irregularities or indeed small-scale regions will tend to disappear. Larger moist regions can therefore expand while smaller moist regions shrink. This is the standard coarsening behaviour, i.e. the geometric simplification of the geometry between the two regions through an increase in spatial scales, observed in reaction-diffusion systems (Bray et al., 2003).

CMWC's inclusion of the $\overline{G(q)}$ term, so that the total reaction term becomes $G_{CMWC}(q,t) = G(q) - \overline{G(q)}$, is important because this ensures that even if $V(q_+) \neq V(q_-)$ (with $V(q)$ defined as above), the system does not simply evolve to $q = q_-$ or $q = q_+$ everywhere. Both values of $q$ persist as coarsening proceeds. Indeed if this sort of constraint is not applied then in most cases the reaction-diffusion system with a bistable reaction evolves everywhere towards one of the stable states. It will





be demonstrated below that the same property holds for the model system being considered in this study, i.e. for a reaction-diffusion system with the reaction term as specified by the right-hand side of (7).

Following the arguments presented above, the stability of the spatially homogeneous RCE state will be determined by the derivative with respect to $q$ of $G_{hq}(q,0) = F_q(q) - (Q/H)F_h(q)$ at $q = 0$, with instability if the derivative is positive, provided that the domain size is large enough that diffusion does not stabilise the system through the action of the $\kappa\nabla^2 q$ term. (This derivative will later be identified as proportional to the negative of the gross moist stability, i.e. the RCE state will be unstable if the gross moist stability is negative.) It will be assumed that the derivative is indeed positive and furthermore that $G_{hq}(q;0)$ is

bistable in the sense that there are $q_+(0)$ and $q_-(0)$ such that $q_-(0) < 0 < q_+(0)$, with $G_{hq}(q_-(0);0) = G_{hq}(q_+(0);0) = 0$ and $G'_{hq}(q_-(0);0) < -0$, $G'_{hq}(q_+(0);0) < 0$. Note that this property of $G_{hq}(q;0)$ implies a similar property, with corresponding $q_-(\overline{F_h(q)})$, $q_0(\overline{F_h(q)})$ and $q_+(\overline{F_h(q)})$, for the more general right-hand side of (7) $G_{hq}(q;\overline{F_h(q)})$ provided that $|\overline{F_h(q)}|$ is not too large, For notational convenience the explicit dependence of e.g. $q_-(\overline{F_h(q)})$ on $\overline{F_h(q)}$ will not be displayed unless essential.

Numerical solutions below will show that if $G_{hq}$ is bistable in the sense defined then the system indeed evolves towards

two values of $q$ and that coarsening occurs. However some further insight can be obtained by assuming that after the initial adjustment the region $q = q_+$ has fills an area fraction $A_+$ and the region $q = q_-$ fills an area fraction $A_-$, with $A_+ + A_- = 1$. The area of the interfaces between the regions is assumed negligible. The configuration is therefore determined by the three unknowns $q_-$, $q_+$ and $A_+$ (or $A_-$).

Then the above equations imply

$$G_{hq}(q_+, \overline{F_h(q)}) = F_q(q_+) - (Q/H)(F_h(q_+) - \overline{F_h(q)}) = 0 \tag{8}$$

$$G_{hq}(q_-, \overline{F_h(q)}) = F_q(q_-) - (Q/H)(F_h(q_-) - \overline{F_h(q)}) = 0 \tag{9}$$

$$\overline{F_h(q)} = A_+ F_h(q_+) + A_- F_h(q_-). \tag{10}$$

These determine any two of the three variables $q_-$, $q_+$ and $A_+$ in terms of the third. For the system to be at a steady state an additional constraint, obtained by integrating (3) over the domain, might seem to be $A_+ F_q(q_+) + A_- F_q(q_-) = 0$, however

this can be deduced from (8) and (9) above and provides no extra information. Therefore it has to be accepted that one piece of further information in addition to the above is required for a unique solution. In general these equations determine a state that is quasi-steady rather than exactly steady. After the initial adjustment, when the required extra information will be determined by the initial conditions (which might, for example, set $A_+$), we expect a further slow time evolution of the three variables and correspondingly of the geometry of the dry and moist regions.

Assume that the overall effect of this slow time evolution can be captured by the classical theory for 1-D reactive diffusive waves, describing the propagation of the thin interfaces between regions of piecewise constant $q$. This suggests the (slow) time evolution equation

$$\frac{dA_+}{dt} = L_{\text{interface}} c_{\text{RD}}(q_+, q_-, \overline{F_h(q)}) \tag{11}$$





where $L_\text{interface}$ is the length of the interface between the regions and $c_\text{RD}$ is the reaction diffusion velocity, with the convention

that this is positive if the region with $q = q_+$ propagates into the region with $q = q_-$. $L_\text{interface}$ will vary in time but is certainly positive. This equation allows a steady state when $c_\text{RD}(q_+, q_-, \overline{F_h(q)}) = 0$.

That such a steady state exists can be deduced by considering the graphs of relevant functions of $q$. As noted previously, for given $\overline{F_h(q)}$ the reaction function is $G_{hq}(q, \overline{F_h(q)}) = F_q(q) - (Q/H)F_h(q) + (Q/H)\overline{F_h(q)}$. Consider first the graph of $G_{hq}(q, 0) = F_q(q) - (Q/H)F_h(q)$ as shown by the curve in Figure 1, which intersects the $q$-axis at $q_-(0)$, $q_0(0)$ and $q_+(0)$.

The value of $G_{hq}(q, \overline{F_h(q)})$ is represented by the vertical distance between this curve and the horizontal line $-(Q/H)\overline{F_h(q)}$, shown on the Figure for various values of $\overline{F_h(q)}$, varying between $C_\text{min} < 0$ and $C_\text{max} > 0$. For $C$ outside this range then $G_{hq}(q; C)$ no longer has three roots. Areas $V_+$ and $V_-$ are marked on the Figure for a particular value of $\overline{F_h(q)}$. The condition $c_\text{RD}(q_+, q_-, \overline{F_h(q)}) = 0$ is satisfied if and only if $V_+ = V_-$. It is clear from the Figure that there is one value of $\overline{F_h(q)}$, $C_s$ say, for which this holds, lying in the range $(C_\text{min}, C_\text{max})$. Substituting this value into (8)-(10) gives the corresponding values of $q_-$

and $q_+$, the dry and moist values of $q$, and $A_+$ the fractional area occupied by the moist region.

A further question concerns the stability of this steady state. It is clear from Figure 1 that $c_\text{RD}$ is an increasing function of $C$ (the area $V_+$ increases and the area $V_-$ decreases as $C$ increases. Suppose that $C > C_s$, so that $V_+(C) > V_-(C)$ and $c_\text{RD}$ is positive, i.e. regions of $q_+$ will propagate into regions of $q_-$. The consequence will be that the relative area occupied by $q_+$ will increase resulting in a decrease in $C = \overline{F_h(q)}$, if $F_h(q)$ is a decreasing function of $q$. Similarly if $C < C_s$ then $C$ will increase,

indicating that the steady state $C = C_s$ is stable.

Note that the above arguments do not describe the process of coarsening, but indicate that the two values of $q$ persist, just as they do for the special case of the Allen-Cahn equation and for the system considered by CMWC, suggesting that coarsening is relevant.

The above reasoning also suggests some of the effects of advective nonlinearity, i.e. the term $\epsilon \boldsymbol{u}[q].\nabla q$. Since moist regions

are associated with convergence and dry regions with divergence, the effect of advection will be to reduce the area of moist regions relative to those of dry regions. This suggests that in a steady state the reaction-diffusion velocity $c_\text{RD}$ has to be positive rather than zero, i.e. that $V_+ > V_-$ rather than $V_+ = V_-$. This also implies that both $q_-$ and $q_+$ are increased relative to their values without the advective nonlinearity.

We will show examples in §3 to demonstrate that the our model system, in regimes where the approximations leading to

(5) and (7) can be justified, naturally evolves to a piecewise constant configuration with both of the values of $q$, $q_-$ and $q_+$, consistent with the relevant form of $G_{hq}(q, \overline{F_h(q)})$.

The strict WTG form of the evolution equation (7) suggests that for the system (1), (2) and (4) locally $q$ will tend to one of two values $q_-$ or $q_+$ as was the case for the CMWC pure reaction-diffusion system. The distinct additional feature of the system being considered here is that there is an associated pattern of convergence and divergence. The WTG balance in (2)

implies that the divergence $\nabla.\boldsymbol{u}$ will also tend to one of two values respectively $D_- = -H^{-1}F_h(q_-, \overline{h}) = Q^{-1}F_q(q_-, \overline{h}) > 0$ or $D_+ = -H^{-1}F_h(q_+, \overline{h}) = Q^{-1}F_q(q_+, \overline{h}) < 0$. Assume that the corresponding values of $\nabla \cdot \boldsymbol{u}$ are $-D_- < 0 < D_+$. Since area integrated $\nabla \cdot \boldsymbol{u}$ is zero we expect that the areas $A_-$ and $A_+$ filled respectively by dry regions and moist regions satisfy $A_- D_- \sim A_+ D_+$.



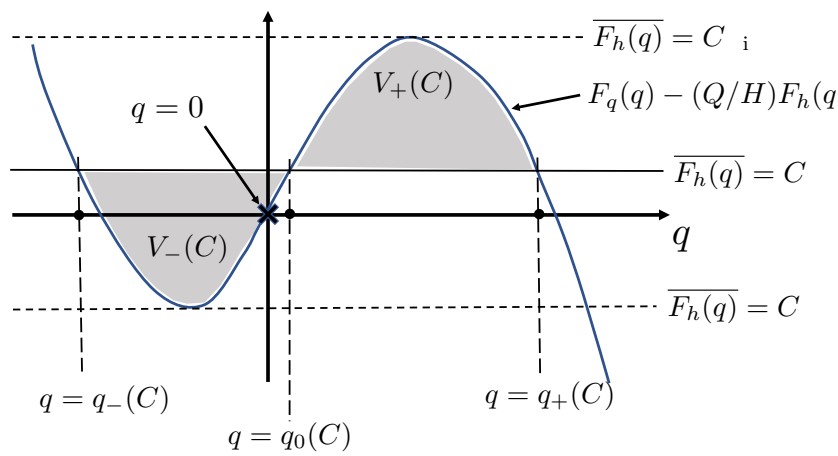

**Figure 1.** Functions of $q$ controlling the behaviour. The curve shows the function $G_{hq}(q, \overline{F_h(q)}) = F_q(q) - (Q/H)F_h(q) + (Q/H)\overline{F_h(q)}$ for the value $\overline{F_h(q)} = 0$. $q = 0$ marked by the cross, corresponds to the homogeneous RCE state. Various horizontal straight lines are shown corresponding to the values $-(Q/H)\overline{F_h(q)}$ for different values of $\overline{F_h(q)}$. The value of the function $G_{hq}(q, \overline{F_h(q)}) = F_q(q) - (Q/H)F_h(q) + (Q/H)\overline{F_h(q)}$ corresponds to the vertical difference between the curve and the relevant line. For $C_{\min} < \overline{F_h(q)}) < C_{\max}$ the curve and the straight line intersect at three values of $q$, denoted by $q_-(\overline{F_h(q)})$, $q_0(\overline{F_h(q)})$ and $q_+(\overline{F_h(q)})$. These are indicated in the diagram for $q_-(\overline{F_h(q)}) = C$. The areas between the curve and the straight line in the intervals $[q_-(C), q_0(C)]$ and $[q_0(C), q_+(C)]$ are denoted by $V_-(C)$ and $V_+(C)$ respectively. Note that it is clear that there is a choice of $C$, with $C_{\min} < C < C_{\max}$ such that $V_-(C) = V_+(C)$.

This non-zero divergence has no consequence for the evolution of the system under the strict WTG approximation, but

if this approximation is relaxed, as is likely to be required at large horizontal scales, then the coupling between moisture and divergence is likely to lead to distinctly different behaviour from that predicted by reaction-diffusion alone. This will be expected if the nonlinear advection term is included, even if WTG can be justified in (2), since this equation will determine the velocity, given the distribution of $q$. The presence of the advection term will imply that an approximate solution of piecewise constant $q$ will no longer be self-consistent at large scales. A typical velocity at length scale $L$ will be $U \sim DL$. This becomes

comparable with the reaction-diffusion velocity when $DL \sim (\kappa\mu)^{1/2}$, i.e. $L = L_{adv} \sim (\kappa\mu/D^2)^{1/2}$. On scales larger than those





of a single aggregated region, the combination of regions of divergence of opposite signs will generate a large scale velocity field. Nonlinear advection will therefore cause aggregation when there is a high density of distinct convergent regions, on a timescale of $\epsilon/D$. The advective regime of aggregation becomes dominant for length scales $L > L_{adv}$. This process will be discussed in more detail in section §3.2.

We can also now give a better estimate for $T_q$, and hence the scale at which WTG will arrest. In the linear instability phase a possible estimate is $T_q \sim \mu^{-1}$. However in the nonlinear aggregation phase a potentially more relevant estimate is $T_q \sim L/c_{\mathrm{RD}}$, i.e. the timescale increases as the length scale increases. In this case where $\alpha = \lambda = f = 0$, then for WTG the first estimate would require $L \ll c\mu^{-1}$. However the second estimate would require $L \ll cL/c_{\mathrm{RD}}$, suggesting that if $c_{\mathrm{RD}} \ll c$ then in the nonlinear aggregation phase the WTG description remains valid at all scales, i.e. aggregation simply proceeds until the length 340    scale is the largest allowed by the geometry.

## 3    Numerical simulations in the WTG regime

The model equations defined above in equations (1)–(3) can be integrated numerically, and in this section we will use this to confirm the previous results and further investigate model behaviour. Numerical details are given in Appendix A. Recall that the thickness variable $h$ and the moisture variable $q$ represent the departure from the spatially homogeneous RCE state. In all 345    simulations reported below the initial condition is taken to be $\boldsymbol{u} = \boldsymbol{0}$, $h = 0$ and $q$ small with $|q| \ll |q_\pm|$

The precise form of the functions $F_h$ and $F_q$ appearing respectively in (2) and (3) is not important to the qualitative behaviour of the system, provided that they together lead to a bistable moisture equation (7). However, in order to investigate the behaviour numerically we will need to define these forms explicitly. For illustrative processes we will use a piecewise linear construction, with

$$F_q(q) = -\mu_1 q \tag{12}$$

$$F_h(q) = \begin{cases} -\mu_2 q_p - \mu_1(q - q_p), & q > q_p \\ -\mu_2 q, & q_m < q < q_p \\ -\mu_2 q_m - \mu_1(q - q_m), & q < q_m. \end{cases} \tag{13}$$

With $\mu_2 > \mu_1 > 0$ this represents larger effective latent heat release of precipitation near to the RCE state. Note that the key quantity $\mu = (d/dq)G_{hq}(q;0)|_{q=0}$ is equal to $-\mu_1 + \mu_2 Q/H$ which we write as $-\mu_1(1 - \mu_2 Q/H\mu_1) = -\mu_1 M$ where $M$ is a normalised gross moist stability of the RCE state. Throughout the paper we choose $\mu_1$ and $\mu_2$ such that $M < 0$, implying 355    that the RCE state is unstable. This simple formulation of $F_h$ and $F_q$ has the advantage that we can easily tune the locations of the fixed points $q_\pm$ using the parameters $q_p$ and $q_m$. Note that the $q_+ > q_p$ and $q_- < q_m$. $Q$ and $H$ are chosen such that $1 - Q/H > 0$, implying that the fixed points $q_p$ and $q_m$ are stable according to the analysis in §2.2. We will tend to choose $|q_p| > |q_m|$, corresponding to a more extreme values of moisture in moist regions than in dry regions. Together with the constraint of zero net heating in steady state this implies small moist regions with strong upwelling and large dry regions weak 360    downwelling, as typically observed in convective aggregation (e.g Muller and Bony, 2015).





| | Figures | $f$-plane regime(s) | $\beta$-plane regime | steady state | $f$ [s$^{-1}$] | $\alpha$ [s$^{-1}$] | $\lambda$ [s$^{-1}$] | $\kappa$ [m$^2$s$^{-1}$] | $\epsilon$ | $Q$ [m] |
|---|---|---|---|---|---|---|---|---|---|---|
| 1 | 2,3,5a | I | - | Y | 0 | 0 | 0 | $10^5$ | 0 | 150 |
| 2 | 5b | I | - | Y | 0 | 0 | 0 | $10^5$ | 0.5 | 150 |
| 3 | 5c | I | - | Y | 0 | 0 | 0 | $10^5$ | 1 | 150 |
| 4 | 7A, 10 | I | - | Y | $10^{-5}$ | $4 \times 10^{-6}$ | $4 \times 10^{-6}$ | $10^5$ | 0 | 150 |
| 5 | 7B | IIa | - | N | $10^{-5}$ | $10^{-7}$ | $10^{-6}$ | $10^5$ | 0 | 150 |
| 6 | 7C | IIb | - | N | $10^{-5}$ | $10^{-6}$ | $10^{-4}$ | $10^5$ | 0 | 150 |
| 7 | 12a | I | - | Y | $10^{-6}$ | $10^{-6}$ | $10^{-6}$ | $10^5$ | 1 | 150 |
| 8 | 12b | I | - | Y | $10^{-5}$ | $10^{-5}$ | $10^{-5}$ | $10^5$ | 0 | 150 |
| 9 | 12c | I | - | Y | 0 | $10^{-5}$ | $10^{-5}$ | $10^5$ | 0 | 150 |
| 10 | 14a | I,III | B | N | $2 \times 10^{-11}y$ | $10^{-5}$ | $10^{-5}$ | $10^5$ | 0 | 150 |
| 11 | 14b | I,III | B | N | $2 \times 10^{-11}y$ | $10^{-5}$ | $3 \times 10^{-5}$ | $10^5$ | 0 | 150 |
| 12 | 14c | I,III | A | N | $2 \times 10^{-11}y$ | $3 \times 10^{-5}$ | $3 \times 10^{-5}$ | $10^5$ | 0 | 150 |
| 13 | 14d | I,II,III | C | N | $2 \times 10^{-11}y$ | $10^{-6}$ | $10^{-5}$ | $10^5$ | 0 | 150 |
| 14 | 14e | I,II,III | C | N | $2 \times 10^{-11}y$ | $10^{-6}$ | $3 \times 10^{-6}$ | $10^5$ | 0 | 150 |
| 15 | 14f, 16a,b | I,III | C | Y | $2 \times 10^{-11}y$ | $10^{-5}$ | $10^{-5}$ | $4 \times 10^5$ | 0 | 150 |
| 16 | 15, 16c,d | I,III | D | Y | $2 \times 10^{-11}y$ | $10^{-6}$ | $10^{-6}$ | $1.5 \times 10^5$ | 0 | 105 |
| 17 | 18a, 19, 20 | I,III | C | N | $2 \times 10^{-11}y$ | $10^{-5}$ | $10^{-5}$ | $4 \times 10^5$ | 1 | 150 |
| 18 | 18b | I,III | B | N | $2 \times 10^{-11}y$ | $10^{-5}$ | $10^{-5}$ | $10^5$ | 1 | 150 |
| 19 | 18c | I,II,III | B | N | $2 \times 10^{-11}y$ | $3 \times 10^{-6}$ | $3 \times 10^{-5}$ | $10^5$ | 1 | 150 |
| 20 | 18d | I,III | D | N | $2 \times 10^{-11}y$ | $10^{-6}$ | $10^{-6}$ | $1.5 \times 10^5$ | 1 | 105 |

**Table 1.** Parameters for all examples of two-dimensional simulations shown in figures. Parameters which remain constant are $\mu_1 = 1/36000$ s$^{-1}$, $\mu_2 = 1/12000$ s$^{-1}$, $c^2 = 300$ m$^2$s$^{-2}$, $q_+ = 0.1Q$ and $q_- = -0.025Q$. For 1–9 the $f$-plane regime is specified on the basis of figure 6. For 10–20 (equatorial $\beta$-plane simulations) the set of $f$-plane regimes encountered as $f$ increases from zero is given. For 10–20 the $\beta$-plane regime is specified on the basis of figure 13. The 'steady state' column denotes whether or not the parameter values allow convergence to a steady state or, for the $\beta$-plane, steadily propagating state, at large times.

Now that the equations are fully defined, the most basic starting point is to consider the system with no rotation or damping, $f = \lambda = \alpha = 0$. The computational domain is taken to be square with sides of length $10^7$m. We will initially take the other parameters in the system as $g = 10$ms$^{-2}$, $H = 30$m, $\mu_1^{-1} = 36000$s, $\mu_2 = 3\mu_1$, $\kappa = 10^5$m$^2$s$^{-1}$, $Q = 15$m and $q_p/Q = 0.1$, $q_m/Q = -0.025$. The precise values of these parameters are unimportant as we are aiming here to understand the general behaviour of the system, and indeed the parameters will be varied throughout the paper, however the values are chosen to give similar moisture time scales to those deduced from the system studied by Sugiyama (2009b). The parameter values used in all of the two-dimensional simulations discussed in the paper are given in Table 1.



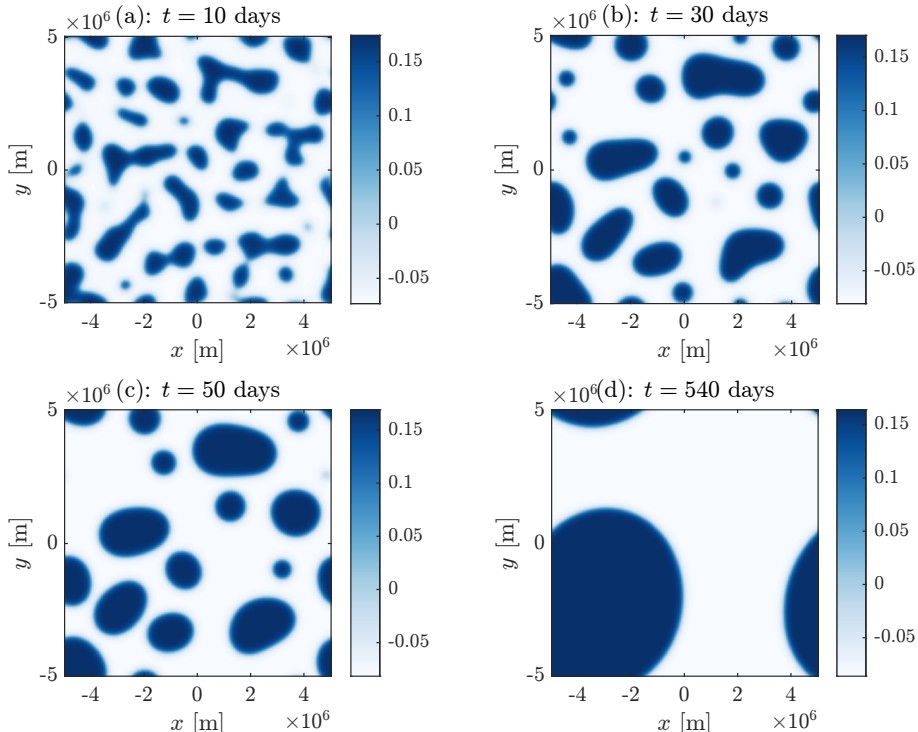

**Figure 2.** A series of snapshots of the perturbation moisture distribution $q/Q$ from a numerical integration with no rotation or damping. Note that the final panel in this series is very close to, but has not reached, steady state, which would be a perfectly round moist region.

## 3.1 $\epsilon = 0$ (Advective nonlinearity excluded)

We first consider the case $\epsilon = 0$, without nonlinear advection of moisture. The evolution of the moisture distribution with time

is shown in figure 2. The qualitative behaviour is similar to CMWC. There is an initial adjustment phase on the time scale $\mu^{-1}$ as the small-scale noise grows. During this initial phase WTG applies only up to $L \sim \mu^{-1}c$, about $10^6$m for the parameters chosen. Hence distinct regions of enhanced and suppressed moisture on this scale, or less, form, with values corresponding to the effective stable fixed points $G_{hq} = 0$, $\partial G_{hq}/\partial q < 0$.

Once this has occurred, the coarsening process proceeds, with the scale of moist and dry regions slowly evolving, consistent

with understanding of the reaction-diffusion system as discussed in the previous section. In particular the evolution of the boundaries occurs on a slow time scale determined by the reaction-diffusion velocity $c_{\mathrm{RD}}$ and the smaller curvature-associated velocity $\kappa/R$, both of which are smaller than the gravity wave speed $c$. Hence the WTG approximation continues to hold as the scale of moist and dry regions increases. In this regime the proportion of the domain filled by each of the moist and dry state is changing, so there is a slow evolution of the mean heating $\overline{F_h(q)}$. This causes a slow change in the locations of the stable

moisture fixed points $q_\pm$, however this is on a longer timescale than those determined by $G_{hq}$ so the $q$ distribution quickly




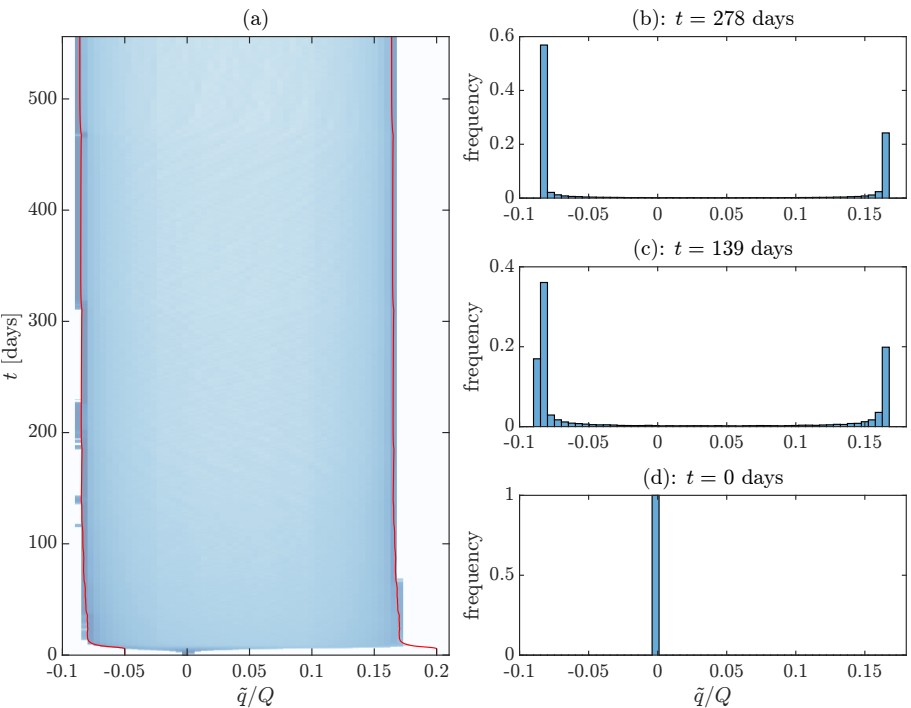

**Figure 3.** Panel (a): A histogram of the perturbation moisture distribution $q/Q$, plotted against time. The shading corresponds to the frequency distribution of the moisture. In panels (b)–(d) the histogram bars have been shown at selected times.

adjusts to the new stable values. These features of the long term evolution of the moisture distribution, the rapid adjustment to values of $q$ close to $q_\pm$ and the subsequent slow evolution of the values of $q_\pm$, is shown in figure 3.

This diffusive growth proceeds to the domain scale, when the areas of the regions are such that there is net zero heating and precipitation, and, consistent with theory (Rubinstein et al., 1989) the length of the boundary is minimised (forming either a circular or a band shaped structure, depending on geometry of the computational domain). At this point a steady state has been reached.

### 3.2 $\epsilon > 0$ (advective nonlinearity included)

We will now consider the system with advective nonlinearity in the moisture equation, $\epsilon > 0$. It will be instructive to begin with the evolution in one dimension, and the evolution of such a model with various parameter choices is shown in figure 4.

The case with $\epsilon = 0$ is shown for comparison purposes in Figure 4(a). The initial perturbation to RCE is linearly unstable, and the system rapidly segregates into moist and dry regions corresponding to the two stable values of $q$. This is the 1-D analogue of the initial stages of the 2-D evolution shown in Figure 2. However in the 1-D cases any continuing coarsening is extremely slow (exponentially small in the ratio of the length scale of moist and dry regions to the reaction-diffusion scale $(\kappa/\mu)^{1/2}$)





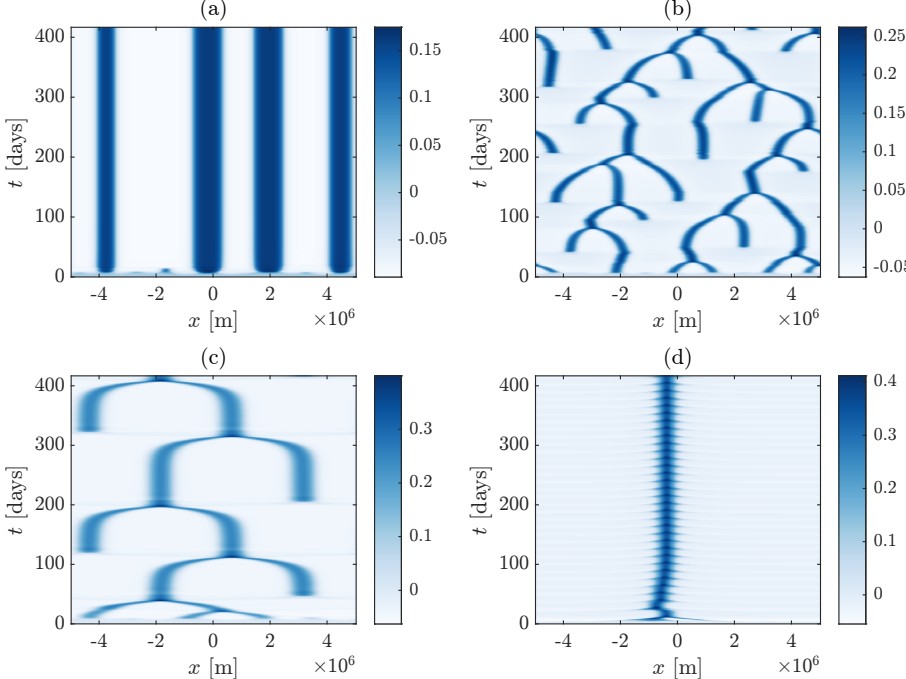

**Figure 4.** The spatial distribution of moisture, showing the evolution of $q/Q$ with time in a 1-D system for (a) the case $\epsilon = 0$, omitting advective nonlinearity, (b) case $\epsilon = 1$ with advective nonlinearity included, (c) as (b) but with $\kappa$ increased from $1 \times 10^5 \mathrm{m}^2 \mathrm{s}^{-1}$ to $4 \times 10^5 \mathrm{m}^2 \mathrm{s}^{-1}$, and (d) as (b), but with the WTG approximation applied explicitly.

since the curvature effect noted previously is absent. This is a well-known property of 1-D reaction-diffusion equations (e.g.
Fife and Hsiao, 1988).

For $\epsilon = 1$, shown in Figure 4(b), the system evolves initially as for $\epsilon = 0$. However once segregation has occurred the evolution of $q$ is determined not only by reaction-diffusion, but also by the effects of velocity convergence associated with moist regions and divergence associated with dry regions. In practice this corresponds to a relative narrowing of moist regions and relative widening of dry regions.

Once the system has segregated into separate moist and dry regions, these, in contrast to the case $\epsilon = 0$, aggregate through the effects of advection. Any part of the domain with a larger than average density of distinct moist regions is associated with a net convergence. The effect, as may be seen in Figure 4(b), is for the repeated merging of nearby moist regions. The time scale for this merging is proportional to $1/\epsilon D_-$, but note that if the separation between moist and dry regions was uniform then there would be no net convergence, i.e. the evolution could be considered as resulting from the instability of a perfectly
periodic array of moist and dry regions and the time scale $1/\epsilon D_-$ would then correspond to the inverse of the growth rate of the instability.

It may be seen that the aggregation does not lead to a steady state. Instead some of the merging of moist regions seem to be very closely followed by the emergence of new moist regions. This behaviour remains when the diffusivity is increased, as





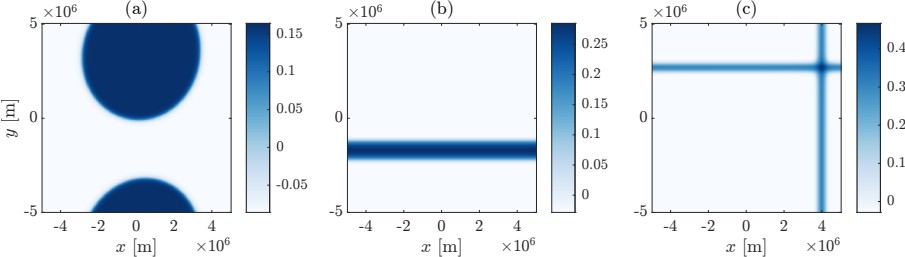

**Figure 5.** The steady state moisture distribution, showing plotting $q/Q$, in two dimensional simulations with no rotation or damping and (a) $\epsilon = 0$, (b) $\epsilon = 0.5$, and (c) $\epsilon = 1$

shown in Figure 4(c). The increased diffusivity implies increasing size of the individual moist regions, as might be expected, and it results in fewer moist regions across the domain as a whole, but it does not inhibit the emergence of new moist regions. The emergence seems to be triggered by gravity waves excited by the merging, which, due to the periodicity of the domain, meet one-another and excite a new moist region. This process then continues with new moist regions continually forming as the previous merge. The important role of gravity waves in this behaviour is supported by simulations in which the WTG approximation is made explicit, i.e. (2) is replaced by $\nabla \cdot \boldsymbol{u} = F_h(q) - \overline{F_h(q)}$ so that $\boldsymbol{u}$ is determined instantaneously by the $q$ field. The evolution for this case, for the diffusivity corresponding to Figure 4(b), is shown in Figure 4(d), where a single moist region is seen to form, with no emergence of new moist regions. However there is a persistent oscillation of the boundary of the single moist region, showing that even under WTG the inclusion of nonlinear advection introduces some subtleties that we have not yet understood.

Having established the implications of nonlinear advection in the 1-D setting we now return to the 2-D setting that is of most interest. In this case, whilst the merging of moist regions is clearly promoted by advection, there is not the same phenomenon of emergence of new moist regions, probably because any gravity waves excited by merger disperse radially and decrease in amplitude as they do so. Simulations show that the system evolves towards a steady state, as was the case for $\epsilon = 0$. However advection changes the geometry of this final steady state, as may be seen from the steady state distributions for different values of $\epsilon$ shown in Figure 5. Unlike with $\epsilon = 0$, where the steady state is governed only by reaction-diffusion, the final state must now also respect a global balance between reaction-diffusion and advection. On the square periodic domain considered in paper this leads to the moisture forming either a band or a cross shape. At smaller values of $\epsilon$ ($\epsilon = 0.5$ is shown) a band forms and at larger $\epsilon$ ($\epsilon = 1$ is shown) a cross is preferred. However at intermediate values either shape, according to details of the initial conditions, may be reached and will persist.

## 4 Breakdown of WTG and implications for aggregation

When frictional and thermal damping and rotation are included in the system, then, as noted previously, there will be an upper limit $L_{\mathrm{dyn}}$, depending on the dynamical parameters $\alpha$, $\lambda$ and $f$, on the scale to which WTG balance can apply. It is expected





that the coarsening to the domain scale exhibited in the previous section will be substantially modified, and perhaps halted, when the scale $L_{\text{dyn}}$ is reached. Since the growth of disturbances to the RCE state is the pre-cursor to the coarsening phase we will begin this section by considering in §4.1 explicit solution of the linear instability problem when $\alpha$, $\lambda$ and $f$ are non-zero. Then in §4.2 we will present semi-quantitative scaling arguments that are potentially relevant to the evolution beyond the linear instability phase. We then present results from numerical simulations in §4.3, both for $\epsilon = 0$ and for $\epsilon > 0$, including a regime diagram that summarises the overall pattern of behaviour as the dynamical parameters vary.

### 4.1 Key insights from the linear instability problem

There have been previous theoretical studies (e.g. Adames et al., 2019) of linear wave propagation and linear instability in systems equivalent to (1)–(3), but it is useful to establish some of the basic properties of the particular model system that we consider in this paper. For the case discussed in §2.2 and illustrated in §3, where $\alpha = \lambda = f = 0$ and the WTG approximation is valid, the linear stability properties of the RCE state are very straightforward and determined by the the sign of the derivative with respect to $q$ of $G_{hq}(q,0)$. We consider the linear stability problem in more detail for $\alpha$, $\lambda$ and $f$ non-zero since this gives insight into the behaviour of the full nonlinear system as revealed by numerical simulation.

Since the $f$-plane is isotropic, we can assume that perturbations vary only in the $x$-direction. Assuming small amplitude perturbations of the form $u = \Re\{\hat{u}\exp(\sigma t + ikx)\}$, with analogous notation for other variables, the linearised forms of (1), (2) and (3) are

$$\sigma\hat{u} - f\hat{v} = -ikg\hat{h} - \alpha\hat{u}, \tag{14}$$

$$\sigma\hat{v} + f\hat{u} = -\alpha\hat{v}, \tag{15}$$

$$\sigma\hat{h} - iHk\hat{u} = -\mu_2\hat{q} - \lambda\hat{h}, \tag{16}$$

$$\sigma\hat{q} - iQk\hat{u} = -\mu_1\hat{q} - \kappa k^2\hat{q}, \tag{17}$$

where $\mu_2 = -F_h'(0)$ and $\mu_1 = -F_q'(0)$, matching the notation used in (13) and (12). As is standard, these define an eigenvalue problem, the solution of which leads to a dispersion relation for the growth rate $\sigma$ which takes the form

$$\sigma^4 + [\lambda + 2\alpha + \mu_1 + \kappa k^2]\sigma^3$$

$$+ [(\mu_1 + \kappa k^2)(\lambda + 2\alpha) + f^2 + c^2k^2 + \alpha^2 + 2\alpha\lambda]\sigma^2$$

$$+ [(\mu_1 + \kappa k^2)(f^2 + c^2k^2 + \alpha^2 + 2\alpha\lambda) + c^2k^2\alpha + (f^2 + \alpha^2)\lambda - g\mu_2Qk^2]\sigma$$

$$+ (\mu_1 + \kappa k^2)c^2k^2\alpha - g\mu_2Qk^2\alpha + (f^2 + \alpha^2)\lambda(\mu_1 + \kappa k^2) = 0. \tag{18}$$

An important simple case is the strict WTG limit with $\alpha = \lambda = f = 0$. This may be considered directly by neglecting the $\sigma\hat{h}$ and $-\lambda\hat{h}$ terms in (16) and then substituting for $\hat{u}$ in (17) to deduce, neglecting the $\kappa k^2$ term,

$$\sigma = -\mu_1 + \frac{Q\mu_2}{H}. \tag{19}$$

Alternatively this may be deduced directly from the full dispersion relation, assuming a balance between the $\sigma^2$ and $\sigma$ terms, corresponding to $ck$ being large compared to $\mu_1$ and $\mu_2Q/H$, and neglecting the apparent $\sigma = 0$ root. This expression for $\sigma$




motivates the previously noted definition of the normalised gross moist stability for the moist shallow water equations,

$$M = 1 - \frac{Q\mu_2}{H\mu_1}. \tag{20}$$

As noted previously, in this paper parameter values will be chosen such that $M < 0$, implying $\sigma > 0$ and moisture mode instability with inverse time scale $\mu = \mu_1|M|$. Note that the WTG approximation applies at small scales, $k \gg \mu/c$. At large scales, with $k \ll \mu/c$, the moisture adjusts on a timescale shorter than that of the dynamics and a steady state balance in the moisture equation is more appropriate. (The large-$k$ and small-$k$ limits in this problem correspond to the moisture-mode and gravity-mode limits identified by Adames et al. (2019)). Using the moisture equation to eliminate the moisture dependence in

the height equation then gives the standard shallow water equations with the gravity wave speed adjusted from $c^2$ to

$$c^2(1 - \frac{\mu_2 Q}{H\mu_1}) = c^2 M. \tag{21}$$

This defines the moist gravity wave speed $c_m^2 = c^2 M$ and implies unstable growth rather than propagation if $M < 0$. $M < 0$ may be therefore identified as a criterion for instability whether or not the WTG approximation is valid. Re-including the effects of moisture diffusion will potentially inhibit instability on length scales comparable to or smaller than $\sqrt{\kappa/\mu_1}$.

Returning to the behaviour of the full system as described by (18), first consider the case where $\alpha$ and $\lambda$ are non-zero, but there is no rotation, $f = 0$. It is helpful to note the solutions for $\sigma$ in the large-scale ($k$ small) and small-scale ($k$ large) limits. Neglecting diffusion, it is straightforward to show that at small $k$ the roots of (18) are $\sigma \simeq -\lambda, -\alpha, -\alpha, -\mu_1$ (i.e. the $\sigma \simeq -\alpha$ root is repeated in the limit as $k \to 0$). Correspondingly at large $k$ there are roots $\sigma \simeq \pm ick - \lambda - \mu_2 Q/H, -\alpha, \mu_2(Q/H) - \mu_1 = -\mu_1 M$. There is therefore stability at small $k$ and, consistent with the WTG analysis, instability at large $k$ if $M > 0$ with the

latter limit unaffected by $\alpha$ and $\lambda$ being non-zero. Introducing diffusion by taking $\kappa > 0$ will not affect the small-$k$ behaviour but will affect the large $k$-behaviour.

Given that all roots for $\sigma$ have negative real part for $k \to 0$ it is possible to deduce whether or not there is instability by seeking conditions, in particular a value of $k$, under which one of the roots has zero real part. One possibility is $\sigma = is$ with $s$ real and non-zero. Substituting this form for $\sigma$ into (18), with $f = 0$, shows that such a root is not possible. (A key simplification

in this case $f = 0$ that can be exploited here is that 4th-order polynomial appearing in (18) has a factor $(\sigma + \alpha)$.) The other possibility is that $\sigma = 0$, implying the condition

$$(\mu_1 + \kappa k^2)c^2 k^2 \alpha - g\mu_2 Q k^2 \alpha + \alpha^2 \lambda(\mu_1 + \kappa k^2) = 0. \tag{22}$$

This is a quadratic for $k^2$ which has real roots only if

$$\kappa \leq \frac{c^2 \mu_1}{\lambda \alpha}(\sqrt{1-M} - 1)^2 \tag{23}$$

where as before $M = 1 - \mu_2 Q/\mu_1 H$. If this condition is satisfied then there is instability. If it is not satisfied then there is no instability. Furthermore it may also be deduced that roots with positive real part must be real, and that for any value of $k$ there can be at most one such root. Given that roots crossing the real axis must do so at $\sigma = 0$, complex roots with positive real part would be possible only if there were two such crossings, from below to above the real axis, as $k$ increased, and those real





roots then combined to give a complex conjugate pair. But this is not possible since the above shows that there are at most two crossings for any $k$ and, given that all roots are below the real axis for small $k$ and (with $\kappa > 0$) for large $k$, then one crossing is from below the real axis to above and the other is from above to below.

When $f$ is non-zero then the conditions for instability are more complicated. The large-$k$ limits of the roots of (18) are unaffected from the expressions given above (which were for $\kappa = 0$, the same holds for $\kappa > 0$. However the small-$k$ limits are now $\sigma \simeq -\lambda, \pm if - \alpha, -\mu_1$, implying stability at small $k$, but note that the previous repeated root $-\alpha$ now becomes the complex conjugate pair $\pm if - \alpha$ (i.e. frictionally damped inertial oscillations). The transition between small-$k$ and large-$k$ now depends on the values of $\alpha$, $\lambda$, $f$, $\kappa$ and other parameters.

The expression (23) for real roots of (18) to cross the real axis generalises to

$$\kappa \leq \frac{c^2 \mu_1 \alpha}{\lambda(\alpha^2 + f^2)}(\sqrt{1-M}-1)^2, \tag{24}$$

however this does not give complete information on when instability is possible because with $f$ non-zero complex conjugate roots may also cross the real axis. Some analytical progress can be made in describing the dependence of roots on the different parameters, but the algebra is complicated. Further details on analysis of (18) is given in Appendix B and the following describes and illustrates this dependence on the basis of a combination of analysis and numerical solution.

Figure 6 maps out different regions of the $(\alpha, \lambda)$ plane for a specific choice of $\kappa$ and for six different choices of $f$, including $f = 0$. Other parameters, $\mu_1$, $\mu_2$, $c^2$, $Q$ and $H$ take the same values as specified in §3. The $(\alpha, \lambda)$-plane may be divided into four regions, each corresponding to a different regime of behaviour. Regime I is where $\Re(\sigma) > 0$ occurs for some $k$ only when $\sigma$ is real. Regimes IIa and IIb are where there are complex $\sigma$ with $\Re(\sigma) > 0$ and with non-zero imaginary parts and Regime III is where there is no instability, i.e. $\Re(\sigma) < 0$ for all $k$. The distinction between IIa and IIb is that in IIb $\sigma$ corresponding to fastest growth over all $k$ has non-zero imaginary part.

For $f = 0$ only Regimes I and III is present and the region of instability simply corresponds to (23). For the largest value of $f$ shown, again only Regimes I and III are present, with Regime I corresponding to (24). For smaller non-zero values of $f$ the boundary between Regimes I and III is again described by (24), but, within $\alpha < f$, Regimes IIa and IIb exist. Note that Regimes IIa and IIb are therefore confined to smaller and smaller values of $\alpha$ as $f \to 0$.

Numerical investigation shows that the boundary between Regime I and Regime IIa, indicated by the dashed curves in Figure 6, is predicted by the condition that (18) has a double root at $\sigma = 0$. This is because close to the $\lambda = 0$ axis a real root crosses the real axis. As $\lambda$ increases a bifurcation point emerges on the solution branch joining to that root and then moves toward the real axis. A transition between crossing of the real axis by a real root and crossing by a complex conjugate pair occurs when the bifurcation point reaches the real axis, i.e. at the point in there is a double root at $\sigma = 0$. For all $f$ such that Regimes IIa and IIb exist the boundary between Regimes IIa and Regime I asymptotes to $\alpha \simeq \lambda$ as $\lambda \to 0$.

The condition for a double root at $\sigma = 0$ must be compatible with the condition (24). Given that the gradient of the dashed curves is observed to reduce as $\lambda$ increases, there will not be compatibility if the gradient of the curve defined by equality in (24) is larger than 1 as $\lambda \to 0$, i.e. if

$$f > c(\mu_1/\kappa)^{1/2}(\sqrt{1-M}-1). \tag{25}$$





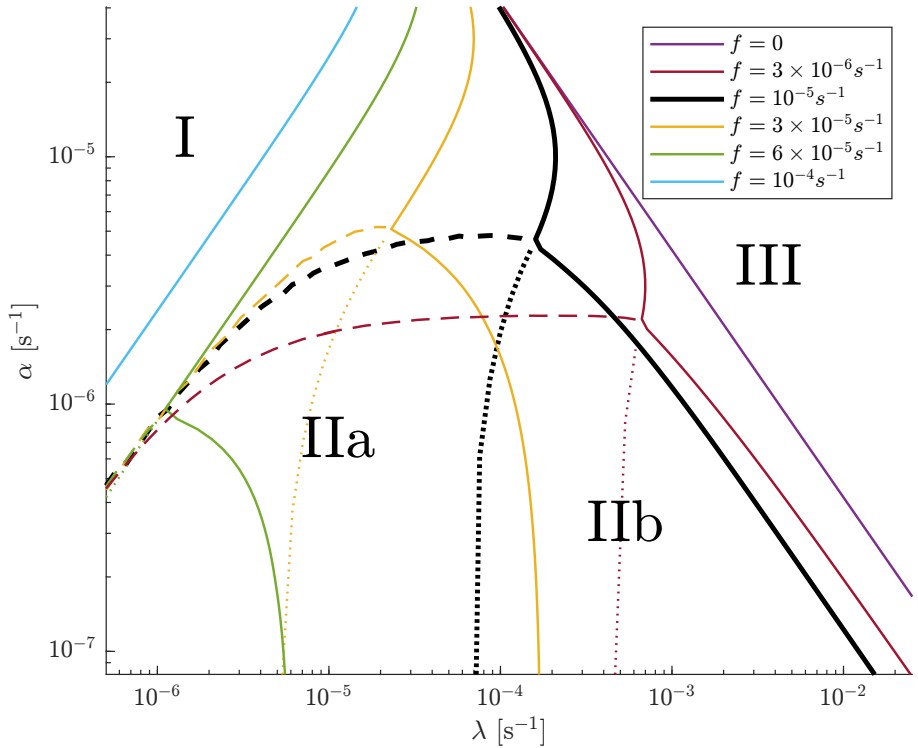

**Figure 6.** A regime diagram of the linear instability behaviour of the model on the $f$-plane. The black curves mark the boundaries between regimes for a value of $f = 10^{-5}\text{s}^{-1}$, and the black regime labels correspond to this curve. The solid black curve marks the boundary between the unstable regimes and the globally stable regime (Regime III). The dashed black curve denotes the boundary separating Regime I on the left, where all unstable modes have zero frequency and Regime IIa on the right, where some unstable modes have a non-zero frequency, and so are not stationary. The dotted black curve separates Regime IIa, to the left, from Regime IIb, to the right, where the fastest growing linear mode is no longer stationary. The curves of four other colours show corresponding boundaries for different values of $f$, with no equivalent of Regime II appearing when $f = 0$ or when $f$ is sufficiently large. A detailed description of the regime structure is given within the text.

When this inequality is satisfied there are no Regimes IIa and IIb and the transition between stability and instability is described completely by (24). For the parameters used to generate Figure 6, the disappearance of regions IIa and IIb between $f =$
$6 \times 10^{-5}\text{s}^{-1}$ and $f = 10^{-4}\text{s}^{-1}$ is consistent with (25).

The boundary between Regime IIb and Regime III corresponds to a double root of (18) on the imaginary axis. The 'quadruple point' where the regions corresponding to Regimes I, IIa, IIb and III touch corresponds to a double zero root of (18) coinciding with the condition for equality in (24). Both the IIb/III boundary and the quadruple point could in principle be described analytically but the algebra is daunting.

Whilst the parameter dependence that is found in the linear stability problem is complicated, two overall rules that seem to hold are first that (not surprisingly) increasing $\kappa$ inhibits instability and second that increasing $f$ also tends to inhibit instability, i.e. the region of the $(\alpha, \lambda)$ plane in which there is instability reduces as these parameters increase. In particular,





for any specified non-zero values of $\alpha$, $\lambda$ and $\kappa$ there is an $f_{\text{stab}}$ such that $f > f_{\text{stab}}$ implies stability. The solid curves shown in Figure 6 bounding Regime III are therefore, for the chosen value of $\kappa$, contours of the function $f_{\text{stab}}(\alpha, \lambda)$ in the $(\alpha, \lambda)$ plane.

The existence of $f_{\text{stab}}$ will be exploited in the description of the equatorial $\beta$-plane behaviour in the following section.

## 4.2 Dynamical arguments

The description above is focused on the linear instability problem and cannot be assumed to extend to the evolution once the growing unstable disturbances have saturated, e.g. in an aggregation phase. More general insight in the evolution can be obtained by considering possible balances in the equations at horizontal scale $L$. Assume that the time scale of evolution is

$T_q$. In the linear instability phase $T_q \sim \mu^{-1}$. However after the unstable growth has saturated $T_q$ may be larger than this. In the aggregation of moist and dry regions described previously, for example, $T_q$ is determined by the diffusivity $\kappa$ and is large if $\kappa$ is small.

If $\alpha \neq 0$ and $\lambda \neq 0$ then, assuming that $T_q \gg \alpha^{-1}, \lambda^{-1}$, a quasi-steady state balance is possible in the dynamical equations, i.e. $\boldsymbol{u}$ is instantaneously determined by the $q$ field according to the quasi-steady balance

$$\alpha\delta - f\zeta = -g\nabla^2 h \tag{26}$$

$$\alpha\zeta + f\delta = 0 \tag{27}$$

$$H\delta + \lambda h = F_h(q), \tag{28}$$

where $\delta$ is divergence and $\zeta$ is vorticity. In this respect we can identify this case with Regime I in the previous section. Eliminating $\zeta$ gives that

$$\delta = -g\alpha(\alpha^2 + f^2)^{-1}\nabla^2 h. \tag{29}$$

Substituting into (28) implies that the local $h$, and hence the local $\boldsymbol{u}$, is determined by $q$ in a surrounding region of scale

$$L_{\text{dyn}} = c(\alpha/\lambda)^{1/2}(\alpha^2 + f^2)^{-1/2}. \tag{30}$$

This defines a dynamical length scale $L_{\text{dyn}}$. WTG balance, the local balance between divergence and heating applies only length scales smaller than this. On length scales larger than $L_{\text{dyn}}$ the dominant balance in (28) is between $\lambda h$ and $F_h(q)$ and

hence, from (29) the divergence is proportional to $\nabla^2 F(q)$ rather than to $F(q)$. Another implication of the above balance, from (27), is that the flow will be dominated by the rotational component if $f \gg \alpha$ and by the irrotational component if $f \ll \alpha$.

The quasi-steady balance above cannot hold when $\alpha = 0$, when (29) would imply $\delta = 0$. However if $q$ is to be maintained away from the $q = 0$ steady state, which is known to be unstable, (4), neglecting the term multiplied by $\epsilon$ requires $\delta$ non-zero. This suggests that a distinct dynamical argument is required when $\alpha$ is small, analogous to the distinct nature of Regime

II discussed in the previous section. If $f \neq 0$, a possible balance assuming small Rossby number, i.e. that the time scale of





evolution of the $q$ field is much larger than $f^{-1}$, is

$$-f\zeta = -g\nabla^2 h \tag{31}$$

$$\zeta_t + f\delta = 0 \tag{32}$$

$$h_t + H\delta + \lambda h = F_h(q), \tag{33}$$

implying

$$(h - (c^2/f^2)\nabla^2 h)_t + \lambda h = F_h(q) \tag{34}$$

i.e. a form of the quasi-geostrophic potential vorticity equation with the potential vorticity changing through the effects of heating $F_h(q)$ and thermal damping. So in this system there are two prognostic equations, one for potential vorticity and one for moisture. In this case it is the second term in the time derivative that corresponds to divergence, so that WTG applies if

$L \ll (c/f)\min(1, (\lambda T_q)^{-1/2})$. Here the length scale appearing is the Rossby radius $c/f$ and the flow may be expected to evolve on the time scale $\lambda^{-1}$ at large times. Note that in this case the rotational part of the velocity field will be stronger than the divergent part.

In both the above cases it appears that the relation between moist heating and divergence becomes non-local at sufficiently large scale, $L_{\mathrm{dyn}}$ when $\alpha$ is large enough to bring the dynamical balance to a quasi-steady state and $c/f$ when $\alpha$ is smaller.

Therefore the aggregation behaviour seen previously is likely to be halted, or at least strongly modified, when these scales are reached. In §4.3 below the nature of this modification is examined by numerical simulation.

The scale $L_{\mathrm{dyn}}$ defined above decreases as $f$ increases, suggesting that the scale of aggregated moist and dry regions will also decrease as $f$ increases. Furthermore the underlying instability of the RCE state requires WTG dynamics to apply and $L_{\mathrm{dyn}}$ therefore also represents an upper limit on the scale of the instability. As $f$ increases $L_{\mathrm{dyn}}$ will reduce to the diffusion

scale $\sqrt{\kappa/\mu}$ and the instability of the RCE state will disappear. This provides an estimate for $f_{\mathrm{stab}}$,

$$\sqrt{\frac{\alpha}{\lambda}}\frac{c}{\sqrt{f^2 + \alpha^2}} = \Gamma\sqrt{\frac{\kappa}{\mu}}, \tag{35}$$

where $\Gamma$ is a non-dimensional parameter. The expression (24) provided by the linear stability calculation is consistent with this reasoning and provides an explicit expression for $\Gamma$ as equal to $\sqrt{-M}/(\sqrt{1-M}-1)$. For the case when $\alpha$ is small, corresponding to Regime II, we have no expression for $L_{\mathrm{dyn}}$ or for the boundary of stability between Regimes II and III in

figure 6 so a corresponding analytic result has not been determined. We do expect a qualitatively similar situation where the system being stabilised by diffusion once the maximum length scale $L_{\mathrm{dyn}}$ becomes sufficiently small. However the fact that the behaviour as $f$ changes of the boundary between Regime IIb and Regime III shown in Figure 6 is geometrically complicated suggests that a simply dynamical estimate of the form of the boundary will be difficult to find.

### 4.3 Numerical simulations

The link between the maximum length scale over which WTG is expected to apply and the spatial scale of aggregation is investigated numerically.





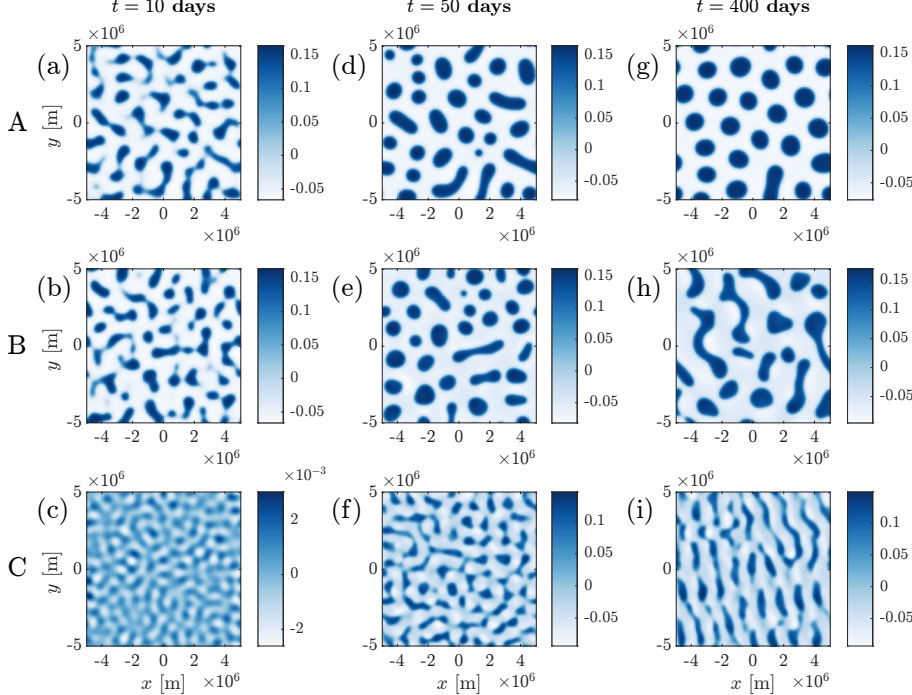

**Figure 7.** A series of snapshots of the perturbation moisture distribution $q/Q$ of a two dimensional simulation, this time with rotation and damping. Row A corresponds to case I in section 4.1. This has $f = 10^{-5}\mathrm{s}^{-1}$, $\alpha = \lambda = 4 \times 10^{-6}\mathrm{s}^{-1}$, giving $L_{\mathrm{dyn}} = 1.6 \times 10^{6}\mathrm{m}$. Rows B and C correspond to case II. Row B has $\alpha = 10^{-7}\mathrm{s}^{-1}$ and $\lambda = 10^{-6}\mathrm{s}^{-1}$, and C has $\alpha = 10^{-6}\mathrm{s}^{-1}$ and $\lambda = 10^{-4}\mathrm{s}^{-1}$.

The effect of non-zero $f$, $\alpha$ and $\lambda$ is now illustrated using numerical simulation, using the same numerical scheme as in section 3. Other parameters, $\mu_1$, $\mu_2$, $c^2$, $Q$ and $H$ take the same values as specified in §3, unless otherwise stated. $f$, $\alpha$ and $\lambda$ are chosen so that there is linear instability, corresponding to Regimes I, IIa and IIb in Figure 6. (Details of these simulations
are also given in Table 1.)

### 4.3.1 $\epsilon = 0$

We begin with $\epsilon = 0$, i.e. excluding nonlinear advection of moisture. A selection of time series of the moisture distribution of the system for different choices of $f$, $\alpha$ and $\lambda$ is shown in figure 7, with each simulation corresponding to a row. In all cases the moisture field $q$ was initialised with small-scale random noise. The first case, in row A, corresponding to Regime I in the $(\alpha, \lambda)$
plane shown in Figure 6 the system initially evolves similarly to the case without damping or rotation, with the formation of distinct moist regions, which evolve and enlarge through aggregation. However the aggregation does not proceed to the domain scale but halts at a smaller scale, with quasi-steady circular moist regions. This is as expected from the previous dynamical discussion. The inclusion of non-zero $f$, $\alpha$ and $\lambda$ implies that WTG balance can hold only up the scale $L_{\mathrm{dyn}}$ and aggregation halts at this scale. Other simulations with $f$, $\alpha$ and $\lambda$ corresponding to Regime I in Figure 6 show similar evolution.



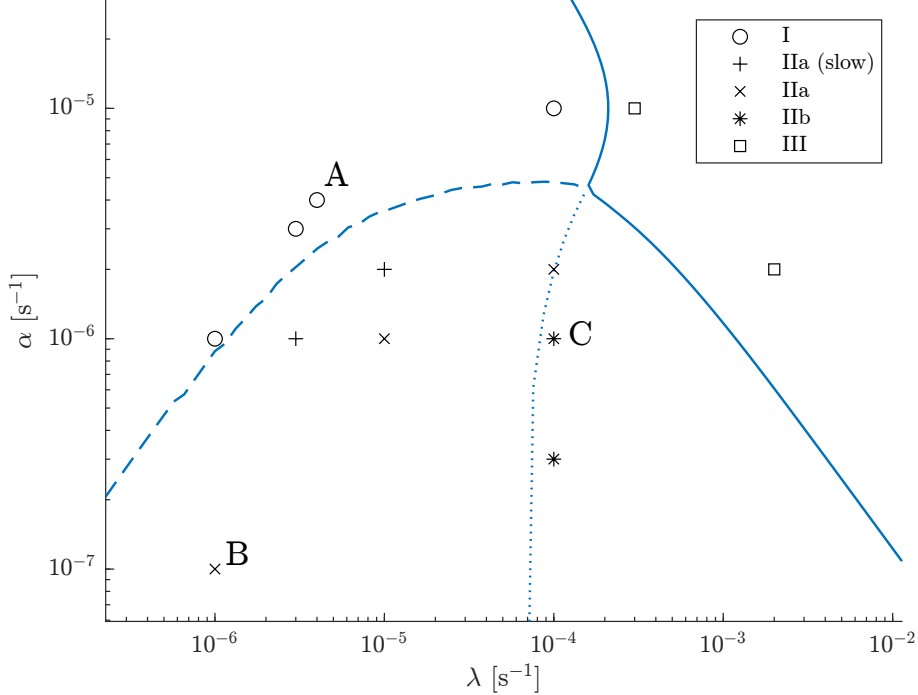

**Figure 8.** The regime diagram curve for $f = 10^{-5}\text{s}^{-1}$ from Figure 6, overlaid with observational data. Each point marked on the Figure correspond to the parameter values for a simulation which was then categorised into one of four regimes. The points corresponding to the moisture distributions shown in Figure 7 are labelled A, B and C.

Rows B and C of Figure 7 correspond to Regime II in the $(\alpha, \lambda)$ plane, B to IIa and C to IIb. For case B there is again an initial segregation and then aggregation process leading to distinct moist and dry regions at some finite scale. However, the long-time distribution is no longer stationary, but continues to evolve in time (without there being any further systematic increase in scale). In case C, whilst there is segregation, there is no clear aggregation stage and the moist and dry regions evolve in time in a manner that is more wave-like that that seen in case B.

To establish that the division of the $(\alpha, \lambda)$ plane, originally motivated by the linear instability properties, provides a useful guide to the behaviour of the ultimate nonlinear evolution, Figure 8 repeats the depiction of the $(\lambda, \alpha)$ plane shown previously, for a single value of $f$, with superimposed symbols indicating whether the nonlinear evolution was aggregated and quasi-steady, as case A above, aggregated and unsteady, as in case B, or propagating, as in case C. Regime IIa, (case B) has been split into two sub-regimes, with a slow regime corresponding to transitional behaviour in which aggregated regions form but

propagation is sufficiently slow that these remain round.

   A possible interpretation of the apparent relation between the properties of the linear instability problem and the evolution observed in the numerical simulations is as follows. In Regime I, as illustrated by simulation A, the linear instability behaviour is essentially that described by the WTG approximation, with the relevant unstable mode having real $\sigma$. Therefore the system





evolves through the the instability to the segregated state determined by the bistability. In Regime IIa, as illustrated by sim-
ulation B, the relevant unstable mode is similar to that in A, with $\sigma$ real and the process of segregation is correspondingly
similar. However the existence of slower growing propagating (complex-$\sigma$) unstable modes at larger scales is relevant to the
nonlinear evolution post-segregation (even if the linear instability modes themselves do not provide a complete description
of the behaviour). (A reduced mathematical model describing the evolution of the segregated state might make this relevance
clearer.) In Regime IIb, illustrated by simulation C, there is a pair of fastest growing modes with complex conjugate $\sigma$ rather
than a single mode with real $\sigma$ and the mechanism for growth is therefore completely different to that described by the WTG
approximation. In fact these modes are better considered as moisture-destabilized inertial waves, and are not moisture modes
since they rely on the fact that the dynamics is not slaved to the moisture field. Consequentially the nonlinear evolution is not
so clearly a segregation into the two states allowed by bistability and instead is better characterised as an evolving field of non-
linear moisture-inertial waves. Note that systematic propagation in this case C and the clear anisotropy of the instantaneous $q$
distribution visible in Figure 7(i) are an indication of spontaneous symmetry breaking rather than of any systematic anisotropy
of the system as specified.

We now focus on the behaviour of the model with parameters chosen from Regime I, where there is aggregation to a quasi-
steady state. The behaviour can be usefully summarised by using the spatial auto-correlation. The auto-correlation $L_{\text{auto}}$ scale
is defined as the minimum radius at which the spatial autocorrelation is a factor of $1/e$ less than its maximum value. The time
evolution of the $L_{\text{auto}}$ for simulations with various damping and rotation rates, and diffusivities, is shown in Figure 9.

The two cases (a) and (b) have $\alpha = \lambda = f = 0$ so, as previously demonstrated in §§2.2–3, aggregation is expected eventually
to proceeds to the domain scale. The evolution of $L_{\text{auto}}$ for both cases (a) and (b) is consistent with this expectation. For case
(b) $L_{\text{auto}}$ reaches a limiting value within the time period shown in the Figure. For case (a) a limiting value is not reached, but
$L_{\text{auto}}$ continues to increase throughout the period shown. The difference between (a) and (b) can explained by the fact that $\kappa$
for (b) is $4\times$ larger that for (a) and therefore, recalling the established theory on reaction-diffusion systems noted in §2.2, that
the rate at which aggregation proceeds is more rapid as $\kappa$ increases.

Other cases have non-zero values of $\alpha$, $\lambda$ and $f$. All these show approach to a finite limiting value indicating that aggregation
ceases. A candidate value for the length scale at which this occurs is $L_{\text{dyn}} = c(\alpha/\lambda)^{1/2}(\alpha^2 + f^2)^{-1/2}$. The corresponding values
deduced from Figure 9 are consistent with this expression in the sense that the ordering as parameters are changed is consistent
with the expression. Note in particular that for given $\alpha$ and $\lambda$ the scale is smaller with $f > 0$ than it is with $f = 0$.

The above has emphasised the evolution from an initial condition where the moisture distribution varies on small scales. It is
also interesting to consider the evolution from an initial condition where the moisture varies on large scales. In particular Figure
10 shows the evolution, with non-zero $\alpha$, $\lambda$ and $f$, from a state in which the initial condition on all fields is a domain-scale
aggregated state generated with $\alpha = \lambda = f = 0$ as previously shown in Figure 2(d). That state relies on the maintenance of $q$
away from the RCE state by the divergence which is determined by the WTG balance in the $h$-equation. When $\alpha$, $\lambda$ and $f$ are
non-zero the relation between divergence and $q$ changes, and WTG balance cannot be maintained on a scale larger than $L_{\text{dyn}}$. In
particular, in the case shown the divergence/convergence values are much smaller in the centre of large moist and dry regions,
where $q$ begins to relax towards the RCE state, i.e. $q = 0$. The system then becomes vulnerable to the radiative-convective



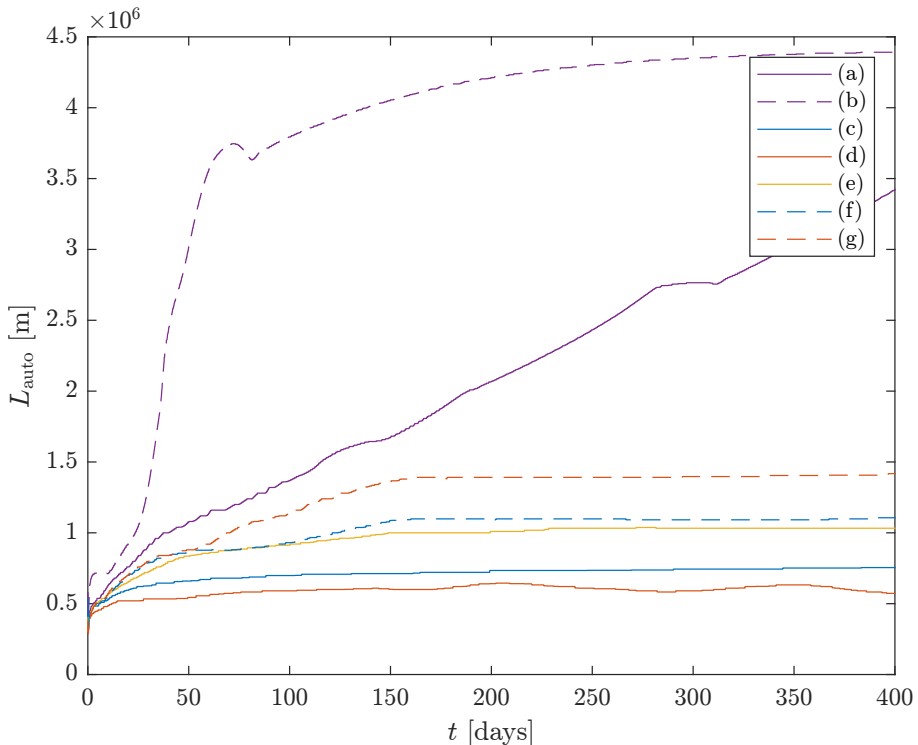

**Figure 9.** A measure of autocorrelation length scale plotted against time for various parameter values, within case I. Curves (a) and (b) have $\alpha = \lambda = f = 0$. Curve (a) has $\kappa = 10^5 \mathrm{m^2 s^{-1}}$ and (b) has $\kappa = 4 \times 10^5 \mathrm{m^2 s^{-1}}$. Curve (c) has $f = 10^{-5} \mathrm{s^{-1}}$ and $\alpha = \lambda = 4 \times 10^{-6} \mathrm{s^{-1}}$ and $\kappa = 10^5 \mathrm{m^2 s^{-1}}$. Curves (d) and (e) show the effect of reducing $\alpha$ and $\lambda$ respectively by a factor of 4. Curves (f) and (g) have parameters as (c) and (d) respectively, but with $f = 0$. The length scale varies consistently with the value of $L_{\mathrm{dyn}}$.

instability. New moist and dry regions at appear at smaller scales and the system begins to aggregate to a scale $L_{\mathrm{dyn}}$. This evolution does not depend on the inclusion of small scale noise in the initial condition; there is simply no large scale steady state possible.

### 4.3.2   $\epsilon > 0$ (advective nonlinearity included)

To understand the effects of nonlinear moisture advection on the behaviour of the system it is again instructive to start from a set of one-dimensional simulations. These are shown in Figure 11 for a particular choice of $\alpha$, $\lambda$ and $f$. The $\epsilon = 0$ case is shown

first, in Figure 11(a) for reference. A rapid evolution to coherent moist and dry regions, as was the case with $\alpha = \lambda = f = 0$, shown in Figure 4(a), but as expected the effect of $\alpha$, $\lambda$ and $f$ non-zero is that the scale of these regions is reduced. Whether or not the system is in an exact steady state or will continue to evolve further on a very long timescale has not been established, but what is more significant is the substantial modification when $\epsilon$ is non-zero, shown in Figure 11(b). The previously identified effect, recalling Figure 4(b), that nonlinear advection tends to narrow moist regions and, additionally to sweep them together





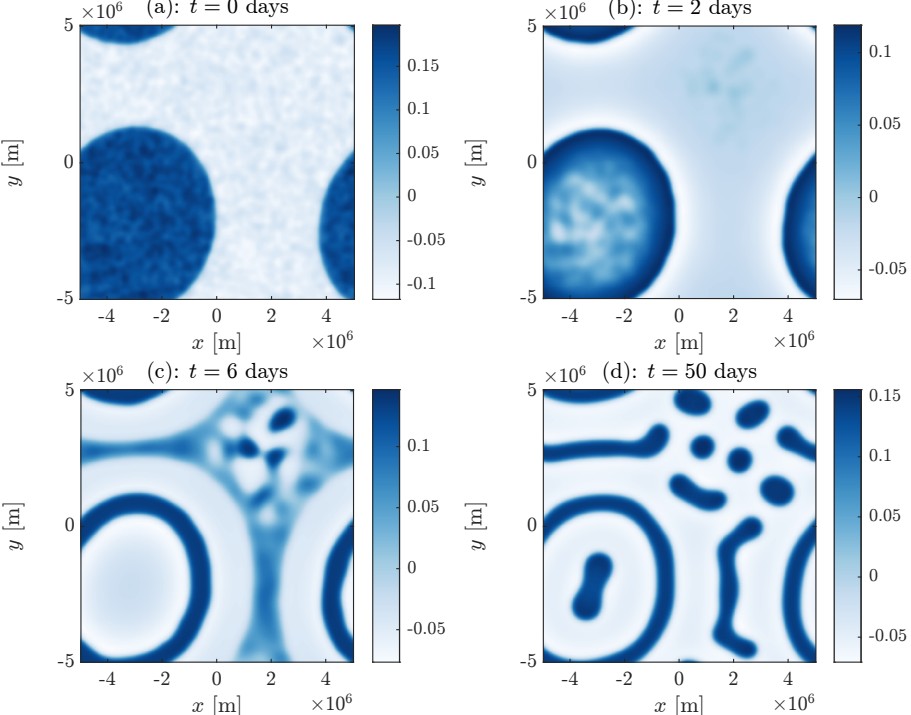

**Figure 10.** The snapshots of the normalised perturbation moisture field $q/Q$ against time for a numerical integration of the model equations with $f = 10^{-5}\mathrm{s}^{-1}, \alpha = \lambda = 4 \times 10^{-6}\mathrm{s}^{-1}$. This has been prepared from the final moisture state of a simulation with no damping and rotation superimposed with some small scale noise. The initial dynamics are the steady state response to the initial large scale moisture distribution. This should be compared to row A of figure 7, which has the same values of the parameters but is initialised from small amplitude small scale noise.

may be seen here to be remain when $\alpha$, $\lambda$ and $f$ are non-zero, however the resulting separation of moist regions is smaller. The phenomenon that 'collision' of moist regions, that leads to a broad dry region, is followed almost immediately to the formation of a new moist region in the centre of the dry region, is also seen here and indeed it also occurs when the dynamics is assumed to be a quasi-steady response to the moisture field, shown in Figure 11(d), suggesting that here this is not generated by a gravity wave but instead results from the fact that a broad dry region, with width greater than $L_{\mathrm{dyn}}$ cannot be maintained as a

steady state. (Recall discussion in the previous sections of the evolution from large-scale moist and dry regions.) Analogous to the behaviour noted for the 2-D system displayed in Figure 7C, the systematic propagation in the positive $x$ direction seen in Figure 11(b) and that in the negative $x$-direction seen in Figure 11(d) result from spontaneous symmetry breaking rather than any imposed asymmetry. In both cases the system is invariant under reflection in $x$.

    Figure 11(c) shows a case with larger diffusivity. Here the merging of moist regions is apparently slowed by the larger

diffusivity, this is presumably because the narrowing effect of advection is opposed by the large diffusion-reaction velocity. As moist regions come close to merging, one is seen to disappear, as seen in panel (c). This is interpreted as a result of the





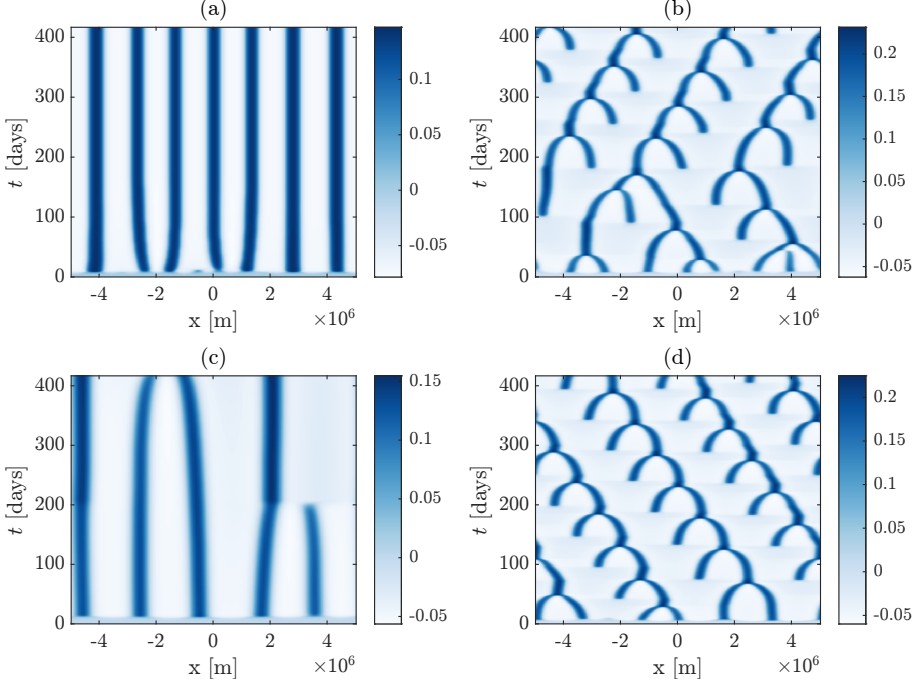

**Figure 11.** Moisture distribution, plotting $q/Q$, against time for a set of one-dimensional simulations with both nonlinear advection and damping, $f = \alpha = \lambda = 10^{-5} \mathrm{s}^{-1}$. (a) has $\epsilon = 0$, (b)–(d) have $\epsilon = 1$, (c) also has $\kappa$ increased by a factor of 4 from $10^{5} \mathrm{ms}^{-1}$ in the other simulations, and (d) has balanced dynamics to remove gravity waves.

reactive-diffusive coarsening, stronger because of the larger diffusion, removing the moist region before the merging action of nonlinear advection is complete.

We will now move on to investigating the two-dimensional system. A series of snapshots of the moisture field is shown in figure 12. Note that each of these cases have $(\lambda, \alpha)$ corresponding to Regime I. We have previously noted that the effect of advective nonlinearity is in the two-dimensional case to give moist regions that a more filamentary than quasi-circular. (For example recall the steady-state moisture distributions, where aggregation has proceeded to the domain scale, for $\alpha = \lambda = f$ shown in Figure 5.) The effect of $\alpha$, $\lambda$ and $f$ non-zero is both to limit any aggregation to a finite scale and to determine the flow pattern resulting from the moisture distribution. When $f \neq 0$ this flow has a substantial rotational component and the advective

effect of this on the filamentary moisture structures is apparent in Figure 12(a) and (b). The example with $f = 0$, shown in Figure 12(c), where the advecting flow is irrotational, is distinctly different, with any curvature of the filamentary structures weaker and resulting from deformation by a spatially structured irrotational flow. As has been noted previously advective narrowing of moist regions means that the maximum magnitude $q$ is affected by diffusion and not simply equal to the predicted value $q_{+}$.





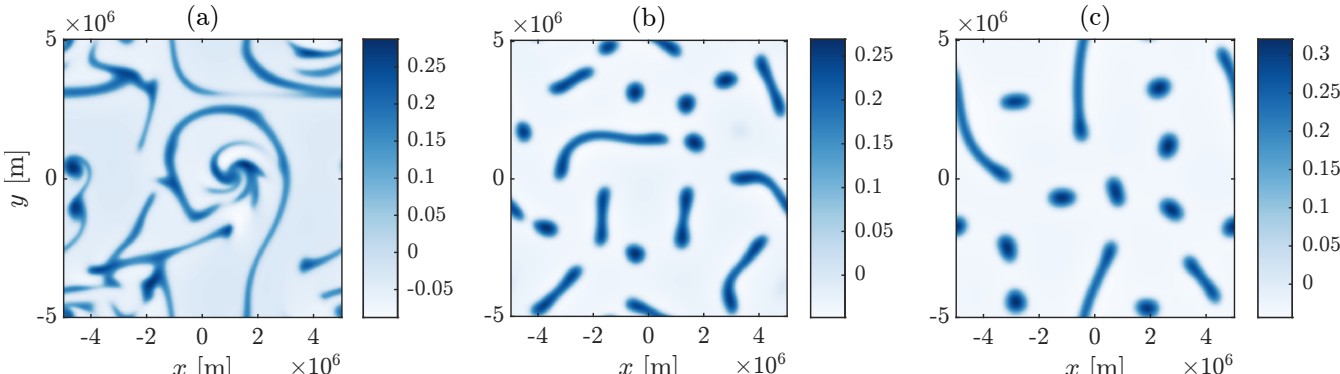

**Figure 12.** The perturbation moisture distribution, $q/Q$, after $10{,}000$ hours for a selection of two dimensional nonlinear simulations. (a) has $f = \alpha = \lambda = 10^{-6}\text{s}^{-1}$, (b) has $f = \alpha = \lambda = 10^{-5}\text{s}^{-1}$, and (c) has $\alpha = \lambda = 10^{-5}\text{s}^{-1}$ but $f = 0$.

## 5 Equatorial $\beta$-plane

In this section we will consider the model on the equatorial $\beta$-plane, i.e. with $f = \beta y$. It has been shown that on the $f$-plane aggregation tends to be inhibited by rotation, in two ways: (i) The upper limit of the scale for the underlying instability of the system is a decreasing function of $f$ and the lower limit is a increasing function of $\kappa$, therefore when $\kappa$ is non zero the instability disappears altogether for $f > f_{\text{stab}} = \beta y_{\text{stab}}$, with the latter equality defining $y_{\text{stab}}$. (ii) The upper limit on the aggregation scale is a decreasing function of $f$. This suggests the possibility on the $\beta$-plane of disturbances largely confined to some equatorial band with $|y| < y_{\text{stab}}$. Such disturbances do indeed form, and we will go on to describe their behaviour.

The regime diagram shown in Figure 6 in Sect. 4.1 provides some insight into how the dynamics might vary with latitude. At the equator, with $f = 0$, either Regime I or Regime III must apply, with Regime III implying that the RCE state is stable. For large $f$ Regime III applies. Whether or not the transition from Regime I to Regime III passes through Regime II will be determined by the values of $\alpha$ and $\lambda$. Generally speaking this will occur when $\alpha/\lambda$ is relatively small. If Regime II is visited then this is likely to be manifested as more complicated behaviour (recall Figure 7) as $y_{\text{stab}}$ is approached. However we will see later in this Section that there are effects on the $\beta$-plane that are not captured by the $f$-plane behaviour as described in Sect. 4.

It was noted in the previous section that, on the $f$-plane, whilst there was sometimes evidence of a selection of a preferred direction (recall Figures 7C and 11(b) and (d)), this selection is purely random. On the $\beta$-plane, however, there is a genuine east-west asymmetry (e.g. as manifested in the well-known Matsuno-Gill steady response to localised heating, which has been generalised to the case where $\alpha$ and $\lambda$ are not equal by Wu et al. (2001)) and it will be of particular interest to determine whether this leads to zonal propagation of moist and dry regions and how such propagation varies with model parameters.

We will begin this section discussing an adjustment to the previous, constant $f$ dynamical arguments and its impact on the local distribution of aggregated regions. We will then discuss the effects of the larger scale equatorial circulation. We will then go on to discuss the behaviour of a series of numerical experiments, in both the $\epsilon = 0$ and $\epsilon > 0$ cases.





### 5.1 Implications of equatorial $\beta$-plane dynamics

Much of the scale analysis of the $f$-plane equations presented in §4.2 was based on a quasi-steady balance in the dynamical equations which led to the relation (29) between $\delta$ and $h$ and hence an estimate $L_{\mathrm{dyn}} = c(\alpha/\lambda)^{1/2}(\alpha^2 + f^2)^{-1/2}$ for the scale on which WTG breaks down and hence as an effective upper limit on scale of aggregation. The same approach, of assuming a quasi-steady balance in the dynamical equations, will now be applied to the $\beta$-plane. The scale $L_{\mathrm{dyn}}$ as defined previously will still be useful, but will now vary with latitude. It is convenient to use the notation $L_{\mathrm{dyn},f}$ to represent the value of $L_{\mathrm{dyn}}$ for a particular value of $f$.

The balance (29) must be modified on the $\beta$-plane because $\partial f/\partial y$ is non-zero and becomes

$$\delta = -\frac{g\alpha}{\alpha^2 + f^2}\nabla^2 h + \frac{g\beta}{f^2 + \alpha^2}\hat{\boldsymbol{n}} \cdot \nabla h, \tag{36}$$

where $\beta = \partial f/\partial y$ and $\hat{\boldsymbol{n}} = (\alpha^2 - f^2, 2f\alpha)/(f^2 + \alpha^2)$ is a unit vector. Substituting (36) into the thickness equation, (2), gives the steady response to a heating as

$$\lambda h + H\delta = \lambda h - \frac{gH\alpha}{\alpha^2 + f^2}\nabla^2 h + \frac{gH\beta}{f^2 + \alpha^2}\hat{\boldsymbol{n}} \cdot \nabla h = F_h(q). \tag{37}$$

One important difference from the corresponding equations for the $f$-plane is there is now a preferred direction in the relation between $\delta$ and $h$, allowing a systematic anisotropy. A second difference is that the coefficients in the equations are now functions of $y$. A local analysis, treating $f$ as constant, may therefore not always be valid. The expressions above suggest a change in character of the system when the length scale is larger than $\alpha/\beta$. Below this scale the first, isotropic, term in (36) is dominant, and, furthermore, the coefficient appearing in this term does not vary significantly on this scale, so a local description will be valid. For length scales larger than $\alpha/\beta$, however, the second term is dominant, suggesting significant anisotropy in the dynamics. The coefficients and the vector $\hat{\boldsymbol{n}}$ will also vary significantly with so significant latitudinal variation is expected and, more fundamentally, a local description may not be valid.

Along with the above it must be taken into account that aggregation is initiated by instability of the RCE state and that this instability is confined to $|y| < y_{\mathrm{stab}}$. Additionally, whilst it is expected that moist anomalies are large confined within this region, note that (37) implies that dynamical effects extend outside, on a length scale of $L_{\mathrm{dyn},f=\beta y_{\mathrm{stab}}}$. When $M$ is large and negative, (35) suggests this scale is $\sqrt{\kappa/\mu}$: a diffusive response extending outside of the unstable equatorial region. When $-M$ is closer to zero, however, this length scale may be significantly larger.

The $f$-plane analysis predicts $L_{\mathrm{dyn}}$ as an upper limit on the scale for aggregation. This suggests $L_{\mathrm{dyn},f=0} = c\sqrt{\lambda/\alpha}$ as the corresponding scale at the equator on the $\beta$-plane. Therefore, on the basis of the arguments above, aggregation at the equator may be isotropic if $L_{\mathrm{dyn},f=0} < \alpha/\beta$ and $y_{\mathrm{stab}} > L_{\mathrm{dyn},f=0}$. If the second condition is not satisfied then the geometry will not allow isotropy. Furthermore, since $L_{\mathrm{dyn},\beta y}$ is a decreasing function of $y$, isotropy will extend to $y = y_{\mathrm{stab}}$ but the characteristic length scales will decrease as $|y|$ increases. In other words we expect the aggregation to be qualitatively similar to the $f$-plane case, except that it will be confined to a region slightly larger than $|y| < y_{\mathrm{stab}}$ and that the characteristic length scale with vary with $y$. If $L_{\mathrm{dyn},f=0} < \alpha/\beta$ and $y_{\mathrm{stab}} < L_{\mathrm{dyn},f=0}$, on the other hand, then the aggregated structures in moisture will be largely confined to $|y| < y_{\mathrm{stab}}$ and therefore extended in $x$ relative to their scale in $y$. However the dynamical signatures will extend to



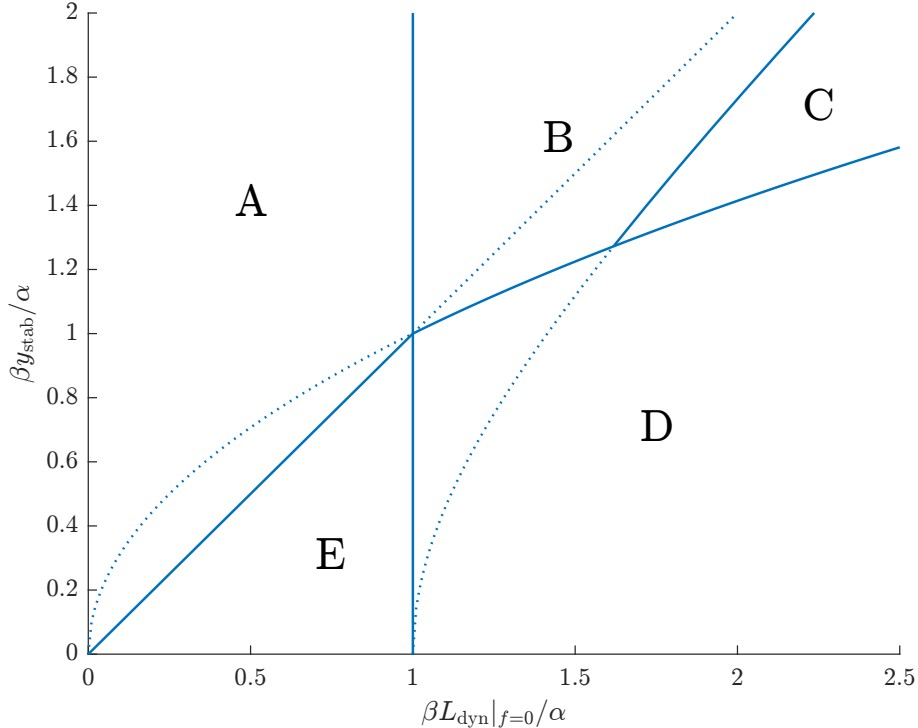

**Figure 13.** A schematic plot of the expected regimes of behaviour of our model on the equatorial $\beta$ plane, shown against varying the latitudinal scale of the equatorial wave response $L_{\mathrm{dyn}}|_{f=0} = c/\sqrt{\alpha\lambda}$ and the latitudinal limit of bistability $y_{\mathrm{stab}}$, both in units of $\beta/\alpha$. Solid lines denote regime boundaries, and the remainder of each curve is dotted. The straight line dividing A from E is given by $y_{\mathrm{stab}} = L_{\mathrm{dyn}}|_{f=0}$. The curve dividing B and C from D is given by $y_{\mathrm{stab}} = L_{\mathrm{eq}}$. The curve separating B from C is given by $y_{\mathrm{stab}} = y_{\mathrm{iso}}$. A brief summary of the characteristics of each Regime is as follows. Regime A: Local $f$ plane analysis valid, isotropic aggregated regions extend to $y_{\mathrm{stab}}$. Regime B: Aggregated regions on equator with $y$-scale $L_{\mathrm{eq}}$, transition to isotropic aggregrated regions at larger $|y|$. Regime C: Aggregated regions on equator with $y$-scale $L_{\mathrm{eq}}$, anisotropic aggregated regions at larger $|y|$. Regime D: Aggregated regions centred on equator with $y$-scale $L_{\mathrm{eq}}$. Regime E: Aggregated regions centred on equator with $y$-scale $y_{\mathrm{stab}}$. In Regimes A, B and C there are multiple aggregated regions in latitude. Regions at different latitudes propagate in the $x$-direction at different velocities. In Regimes D and E all aggregated regions are centred on the equator and there is coherent propagation in the $x$-direction.

$|y| = L_{\mathrm{dyn},f=0}$. This suggests two distinct regimes of behaviour in a parameter space defined by $y_{\mathrm{stab}}\beta/\alpha$ and $L_{\mathrm{dyn},f=0}\beta/\alpha = c\beta\alpha^{-3/2}\lambda^{-1/2}$. We will label these as Regime A ( $L_{\mathrm{dyn},f=0} < \alpha/\beta$, $L_{\mathrm{dyn},f=0} < y_{\mathrm{stab}}$) and Regime E ( $L_{\mathrm{dyn},f=0} < \alpha/\beta$, $y_{\mathrm{stab}} < L_{\mathrm{dyn},f=0}$ ). These two Regimes are shown in the $(L_{\mathrm{dyn},f=0}, y_{\mathrm{stab}})$ plane in Figure 13. We will add further regimes to this diagram and discuss them in more detail below.

We now consider the case where $L_{\mathrm{dyn},f=0} > \alpha/\beta$. The non-isotropic terms in (36) and (37) must be taken into account. Furthermore close to the equator a local analysis is no longer adequate. A natural $y$-scale close to $y = 0$, obtained by requiring a balance between the first and second terms in the middle expression in (37) is $L_{\mathrm{eq}} = (c/\beta)^{1/2}(\alpha/\lambda)^{1/4}$ as obtained by





Wu et al. (2001) in their generalisation of the Matsuno-Gill problem. The corresponding $x$-scale, however, agrees with the length scale from the local analysis at the equator, $L_{\mathrm{dyn},f=0}$. Note that $L_{\mathrm{eq}} = \sqrt{L_{\mathrm{dyn},f=0}\,\alpha/\beta}$ and that $L_{\mathrm{eq}}$ therefore always lies between $\alpha/\beta$ and $L_{\mathrm{dyn}}$, i.e. when $L_{\mathrm{dyn},f=0} > \alpha/\beta$, $\alpha/\beta < L_{\mathrm{eq}} < L_{\mathrm{dyn},f=0}$. The structure of (37) implies that a moisture anomaly localised within region $|y| < L_{\mathrm{eq}}$ will force a dynamical anomalies that extend across the region $|y| \sim L_{\mathrm{eq}}$. This
implies a distinct Regime D ( $\alpha/\beta < L_{\mathrm{dyn},f=0}$, $y_{\mathrm{stab}} < L_{\mathrm{eq}}$) similar to Regime E, with latitudinal confined moisture anomalies driving broader dynamical anomalies.

Now consider ($L_{\mathrm{dyn},f=0} > \alpha/\beta$, $y_{\mathrm{stab}} > L_{\mathrm{eq}}$). In this case, since $y_{\mathrm{stab}} > L_{\mathrm{eq}}$, the unstable region is large enough to allow multiple aggregated structures in the $y$-direction. Since we have $L_{\mathrm{dyn},0} > \alpha/\beta$ there is anisotropy at the equator, but since $L_{\mathrm{dyn}}$ is a decreasing function of $f$ the anisotropic region is expected to extend only to $y$ such that $L_{\mathrm{dyn},\beta y} = \alpha/\beta$, hence

$$y = y_{\mathrm{iso}} = L_{\mathrm{dyn},f=0}\left(1 - \frac{\alpha^2}{\beta^2 L_{\mathrm{dyn},f=0}^2}\right)^{\frac{1}{2}}, \tag{38}$$

which defines $y_{\mathrm{iso}}$. For $|y| > y_{\mathrm{iso}}$ we expect isotropic aggregation. This implies two further regimes, Regime B ( $L_{\mathrm{dyn},f=0} > \alpha/\beta$, $y_{\mathrm{stab}} > \max\{L_{\mathrm{eq}}, y_{\mathrm{iso}}\}$) in which there is anisotropic aggregation in $|y| < \max\{L_{\mathrm{eq}}, y_{\mathrm{iso}}\}$ and isotropic aggregation in $\max\{L_{\mathrm{eq}}, y_{\mathrm{iso}}\} < |y| < y_{\mathrm{stab}}$ and Regime C ( $L_{\mathrm{dyn},f=0} > \alpha/\beta$, $L_{\mathrm{eq}} < y_{\mathrm{stab}} < y_{\mathrm{iso}}$) in which there is anistropic aggregation across the entire region $|y| < y_{\mathrm{stab}}$.

Figure 13 shows all five of the Regimes A–E. It should additionally be noted that if $y_{\mathrm{stab}}$ is very small compared to dynamical length scales then instability may be inhibited. This might justify defining a further distinct regime, but since the priority has been to interpret behaviour seen in cases where there is instability the criteria for such a regime has not been investigated in detail.

Having identified the different dynamical regimes that characterise the system on the $\beta$-plane we now consider the implica-
tions for spatial propagation of the aggregated moist and dry regions. It is useful to consider the moisture evolution equation. Using (28) to substitute for $\nabla \cdot \boldsymbol{u}$, (4) becomes

$$q_t + \epsilon \nabla \cdot (\boldsymbol{u}[q]q) - \kappa \nabla^2 q = F_q(q) - \frac{Q}{H}(F_h(q) - \overline{F_h(q)}) + \frac{\lambda Q}{H}(h - \overline{h}). \tag{39}$$

This differs from the WTG form (7) by the final term on the right-hand side, which is non-zero unless $h$ is spatially uniform. Note that the contributions to $q_t$ that arise under WTG are not expected to lead to systematic propagation since the relation
between these terms and $q$ is isotropic.

The extra term $(\lambda Q/H)(h - \overline{h})$ can potentially cause systematic propagation if its relation to $q$, as expressed by (37) is anisotropic. As has been noted previously this relation is isotropic on the $f$-plane, implying no propagation in that case. Under circumstances where a local analysis of (37) is appropriate, it is useful to exploit the analogy between (37) and a damped advection-diffusion equation, with a source term $F_h(q)$. The advecting velocity is $gH\beta/(f^2 + \alpha^2)\hat{\boldsymbol{n}}$. For a positive $q$ anomaly,
given that $F_h(q)$ is negative, $h$ will therefore be negative in the direction $\hat{\boldsymbol{n}}$ and hence, according to (39), $q_t$ will be negative in the $\hat{\boldsymbol{n}}$ direction and positive in the $-\hat{\boldsymbol{n}}$ direction, implying propagation of the $q$ anomaly in the $-\hat{\boldsymbol{n}}$ direction. This direction is westward if $f < \alpha$ and eastward if $f > \alpha$.





Local analysis of (37) applies for all $y$ in Regimes A and E. These have been identified previously as isotropic at leading-order, but there is weak anisotropy and this will lead to westward propagation of aggregated moist regions close to the equator.

It follows that a sufficient condition for westward propagation at the equator is $L_{\mathrm{dyn},f=0} < \alpha/\beta$. In case A the instability and hence aggregation, may extend to $y$ such that $|f| > \alpha$, i.e. $|y| > \alpha/\beta$, and there propagation will be eastward. The same analysis is relevant to regions such that $|y| > y_{\mathrm{iso}}$ in Regime B. Since this corresponds to $|f| > \alpha$ the propagation in these regions will be eastward.

Since the equatorial region $|y| < L_{\mathrm{eq}}$ in Regimes A, B and C is broader than $\alpha/\beta$, and the direction of the vector $\hat{\boldsymbol{n}}$ will

change direction when varied across this region, the implication of the local analysis is that there will be a relative westward shift of the moisture anomalies near the equator and a relative eastward shift elsewhere, generating a "<" shape. The overall propagation speed is likely to depend on the latitudinal extent of the equatorial moist region, i.e. on the ratio $L_{\mathrm{eq}}\beta/\alpha$. A wider region, with a larger value of this ratio will have a larger proportion in which $-\hat{\boldsymbol{n}}$ is directed in the eastward region shift, and hence is likely to have greater more eastward propagation.

The dependence of propagation speed on latitude suggests that when there are multiple aggregated structures in latitude, i.e. in Regimes A, B and C, there will not be a single propagation speed. If there is a dominant speed for structures at the equator then that will be different from, and more westward, those that for structures at higher latitudes.

An alternative way to understand the behaviour near the equator in Regimes B, C and D is through decomposition into equatorial waves, following Wu et al. (2001). This offers a different approach to describing solutions of (37), for $h$ given

$F_h(q)$ that does not require regarding the coefficients as locally constant. For reference, the method of calculation is set out in Appendix C. In this case the correction $(\lambda Q/H)(h - \overline{h})$, appearing in (39), to the WTG convergence associated with an equatorial heating anomaly will be divergent responses to the east from Kelvin waves, acting to shift the moist anomaly to the west, and to the west from Rossby waves, acting to shift the moist anomaly to the east. It is the relative size of these two wave responses that determines the net propagation of moist regions in this system, with a strong Rossby wave response implying

eastward propagation. The fact that the Kelvin wave response decays away from the equator on a scale $L_{\mathrm{eq}}$ whereas the Rossby wave reponse can be excited at any latitudinal scale suggests that the Kelvin wave response will be relatively weaker and therefore there will be eastward propagation if moist regions are significantly wider that $L_{\mathrm{eq}}$ (though it should be noted that it has previously been argued that $L_{\mathrm{eq}}$ is the natural latitudinal scale of aggregated moist anomalies). More specifically the fact that the Kelvin wave response is likely to be more localised near the equator than the Rossby wave response suggests that moist

anomalies will tend to have a '<' shape, supporting the prediction of this shape from the more heuristic local analysis. This role of the Kelvin and Rossby wave reponses in the propagation mechanism has previously been identified by Sugiyama (2009a, b).

The arguments presented above may be used to formulate a quantification of contributions of different processes to zonal propagation that may be applied to the simulations discussed in the following section. This quantification is useful where the disturbances may be considered to be coherently propagating at a speed $U$, which is possible when the latitudinal structure

of the disturbances is dominated by a single moist anomaly centred on the equator, corresponding to Regimes D and E. In a reference frame moving at constant speed $U$ in the zonal direction, the moisture distribution will then be steady and the





moisture equation (4) may be re-written as

$$Uq_x = -Q\nabla.\boldsymbol{u} - \epsilon\nabla.(q\boldsymbol{u}) + F_q + \kappa\nabla^2 q. \tag{40}$$

Multiplying by $q_x$ and integrating over the domain implies that the propagation speed $U$ is given by

$$U = \iint \left( -Q\nabla.\boldsymbol{u} - \epsilon\nabla.(q\boldsymbol{u}) + F_q + \kappa\nabla^2 q \right) q_x \, dA \Big/ \iint q_x^2 \, dA. \tag{41}$$

This may be interpreted as an expression of the propagation speed as a sum of contributions from different individual terms in the moisture equation. We can exploit this, decomposing the divergence into a weak temperature gradient part $D_{WTG}$ defined from (6) and its departure from WTG, $D'$.

$$U = \iint \left( -QD' - \epsilon\nabla.(q\boldsymbol{u}) - Q/H(F_h - \bar{F}_h) + F_q + \kappa\nabla^2 q \right) q_x \, dA \Big/ \iint q_x^2 \, dA. \tag{42}$$

Since $F_q$ and $F_h$ are functions of $q$ only and $\bar{F}_h$ is constant, the corresponding terms in the integral are total derivatives and so vanish when integrated over a periodic domain. We can also write the diffusive term as $q_x\nabla^2 q = (\frac{1}{2}[q_x^2 - q_y^2])_x + (q_xq_y)_y$. The diffusive term therefore also does not contribute to the integral. The coherent propagation of moist regions can therefore decomposed into parts due to the departure of the divergence from WTG and nonlinear advection.

$$U = \iint \left( -QD' - \epsilon\nabla.(q\boldsymbol{u}) \right) q_x \, dA \Big/ \iint q_x^2 \, dA. \tag{43}$$

Other terms in the moisture equation do not contribute. Note however that this does not mean that the propagation speed is independent of the value of $\kappa$, for example. Diffusivity still plays a role in setting the width of the unstable region $|y| < y_{\text{stab}}$ and the shape of individual moist regions. The term including $D'$ may be further decomposed into separate contributions from Rossby and Kelvin wave parts of the dynamical response to the heating implied by the moisture field. The method for calculating the Kelvin and Rossby wave responses is set out in Appendix D.

## 5.2   Numerical Simulations

As with the $f$-plane case previously, much understanding of the behaviour on an equatorial $\beta$-plane can be gleaned from numerical simulations. Numerical details are again in Appendix A and details of parameter values for simulations for which results are displayed are included in Table 1.

A particular focus in analysing the simulations will be on the zonal propagation of disturbances. For each simulation a zonal
phase speed $U$ may be estimated by identifying the dominant zonal wavenumber in the moisture field at the equator and then tracking its time evolution. In some cases, where the latitudinal structure of the disturbances is dominated by a single moist anomaly centred on the equator, the disturbances may be considered to be steadily propagating. When the latitudinal structure is more complicated, with multiple moist regions at different latitudes, the assumption of steady propagation is not applicable, since different moist regions may propagate at different speeds. Regions further from the equator are associated with more
rapid eastward propagation. A notional phase speed $U$ can instead be calculated as the mean phase speed of the dominant mode in the moisture evolution along the equator.



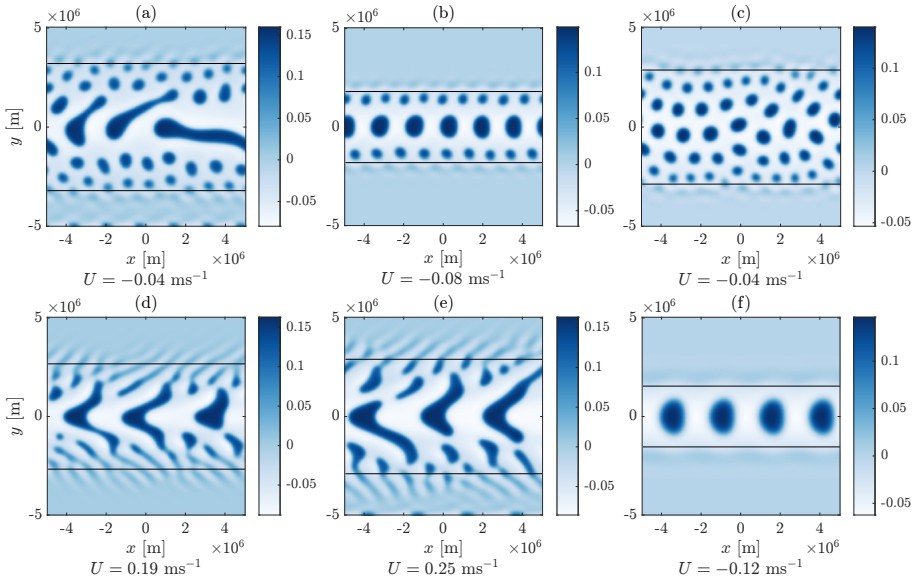

**Figure 14.** A selection of perturbation moisture distributions $q/Q$ from various beta plane simulations, annotated with their propagation speed. (a) has damping rates $\alpha = \lambda = 10^{-5}\mathrm{s}^{-1}$. (b) is as (a) but with $\lambda$ increased by a factor of 3, this has $\alpha = 10^{-5}\mathrm{s}^{-1}$, $\lambda = 3 \times 10^{-5}\mathrm{s}^{-1}$.. (c) then has $\alpha$ also increased by a factor of 3, to $\alpha = \lambda = 3 \times 10^{-5}\mathrm{s}^{-1}$. (d) is as (a) but with $\alpha$ reduced by a factor of 10 for $\alpha = 10^{-6}\mathrm{s}^{-1}$, and $\lambda = 10^{-5}\mathrm{s}^{-1}$. (e) then has $\lambda = 3 \times 10^{-6}\mathrm{s}^{-1}$, and $\alpha = 10^{-6}\mathrm{s}^{-1}$. (f) is as (a) but with the diffusivity $\kappa$ increased by a factor of 4 to $\kappa = 4 \times 10^{5}\mathrm{m}^2\mathrm{s}^{-1}$. The red horizontal lines denote $y = y_{\mathrm{stab}}$.

### 5.2.1 $\epsilon = 0$

We will again start with the case where nonlinear advection is excluded. The results of a set of simulations for various diffusivities and damping rates such that $L_{\mathrm{eq}} < y_{\mathrm{stab}}$ are shown in Figure 14. In all panels the red horizontal lines denoting $y = y_{\mathrm{stab}}$,

calculated on the basis of the $f$-plane linear instability analysis, mark the boundary between the low-latitude region of instability of the RCE state and the higher latitude region where RCE is stable. The early time evolution and up-scale growth is similar to the constant-$f$ cases, with the scale of the late time aggregated regime larger than that of the disturbances that emerge from small-scale noise during the initial unstable growth. In this figure only panel (f) corresponds to steady propagation, so for comparison purposes in all panels the speed has been calculated as the mean phase speed of dominant wavenumber zonal mode

at the equator. As in the $f$-plane cases, the timescale for upscale growth becomes very slow at large scales. For consistency, the plots here are all shown after 400 days of simulation even though there may at that time be continuing systematic growth of the spatial structure.

The spatial distribution of aggregated regions is consistent with the previous discussion of the $f$-plane behaviour, in §4.2, developed further for the $\beta$−plane in §5.1. At the equator, where $f = 0$, we expect the scale of aggregated regions to follow

$c/\sqrt{\alpha\lambda}$, up to quantisation by the domain size. This is consistent with the structure seen in panels (a)–(c), where the predicted scale decreases by a factor of $\sqrt{3}$ between each panel and the zonal scale decreases accordingly. Also as expected, the scale



of aggregated regions decreases away from the equator as $f$ increases. This is especially clear in panel (c) and also visible in panels (a) and (b).

In panels (a), (b), (d), (e) and (f) the spatial scale of moist regions at the equator is sufficiently large, $L_{\mathrm{dyn}, f=0} > \alpha/\beta$, so
that the anisotropic term becomes dominant in (36). According to the classification defined in figure 13 Panels (a) and (b) reside in Regime B. Consistent with this a region of non-isotropic aggregation forms near the equator and extends to over part (corresponding to $|y| < y_{\mathrm{iso}}$, of the unstable region, though in panel (b) the anisotropic effect is weak. The anisotropic equatorial moist regions are locally shifted in the local direction of $-\hat{\boldsymbol{n}}$, in contrast to the quasi-circular structures seen at larger $|y|$ where isotropy applies. The increase in damping rates from panel (b) to panel (c), so that (c) lies in Regime A,
decreases the scale of equatorial moist regions sufficiently that the aggregation is isotropic across the entire unstable region.

Panels (d), (e) and (f) reside in regime C and the extension of the anisotropic region up to $|y| = y_{\mathrm{stab}}$ is consistent with this. Panel (f) is an example of the system with increased diffusivity and all other parameters the same as those in panel (a). The main differences in (f) relative to (a) are the decrease in $y_{\mathrm{stab}}$, and the slight increase in scale due to the increased with of the diffusive boundaries between the moist and dry states. Panel (f) shows, in some sense, marginal behaviour between regimes
C and D. Whilst the aggregated regions near the equator are confined to $|y| < y_{\mathrm{stab}}$, they are sufficiently wide that no further disturbances form at larger $y$. The aggregation takes the form of a series of uniformly propagating regions, qualitatively similar to regime D.

Evidence of the $f$-plane Regimes may also be seen in the change in structure with latitude. Panels (a)–(c) and (f) all have a direct transition from Regime I to III at $y = y_{\mathrm{stab}}$, and accordingly aggregated regions remain circular up to this boundary.
Panels (d) and (e), however, have an intermediate range of $y$ for which the local $f$-plane behaviour is in Regime II. The moist and dry regions near the boundaries in these cases are no longer circular and are far more transient, similar in character to the structure seen in figure 8C.

A case with $L_{\mathrm{eq}} > y_{\mathrm{stab}}$ is shown in Figure 15. This has parameters chosen to lie in regime D. The value of $Q$ has been decreased relative to previous cases shown, to 105m, and this has reduced the magnitude of the negative gross moist stability at
RCE, hence reducing $y_{\mathrm{stab}}$ to be less than $L_{\mathrm{eq}}$. The unstable region is then sufficiently narrow that the equatorial wave response to the moist heating anomalies arising from the instability spread into $y > y_{\mathrm{stab}}$. The associated convergence drives moisture anomalies and eventually a self-consistent balance between the dynamical and moisture fields is reached, with similar length scales for each, with the zonal scale given by $c/\sqrt{\alpha\lambda}$ and the latitudinal scale by $L_{\mathrm{eq}}$.

In contrast to the more complicated cases shown in Figure 14, the structure here can be regarded as steadily propagating and
the decomposition expressed by (41) may be applied. Given that $\epsilon = 0$ the primary focus is on the $Q\nabla.\boldsymbol{u}$ term. It is assumed that the flow $\boldsymbol{u}$ is a quasi-steady response to the heating $F_h(q)$. Following the standard procedure for equatorial dynamics, as applied e.g. by Wu et al. (2001) the divergence may be written in terms of the heating as a combination of non-local Kelvin and Rossby responses, together with a local (convergence) that balances the heating as implied by WTG. The method for calculating the Kelvin and Rossby wave responses is set out in Appendix D.

An example of the contributions to the propagation speed of equatorial moist regions of the Rossby and Kelvin wave convergence response calculated using (41) with the convergence term split into Rossby and Kelvin components is shown in figure




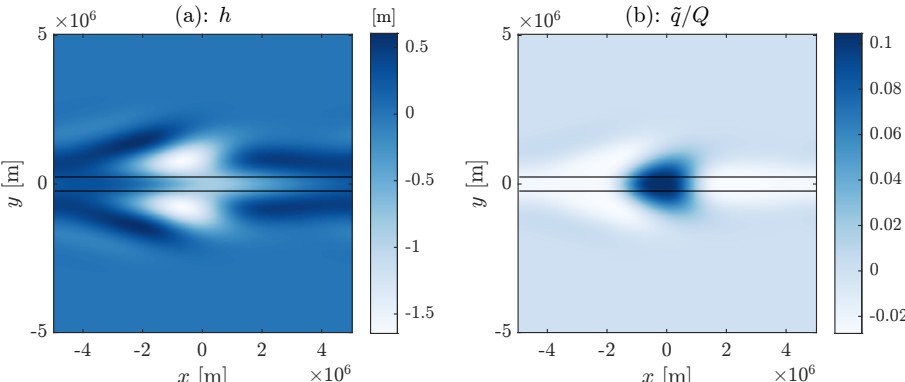

**Figure 15.** The thickness $h$ (a) and perturbation moisture $q/Q$ (b) fields for a simulation of the system on an equatorial beta-plane. This has $\alpha = \lambda = 10^{-6}\text{s}^{-1}$. The unstable region has been narrowed by decreasing the magnitude of the negative GMS at RCE. For this, we have set $Q = 105\text{m}$. The horizontal lines show $y = y_{\text{stab}}$. This disturbance propagates to the east at a speed of $0.15\text{ms}^{-1}$.

16 for both of the uniformly propagating examples. Row (a)–(b) is the case originally shown in Figure 14(f) and row (c)–(d) is the case in Figure 15. As expected the propagation direction does indeed correspond to the relative strength of the Kelvin wave response. Row (a)–(b) has a stronger equatorial Kelvin wave response, and hence the moist and dry regions travel westward,

whereas in row (c)–(d) the Rossby wave response is stronger and eastward propagation is observed. In both cases the contributions from each of the wave types is of a similar magnitude, and there is significant cancellation. This has two implications: the propagation speeds will tend to be smaller than expected from a dynamical scaling argument, and the propagation speed and direction is sensitive to the latitudinal structure of the moisture, since the details of that structure determine the relative strength of the Kelvin and Rossby wave responses.

The arguments in §5.1 suggest that the parameter $c\beta/\alpha^{3/2}\lambda^{1/2}$ is important, with eastward propagation preferred when this is large. This is consistent with the numerical experiments summarised in figure 17. There is a clear positive correlation, however as expected the variation in speed is not completely explained. For example, the system also shows a weak, complicated, dependence of propagation speed and direction on diffusivity, with increasing $\kappa$ both widening the boundaries between moist and dry regions and reducing $y_{\text{stab}}$.

Regions which propagate to the east tend to have a '<' shaped spatial structure. This structure is also noted in Sugiyama (2009b). This is due to the same asymmetry of the Kelvin and Rossby responses that leads to the eastward propagation. The Kelvin wave divergence to the east is necessarily localised near the equator, however the Rossby wave response can be at any latitudinal scale. Hence, to the east of the moist region there is reduced moisture convergence near the equator compared to further from the equator, however to the west of the moist region there is reduced moisture convergence at all latitudes. There is

therefore a tendency for off-equatorial ($y > L_{eq}$) regions of moisture to shift to the east. For a large moist region at the equator extending into this region, an eastward tilt will be generated away from the equator.



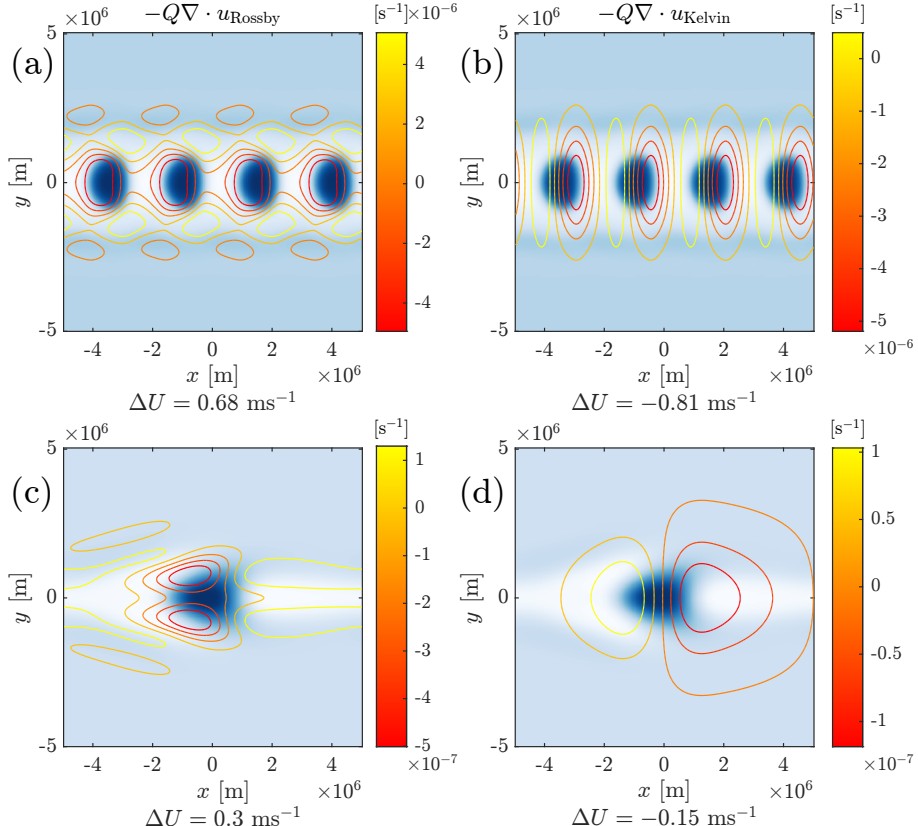

**Figure 16.** The spatial structure (contours) of the Rossby (left column) and Kelvin (right column) wave response to the moisture distribution (shading) of a uniformly propagating structure in a beta-plane simulation. The contribution of each wave to the propagation speed is labelled. Row (a)–(b) is the final state of the simulation in figure 14f, and row (c)–(d) is the simulation shown in figure 15. Note that the colourbar on each figure has a different scale.

### 5.2.2 $\epsilon > 0$

A set of examples with $\epsilon = 1$ is shown in figure 18. The change in spatial structure of the moisture field from $\epsilon = 0$ to $\epsilon = 1$ is largely similar to that seen previously in the constant-$f$ case (Figure 7 versus Figure 12). The associated convergence causes

moist regions to narrow, and rotational flows at larger $f$ lead to spirals in the moisture distribution.

Of the cases shown, (a), in regime C, and (d), in regime D, may be considered as leading to a coherent steadily propagating disturbance (although the off-equatorial round regions in (a) are still unsteady). Panel (b) corresponds to regime B and is closer to local quasi-isotropic aggregation, with features at different latitudes propagating at different speeds. This is highly unsteady, though the narrow propagation of the nonlinear advective convergence close to the equator is preferentially in the

zonal direction. Panel (c) resides near the boundary of regime B and C. The evolution is still very unsteady, with more evidence of rotational advection, particularly away from the equator where $f$ is non-zero.





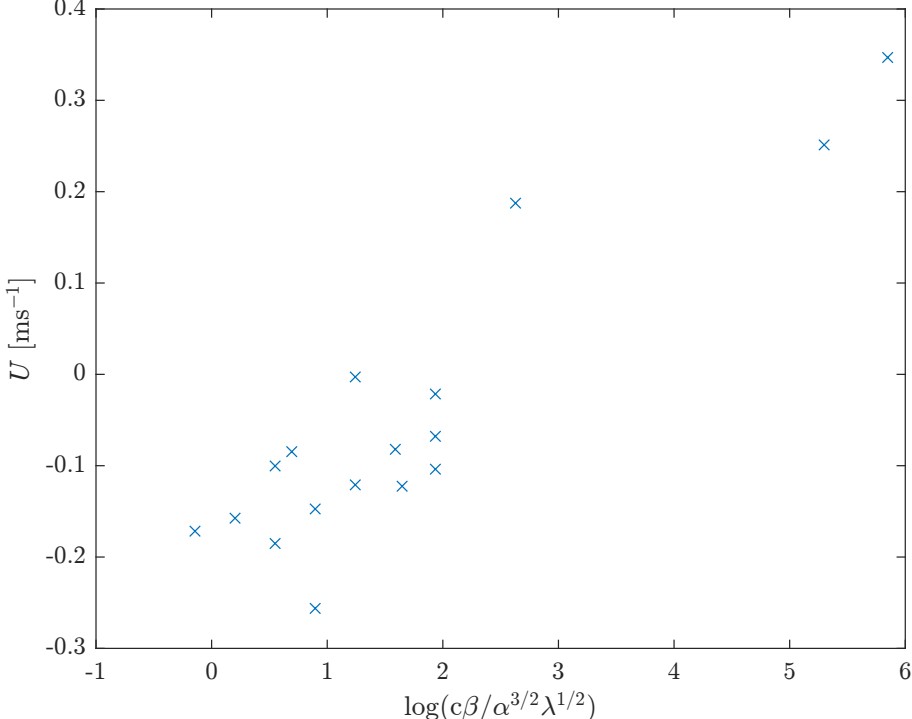

**Figure 17.** Propagation speed plotted against $c\beta/\alpha^{3/2}\lambda^{1/2}$ for a series of simulations with varying $H, g, \alpha, \lambda$ and $, \beta$.

For the steadily propagating cases the same technique as previously, based on (43) may be used to decompose the different contributions to propagation as shown in Figure 19. The nonlinear moisture advection term $\epsilon\nabla.(q\boldsymbol{u})$ term is now present. It is helpful to separate out the contribution from any zonal mean zonal flow. This is present because the segregation/aggregation process, with the form chosen for $F_h(q)$ and $F_q(q)$ tends to lead to a systematic zonal mean latitudinal structure in $q$, with greater $q$ at low latitudes. This leads to a corresponding structure in $h$ and hence in $u$. This zonal mean structure is present when $\epsilon = 0$, but then does not have any advective effect on $q$. The zonal mean $u$ usually took the form of two off-equatorial jets, centred where the latitudinal gradient in $h$ was largest. This zonal flow can potentially have a strong effect on propagation, but in the simulations shown here the generation of a strong zonal flow is avoided by choosing $\lambda$ sufficiently large that variations in $h$ are small. In practice this means $\lambda = \mathcal{O}(10^{-5}\mathrm{s}^{-1})$, well within the range discussed in the uniform rotation case.

For reference, the zonal mean $u$ for the simulation in figure 19 is shown in figure 20. In this simulation the global mean $u$ is $0.030\mathrm{ms}^{-1}$. The damping still allows a non-zero zonal mean $u$, however this is significantly reduced. The domain mean $u$ is constrained to be near zero so there is no advection by a uniform background flow.

Returning to the decomposition into different contributions to propagation shown in Figure 19, it may be seen that the linear divergence term (which has not been separated into Rossby and Kelvin contributions in this case, but the fact that the corresponding $\Delta U$ is positive implies that the Rossby contribution dominates), the term associated with advection by the zonal mean flow and the remaining nonlinear advection term are all comparable in magnitude.



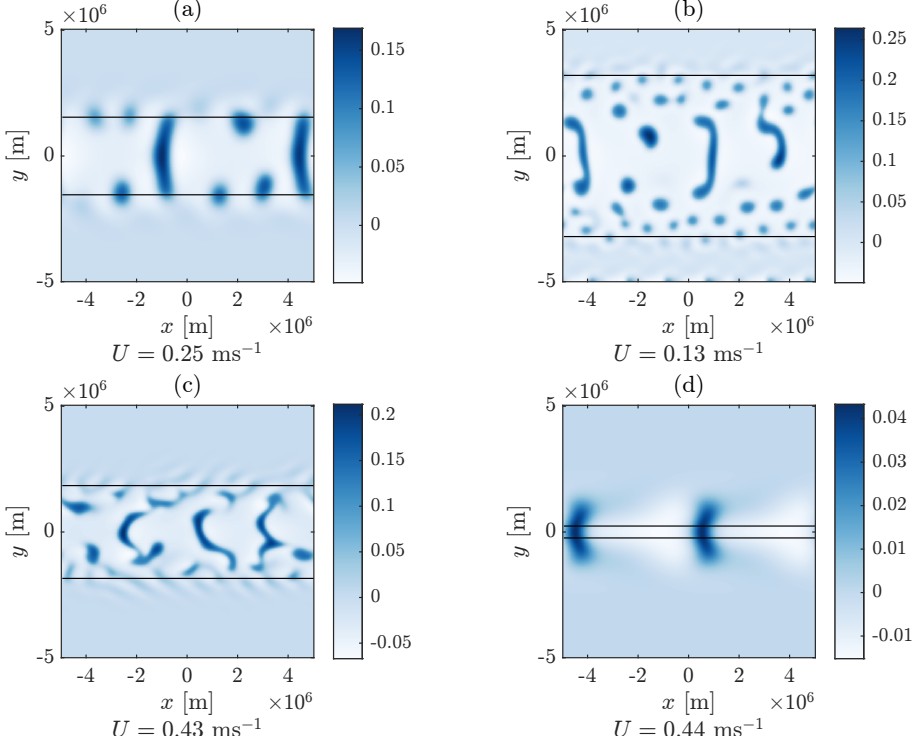

**Figure 18.** Snapshots of the normalised perturbation moisture distribution $q/Q$ after 400 days in simulations with nonlinear dynamics ($\epsilon = 1$) on an equatorial beta-plane with (a) $\kappa = 4 \times 10^5 \mathrm{m^2 s^{-1}}$, $\alpha = \lambda = 10^{-5} \mathrm{s^{-1}}$, the same as Figure 14f. Panel (b) has $\kappa = 10^5 \mathrm{m^2 s^{-1}}$ and $\alpha = \lambda = 10^{-5} \mathrm{s^{-1}}$, the same as figure 14a and panel (c) has $\kappa = 10^5 \mathrm{m^2 s^{-1}}$, $\alpha = 3 \times 10^{-6} \mathrm{s^{-1}}$ and $\lambda = 3 \times 10^{-5} \mathrm{s^{-1}}$, and does not correspond directly to a previous figure. Panel (d) illustrates the case with $L_{\mathrm{eq}} > y_{\mathrm{stab}}$, and has the same parameters as figure 15. These reside in regimes C, B, near the border of B and C, and D respectively.

## 6 Conclusions

In this study we have presented a single layer model for convective aggregation and its connection to large-scale dynamics.
The linearised shallow water equations, governing the dynamics, are augmented with a moisture equation. The moisture field affects the dynamics via a heating term, i.e. a forcing term in the thickness equation, and the dynamics affects the moisture through convergence alone ($\epsilon = 0$) or with the additional effect of horizontal advection ($\epsilon > 0$). The form of the moisture-dependent precipitation term and the moisture-dependent heating term are such that under the WTG approximation the spatially homogeneous state in which both precipitation and heating are zero (the 'RCE state') is unstable and the system is bistable
with moist and dry stable states (interpreted respectively as convecting and non-convecting). In this regime and with $\epsilon = 0$ the behaviour is described by a nonlinear reaction-diffusion equation for $q$, very similar to that presented by Craig and Mack (2013) and Windmiller and Craig (2019) (CMWC). As discussed by CMWC, the system exhibits a well-known spatial coarsening that may be interpreted as a representation of convective aggregation.



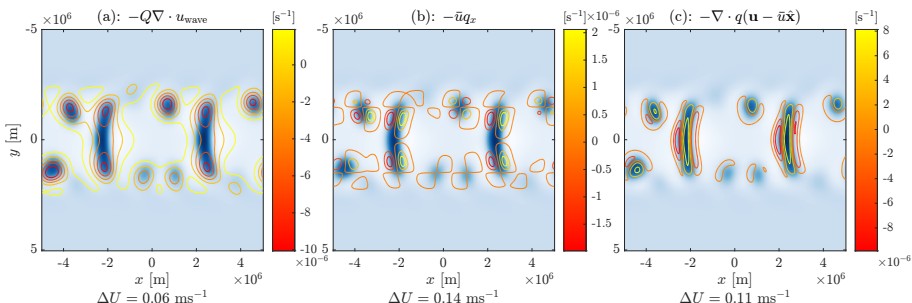

**Figure 19.** The spatial structure (contours) of the (a) linear wave, (b) advection by the zonal mean flow $\bar{u}$ and (c) nonlinear advection by the wave component contributions of the dynamical response to a moisture distribution (shading) of a propagating structure in a simulation on the beta-plane with nonlinear advection. The contribution of each wave to the propagation speed is labelled, and has been calculated using (41).

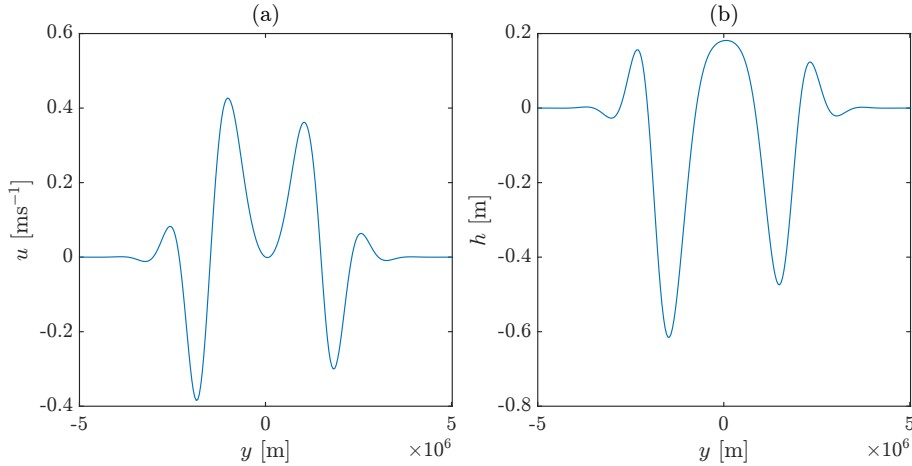

**Figure 20.** The zonal mean velocity (left) and height (right) fields for the state of the nonlinear beta-plane simulation shown in figure 19 .

The difference between our model system and that in the CMWC case is that in the latter the form of the reaction term is

determined solely through the dependence of precipitation on $q$. In our case it is determined in addition by the $q$-dependent heating, which couples to the moisture equation through the dynamics. This allows us to include more general dynamical effects beyond the WTG approximation, including the impact of thermal and frictional damping, nonlinear advection, and rotation, including the extension to the equatorial $\beta$-plane. Under WTG dynamics with $\epsilon = 0$ the system, as that studied by CMWC, coarsens to the largest available scale. This remains the case when nonlinear advection ($\epsilon > 0$) is included, however

the coarsening process is modified and the end state is different. Thermal and frictional damping and $f$- plane rotation in combination set a dynamical scale $L_{\mathrm{dyn}}$ which is an upper limit on the validity of WTG. The result is that coarsening proceeds only to this scale and then ceases. Depending on the relative values of thermal and mechanical damping and rotation the final



state may be essentially steady, or it may be unsteady, with, in some cases, a symmetry breaking leading to loss of spatial isotropy and to propagating structures. The nature of the linearly unstable modes provides some guidance to the type of the final state that is observed. If $f$ is large enough, with the critical value $f_{\text{stab}}$ depending on moisture diffusivity $\kappa$ as well as other parameters defining the system, then the RCE state is stable.

Many aspects of the behaviour of a $\beta$-plane may be interpreted in terms of the previously discussed $f$-plane behaviour. A latitude $y_{\text{stab}}$ may be defined by $f_{\text{stab}} = \beta y_{\text{stab}}$. Disturbances resulting from the instability of the RCE state are largely confined within the latitudinal band $(-y_{\text{stab}}, y_{\text{stab}})$, with some penetration outside of this band as a result of the non-locality of the dynamics, typically with a scale of $L_{\text{eq}}$ or the local value of $L_{\text{dyn}}$, whichever is the larger. When thermal and mechanical damping are strong enough, specifically when $c\beta\alpha^{-3/2}\lambda^{-1/2} < 1$, the evolution of the system is similar to that observed on the $f$-plane, with spatial modulation corresponding to the spatial variation of the value of $f$. The weak anistropy introduced by the $\beta$-effect leads to zonal propagation. The propagation is incoherent, with structures at different latitudes propagating at different speeds, however those on the equator propagate to the west. When $c\beta\alpha^{-3/2}\lambda^{-1/2} > 1$ the structures that form in the $q$-field are more strongly anistropic and coherently propagating and it is helpful to formulate a description of the dynamics in terms of equatorial Kelvin and Rossby waves. If the former dominate the dynamical response to the $q$-anomalies there is propagation to the west, if the latter there is propagation to the east. Nonlinear moisture advection enhances eastward propagation but the enhancement is not large.

The $\beta$-plane results that we present, particularly for the $c\beta\alpha^{-3/2}\lambda^{-1/2} < 1$ regime, have significant common ground with those presented by Sugiyama (2009b). The behaviour observed in both studies is similar, with small wavenumber, slowly propagating, moist regions forming at the equator. Indeed, as noted previously, one might argue that we are re-examining aspects of the models developed by Sugiyama (2009a, b). These models include WISHE, which we do not, and have a physically derived formulation of the forcing terms $F_q$ and $F_h$. We on the other hand emphasise the general implications of the bistability of the system, as implied by the form of $F_q(q) - (Q/H)F_h(q)$ and then choose very simple ad hoc forms of $F_q(q)$ and $F_h(q)$ to provide such bistability. We also deliberately trace the behaviour of the system through a sequence starting with WTG dynamics, and hence the reaction-diffusion behaviour discussed by CMWC, and finishing with the $\beta$-plane with thermal and mechanical damping and nonlinear moisture advection that was the focus of Sugiyama (2009b), thereby provide a new perspective on the latter.

One difference between the conclusions of this study and those of Sugiyama (2009b) is that the latter suggests that diffusion is an important contributor to the eastward propagation of moist regions in the nonlinear advection case. We, however, conclude that the propagation is due to a combination of nonlinear advection and the displaced convergence associated with the Rossby wave response. This difference in interpretation arises in part from the diagnostic approaches used to measure the impact of different terms of the moisture tendency. We assume a uniformly propagating disturbance and then calculate a contribution from each moisture tendency term to the speed of propagation using (19), taking into account the entire spatial distribution. As noted previously, according to that approach the net contribution of the diffusive term to propagation is zero. In contrast, whereas Sugiyama (2009b) compares the tendencies at $y = 0$. Only comparing at $y = 0$ will overestimate the contribution of the diffusive tendency to the overall propagation. In a "<" shaped aggregated region, the negative curvature at the equator to





the east increases the diffusive speed of the boundary whereas the opposite is true away from the equator where the sign of the curvature changes. The fact that according to (19) the net contribution of diffusion to propagation is identically zero does not,

of course, rule out the possibility that the form of the terms that do contribute to net propagation are affected by diffusivity. Certainly there is evidence of diffusivity dependence of propagation speed in the results presented, for example cases (a) and (f) in Figure 14, where increasing diffusivity increases westward propagation speed and cases (a) and (b) in Figure 18, where increasing diffusivity increases eastward propagation speed. A further difference from Sugiyama (2009b) is that we have not assumed $h$-dependence in $F_h$ and $F_q$. As has been noted previously, this follows other moisture-mode models including Sobel

and Maloney (2012, 2013); Adames and Kim (2016), but a preliminary assessment of the effect of including $h$-dependence is given in Appendix E and concludes that the formation, evolution and propagation of moist and dry regions, as described in §5 above remains broadly unchanged when this is included.

Recent review articles on convective aggregation, for example Wing et al. (2017) and Muller et al. (2022), include the CMWC reaction-diffusion model in their discussion of different mechanisms and models. The model we have presented above

generalises CMWC by linking the moisture and large-scale dynamical equations. In the categorisation given by Wing et al. (2017) our model fits the category of a long-wave radiation feedback with an additional advective process feedback included if $\epsilon > 0$. The CMWC model on the other hand, which relies on the form of the moisture dependence of precipitation, is categorised as a moisture feedback. Unlike the CMWC model, our model, by including dynamics, provides an upper limit on the scale of aggregation scale, which is finite and determined by thermal and momentum damping rates when $f = 0$ and reduces as $f$

increases. How this relates to evidence from GCM and Cloud Resolving Model simulations (Muller et al., 2022) boundary layer processes are important in determining the upper limit on aggregation scale remains to be determined.

Whilst most high-resolution three-dimensional modelling of convective aggregation has focused on the non-rotating case, there has been some recent investigation of the $f$-plane and $\beta$-plane cases, (Carstens and Wing, 2022, 2023). Some aspects of the behaviour reported in those papers is seen in our much simpler model. On the $f$-plane circular moist, i.e. convecting,

regions form and the scale of these regions decreases as $f$ increases, scaling as $1/f$ in both models at large rotation rates. However the physics of this behaviour is likely to be very different between the two models. In the CRM studies of (Carstens and Wing, 2022) the structure for larger values of $f$ is dominated by tropical cyclones. These have no clear analogue in our model which neglects advective nonlinearity in the momentum equation. (Carstens and Wing, 2022) identify an intermediate range of $f$ within which convective aggregation simply does not occur. One possibility is that this corresponds to our $f > f_{\text{stab}}$,

with formation of tropical cyclones being a distinct process occurs that occurs at larger $f$ value in the CRM, but which is simply absent in our model. On the $\beta$-plane (Carstens and Wing, 2023) identify the dominant structures that arise out of convective aggregation at low latitudes as convectively coupled Kelvin waves, whereas in our model the structures are clearly moisture-mode in character, with quasi-steady dynamical fields. In future work it would be interesting to investigate further whether there were are parameter regimes in which our model also shows moisture-modified Kelvin waves as the dominant

low-latitude structures.

Part of the motivation for the model presented in this paper is as a basis for understanding the MJO. The model considered by Sugiyama (2009b) was very similar to the $\beta$-plane form of our model and our conclusions are similar. There is organisation into





distinct moist and dry regions on the equator and these propagate in the zonal direction. As Sugiyama (2009b) we emphasise that these propagating disturbances are fundamentally nonlinear and their structure and propagation characteristics are not captured by a linear stability analysis. However, the direction of propagation can be westward or eastward and the speed of propagation is consistently much less than 1 ms$^{-1}$ when the observed MJO phase speed is around 5 ms$^{-1}$. Furthermore, even when the propagation is eastward propagating disturbances, the spatial structure of eastward propagating moist regions also necessarily forms a '<' shape, compared to the '>' structure of the observed MJO (Adames and Wallace, 2014).

It seems likely that, for a model of this form to produce more realistic MJO-like behaviour, further physics needs to be included. Recent simple one-dimensional models for the QBO (Sobel and Maloney, 2013; Adames and Kim, 2016), which are based on prescribed latitudinal structure of the flow variables, have included extra effects such as synoptic eddy drying, boundary-layer convergence of moisture (Adames and Wallace, 2014) and latitudinal gradients in background moisture (Adames and Kim, 2016) and have shown that these effects lead to enhanced eastward propagation. However the results we have presented in §5 of this paper demonstrate that propagation speed and direction are sensitive to latitudinal structure, so there are disadvantages to prescribing this *a priori*. We are currently including of some of these effects in the two-dimensional model described in this paper with the aim of capturing the behaviour of nonlinear MJO-like disturbances in a model where the two-dimensional structure emerges rather than being prescribed. Such a model could also potentially be used to study a broader class of moist tropical variability (Wang and Sobel, 2022).

*Video supplement.* Videos associated with the figures shown in this paper are available at
https://drive.google.com/drive/folders/1AG7Mjn0Nk4Gp_1wv0la9NutXAHdUFQuz?usp=sharing



## Appendix A: Numerical Details

To solve (1)–(3) numerically, we will use standard methods for solving the shallow water equations. The equations will be discretised on an Arakawa C-grid with the grid spacing chosen to be sufficiently small as to avoid numerical instability. In general we take a grid spacing of $d = 4 \times 10^4$ m and a time step of $112.5$ s. The system is then numerically stepped forward in time using a third order Adams-Bashforth scheme.

On the $f$-plane, we will take (doubly) periodic boundaries, whereas on the equatorial $\beta$-plane rigid north and south boundaries will be used along with periodic east and west boundaries. At rigid boundaries, a sponge layer will be included to avoid edge effects. This takes the form of a linear damping term included in the velocity and thickness fields. On the periodic channel we use on the $\beta$-plane this takes the form

$$\alpha_{\text{sponge}} = 1 \times 10^{-5} \left[ \exp\left( -\frac{70(L_y - 2y)}{L_y} \right) + \exp\left( \frac{70(L_y + 2y)}{L_y} \right) \right] \text{s}^{-1}. \tag{A1}$$

The magnitude decays over a scale of $1/70^{\text{th}}$ of the domain. Any numerical results do not depend on the magnitude or decay scale of this damping term.

## Appendix B: The $f$-plane stability boundary for small $\alpha$ and $\lambda$

This appendix provides more details of the analysis of the dispersion relation (18), repeated here as

$$\sigma^4 + [\lambda + 2\alpha + \mu_1 + \kappa k^2]\sigma^3$$
$$+ [(\mu_1 + \kappa k^2)(\lambda + 2\alpha) + f^2 + c^2 k^2 + \alpha^2 + 2\alpha\lambda]\sigma^2$$
$$+ [(\mu_1 + \kappa k^2)(f^2 + c^2 k^2 + \alpha^2 + 2\alpha\lambda) + c^2 k^2 \alpha + (f^2 + \alpha^2)\lambda - g\mu_2 Q k^2]\sigma$$
$$+ (\mu_1 + \kappa k^2)c^2 k^2 \alpha - g\mu_2 Q k^2 \alpha + (f^2 + \alpha^2)\lambda(\mu_1 + \kappa k^2) = 0. \tag{B1}$$

This is a quartic equation for $\sigma$ with coefficients that are either linear or quadratic in $k^2$. Some useful information is provided by considering the long-wavelength ($k \to 0$) and short-wavelength ($k \to \infty$) limits, both when $\kappa = 0$ and when $\kappa > 0$. With $\kappa = 0$ in the large-$k$ limit there are roots $\sigma \simeq \pm ick - \lambda - \mu_2 Q/H, -\alpha, \mu_2(Q/H) - \mu_1 = -M\mu_1$ representing, respectively, two gravity waves damped by moisture effects, a vorticity disturbance damped by friction and a moisture mode (stable or unstable according to whether $M$ is positive or negative). When $\kappa > 0$ all roots have negative real part as $k \to \infty$, i.e. the system is stabilised by diffusion of moisture. The corresponding small-$k$ limits (in which moisture diffusion does not play a role at leading order) are $\sigma \simeq -\lambda, \pm if - \alpha, -\mu_1$, implying stability at small $k$.

Given that there is stability at small $k$ and stability at large $k$ instability at some intermediate $k$ therefore requires a root of (B1) to cross the real axis. This can occur at $\sigma = 0$ or at $\sigma = is$ where $s \neq 0$ is real. $\sigma = 0$ implies that $(\mu_1 + \kappa k^2)c^2 k^2 \alpha - g\mu_2 Q k^2 \alpha + (f^2 + \alpha^2)\lambda(\mu_1 + \kappa k^2) = 0$ for some $k^2 > 0$ and it is the condition that this equation for $k^2$ has real and positive roots that leads to conditions (23) (for $f = 0$) and (24) (for $f \neq 0$).





The $\sigma = is$ case is more complicated. Substituting into (B1) and isolating real and imaginary parts gives two equations to be satisfied by $s^2$ and hence implies a constraint on the coefficients in (B1) and an quartic equation for $k^2$, which must have real and positive roots. Further analytic progress is challenging.

The case $f = 0$ is relatively straightforward. (B1) has a factor $(\sigma + \alpha)$ and this simplifies the algebra and allows the $\sigma = is$ case to be excluded. Furthermore the fact that $\sigma = 0$ occurs at two values of $k^2$, at most, together with stability at small $k$ and large $k$, implies that $\sigma$ complex with $\Re(\sigma) > 0$ is not possible, and that there is a single unstable mode with real $\sigma$.

For the $f \neq 0$ case numerical investigation shows that when $\lambda$ is small the real $\sigma$-axis is crossed only at $\sigma = 0$ and that as $\lambda$ increases the transition to complex $\sigma$ with positive real part occurs when there is a double root at $\sigma = 0$. This occurs when both the constant term and the coefficient of the linear term in (B1) vanish simultaneously. Eliminating $k^2$ gives a constraint on $\lambda$ and $\alpha$ (for given values of other parameters such as $f$ and $\kappa$) which defines the boundary between regions I and IIa in Figure 6. For illustration, the resulting expression when $\kappa = 0$ is

$$\lambda = \lambda_*(\alpha) = \frac{(\alpha^2 + f^2)(Q\mu_2/H - \mu_1)\mu_1\alpha}{(Q\mu_2/H - \mu_1)(f^2\mu_1 - f^2\alpha - \alpha^2\mu_1 - \alpha^3) - \alpha\mu_1(f^2 + \alpha^2)}. \tag{B2}$$

In the small-$\alpha$ limit it corresponds to $\lambda \simeq \alpha$. Since this expression for $\lambda_*(\alpha)$ ignores diffusivity it will provide an accurate description of the boundary between regions I and IIa only when that is approximately independent of diffusivity, which, from inspection of Figure 6 appears to be the case when $\lambda$ and $\alpha$ are small.

The boundary between regions IIb and III corresponds to a double root of (B1) of the form $\sigma = is$, $s \neq 0$. However some information about the small-$\alpha$ part of this boundary may be obtained by considering the special case $\alpha = 0$. Then there is a root that is $\mathcal{O}(k^2)$ for small $k$, with real part that has the same sign as $\lambda\mu_1(\mu_2(Q/H) - \mu_1) - f^2(\lambda + \mu_2(Q/H))$. It follows that for $f^2 < (\mu_2(Q/H) - \mu_1)\mu_1$ there is instability at small $k$ for large $\lambda$, therefore stabilization requires non-zero $\alpha$. On the other hand for $f^2 > (\mu_2(Q/H) - \mu_1)\mu_1$ then there is stability at large $\lambda$ independent of the value of $\alpha$. This explains the different forms of the boundary between regions IIb and III seen in Figure 6 for $f = 0$ and $f = 10^{-5}\mathrm{s}^{-1}$ on the one hand and $f = 3 \times 10^{-5}\mathrm{s}^{-1}$ and $f = 6 \times 10^{-5}\mathrm{s}^{-1}$ on the other.

Finally we consider the regime where $\alpha$ and $\lambda$ are both small. This is potentially described by setting $\alpha = \lambda = 0$ in (18), leading to

$$\sigma^3 + \left(\kappa k^2 + \mu_1\right)\sigma^2 + \left(f^2 + c^2 k^2\right)\sigma + f^2\mu_1 + c^2\kappa k^4 + f^2\kappa k^2 + c_m^2 k^2 \mu_1 = 0, \tag{B3}$$

with the moist gravity wave speed $c_m^2 = c^2 M < 0$ for the potentially-unstable case. Denote the roots of this polynomial as $\sigma_1$, $\sigma_2$ and $\sigma_3$.

First assume that there is a purely imaginary root $\sigma = is$, with $s$ real. Substituting into (B3) and separating the real and imaginary parts gives

$$s^3 = (f^2 + c^2 k^2)s, \tag{B4}$$

$$(\mu_1 + \kappa k^2)s^2 = \mu_1 f^2 + \mu_1 c_m^2 k^2 + f^2\kappa k^2 + c^2\kappa k^4. \tag{B5}$$





Eliminating $s$ gives that

$$(\mu_1 + \kappa k^2)(f^2 + c^2 k^2) = \mu_1 f^2 + \mu_1 c_m^2 k^2 + f^2 \kappa k^2 + c^2 \kappa k^4. \tag{B6}$$

For $k > 0$, this is true only in the dry case $c_m^2 = c^2$. Since we are interested in $c_m^2 < 0 < c^2$, purely imaginary roots are not possible. Therefore any change in stability happens at $\sigma = 0$.

For small $k$, (B3) has roots $\sigma = -\mu_1$ and $\sigma = \pm i f - k^2 (2g\mu_2 \mu_1 f^2 Q \pm 2if(f^2 c^2 + c_m^2 \mu_1^2))/4(f^4 + f^2 \mu_1^2)$, both of which have negative real part. For instability, a change of stability to occur at some $k > 0$. This requires the constant coefficient of (B3) to change sign, i.e.

$$\mu_1 f^2 + \mu_1 c_m^2 k^2 + f^2 \kappa k^2 + c^2 \kappa k^4 = 0. \tag{B7}$$

If the system is to be stable, this must have no real roots $k^2 > 0$. Nondimensionalise with $\hat{f} = f\sqrt{\kappa/\mu_1}/c$ and $\hat{k} = k/\sqrt{\kappa/\mu_1}$, for

$$\hat{f}^2 + M\hat{k}^2 + \hat{f}^2\hat{k}^2 + \hat{k}^4 = 0. \tag{B8}$$

The transition from two roots to none happens when the discriminant of this expression (as a quadratic in $k$) vanishes, or

$$(M + \hat{f})^2 - 4\hat{f}^2 = 0 \tag{B9}$$

This has four roots,

$$\hat{f} = \pm(1 - \sqrt{1 - M}) \tag{B10}$$

corresponding to the change in global stability, and

$$\hat{f} = \pm(1 + \sqrt{1 - M}), \tag{B11}$$

which are a spurious solution arising from roots with $k^2 < 0$. (When $\hat{f}^2 > 1 - M$, the left-hand side of (B8) is strictly positive for $k^2 > 0$).

With zero damping, and assuming positive we therefore expect instability if and only if

$$f < f_{\text{stab}} = \frac{c}{\sqrt{\kappa/\mu_1}}(\sqrt{1 - M} - 1), \tag{B12}$$

where $f_{\text{stab}}$ is defined by this equality. Since only one mode may be unstable in this case the unstable mode must have a purely real growth rate and therefore cannot propagate.

When $f_{\text{stab}}$ is defined by (24), we can consider the limit as $\alpha \to 0$ and $\lambda \to 0$. This limit is not well defined, it depends on the value of $\alpha/\lambda$.

The condition $f = f_{\text{stab}}(\lambda, \alpha)$ defines a function on the $(\lambda, \alpha)$ plane, as the maximum $f$ for given $\alpha$ and $\lambda$ (with dependence on other parameters such as $\kappa$ suppressed) for which the moist shallow water equations (1), (2) and (3) permit unstable modes.





Contours of this function are shown in Figure 6, as the bounding curves for Regime III. This surface is expected to be continuous at all points but the origin, $\alpha = \lambda = 0$.

There is a value $f_I$ such that the region of $(\lambda, \alpha)$ space defined by $f_{\text{stab}}(\lambda, \alpha) > f_I$ contains no parameters corresponding to

Regime II, i.e. the unstable roots of (18) are real for all $k$. Within this region, the stability boundary is given by (24), rewritten here as a curve in the $(\lambda, \alpha)$ plane,

$$\lambda = \frac{c^2 \mu_1 \alpha}{\kappa(\alpha^2 + f^2)}(\sqrt{\mu_2 Q / \mu_1 H} - 1)^2. \tag{B13}$$

(Note that as $f$ is made arbitrarily large this curve moves closer to the $\alpha$-axis, but does not meet it, so there is always some range of $\alpha$ such that there is instability for $\lambda = 0$.) Within this region, the transition with varying $k$ of a single real mode

from unstable to stable corresponds to the change in sign of the constant term of (18). When $\alpha = \lambda = 0$, this constant term is identically zero. We therefore have a degenerate case where the previously unstable mode is zero for all $k$. The limit as $\alpha \to 0$ and $\lambda \to 0$ of $f_{\text{stab}}$ in this region is therefore not expected to be well defined.

In the other section of the $(\alpha, \lambda)$ plane, where $f_{\text{stab}} < f_I$, instability both near the origin and near $f = f_{\text{stab}}$ relies on a combination of two complex roots of (18). If $\alpha \to 0$ and $\lambda \to 0$ only one of these roots can become identically zero so we

expect the other to become the unstable mode, and hence the limit of $f_{\text{stab}}$ should be well defined.

Since $f_{\text{stab}}$ is continuous, apart from at the origin, we also expect the limit along the boundary between each of the cases discussed above, $f_{\text{stab}} = f_I$ to give the correct value of $f_{\text{stab}}(0,0)$. This is a curve of constant $f_{\text{stab}}$ and therefore

$$f_I = f_{\text{stab}}(0,0) = \left( \frac{c^2 (\sqrt{\mu_2 Q / \mu_1 H} - 1)^2}{\kappa / \mu_1} \right)^{1/2}. \tag{B14}$$

The curve $f_{\text{stab}}(\lambda, \alpha) = f_{\text{stab}}(0,0)$ has the expected form, $\alpha = \lambda$, when $\alpha \ll f$.

**Appendix C: The Equatorial Wave Response to a Moist Heating**

The full derivation of linear equatorial wave response to a heating with general thermal damping $\lambda$ and friction $\alpha$ is derived by Wu et al. (2001). The setup and relevant results are repeated here for reference.

We look for equatorially trapped steady state solutions to the dynamical equations (1) and (3), under the long wave approximation. These satisfy

$$\alpha u - \beta y v = -g h_x, \tag{C1}$$

$$\beta y u = -g h_y, \tag{C2}$$

$$\lambda h + H \nabla \cdot \boldsymbol{u} = F, \tag{C3}$$



with $u$, $v$ and $h \to 0$ as $y \to \pm\infty$. For simplicity we will take a domain of effectively infinite zonal length, i.e. with length much greater than the extent of the equatorial wave response. We may also assume that $F(x,y)$ has mean zero, as a non-zero spatial mean part will be balanced by a change in the mean $h$ and not contribute to the divergence or equatorial wave response.

We must first decompose the heating in terms of parabolic cylinder functions $\tilde{D}_n$ as

$$F = \sum_{n \geq 0} F_n \tilde{D}_n \left( \frac{y}{L_{eq}} \right), \tag{C4}$$

with $L_{eq} = \sqrt{c/\beta} \sqrt[4]{\alpha/\lambda}$.

The relevant parts of the solutions are then the equatorial Kelvin wave, which has

$$h_0 = -\frac{1}{2} \int_{-\infty}^{x} \sqrt{\frac{\alpha}{\lambda}} c^{-1} \exp\left( -\frac{\sqrt{\alpha\lambda}(x-u)}{c} \right) F_0(u) \, du \, \tilde{D}_0 \left( \frac{y}{L_{eq}} \right). \tag{C5}$$

and the equatorial Rossby waves, for $n = 1, 2, 3, \dots$,

$$h_n = -\frac{1}{2} \int_{x}^{\infty} \sqrt{\frac{\alpha}{\lambda}} c^{-1} \exp\left( \frac{(2n+1)\sqrt{\alpha\lambda}(x-u)}{c} \right) (F_{n-1}(u) + 2n F_{n+1}(u)) \, du \left( \tilde{D}_{n+1} \left( \frac{y}{L_{eq}} \right) + 2(n+1) \tilde{D}_{n-1} \left( \frac{y}{L_{eq}} \right) \right). \tag{C6}$$

The divergence is then calculated using equation (C3). This gives

$$D[F] = F/H - \lambda \sum_{n \geq 0} h_n / H \tag{C7}$$

For our purposes it is worth noting that for a heating, $F < 0$, the equatorial wave response has $h < 0$, and hence the associated divergence is positive. This is the opposite sign to the WTG divergence associated with a heating.

## Appendix D: Decomposition of the Divergence into Equatorial Wave Components

For diagnostics of numerical simulations we will want to calculate the expressions in Appendix C numerically, including separating the divergence into weak temperature gradient and equatorial Kelvin and Rossby components. The expression for the divergence (C7) can simply be split into parts

$$D[F] = D_{WTG}[F] + D_K[F] + D_R[F], \tag{D1}$$

where

$$D_{WTG} = \frac{1}{H} (F), \tag{D2}$$

$$D_K = -\lambda h_0[F]/H, \tag{D3}$$



and

$$D_R = -\lambda \sum_{n \geq 1} h_n[F]/H. \tag{D4}$$

The first two of these are simple to calculate numerically, using the expressions in Appendix C with $F = F_h - \bar{F}_h$. The Rossby wave component will then be calculated as the residual

$$D_R = D - D_{WTG} - D_K. \tag{D5}$$

This will introduce some error as the numerical fields are not exactly quasi-steady, however it also sidesteps a few issues with calculating the Rossby wave response. The first of these is that simple theoretical expressions for the Rossby wave response

to a given heating assume the long-wave approximation, whereas this assumption is not needed for the Kelvin wave and may not be valid for certain simulations. The second is that the Rossby wave response requires forcing due to all terms in (C4). For a finite discretised domain these will be both poorly resolved and cut off by boundaries. This leads to errors in the projection, since the modes are no longer orthogonal.

## Appendix E: The dependence of precipitation parameterisation on temperature

The impact of including a temperature dependence in the precipitation, as done by Sugiyama (2009a, b), will be briefly discussed in this section. The coupling terms between the moisture and thickness equations in these papers are assumed to be functions of the precipitation $P$, which is assumed to depend only on the moist static energy, $c_p T + L_v q$ in standard units, and the nonlinear combination of $P$ with a surface flux term. The terms thus take the form $F_h(q - h/\gamma, h)$ and $F_q(q - h/\gamma, q)$, where $\gamma$ is proportional to $L_v/c_p$. This form is motivated in Sugiyama (2009a, b) as a Betts-Miller type representation, but

the alternative interpretation as a representation of the dependence of precipitation on free-tropospheric humidity, which was emerging at the time of those papers, is noted and that interpretation would not require the $h$-dependence.

The forcing in this paper has been defined differently, we have taken a simple linear form for all forcing terms except for a nonlinear $F_h(q)$, which provides an effective limit on the magnitude of the moisture variable. Two initial questions which arise from the temperature dependence of $F_h$ and $F_q$ are: how does the behaviour change if the nonlinear term limits $q - h/\gamma$ rather

than $q$, and, what is the effect of a linear $h$ term in the moisture equation?

In first case we can get some insight by redefining the moisture variable. In this case the moisture terms become $F_h(q - h/\gamma)$ and $F_q(q - h/\gamma)$. Subtracting $1/\gamma \times (2)$ from (3) gives

$$(q - h/\gamma)_t + (Q - H/\gamma)\nabla \cdot \boldsymbol{u} = F_q(q - h/\gamma) - \gamma^{-1}F_q(q - h/\gamma) + \lambda h/\gamma + \kappa\nabla^2 q. \tag{E1}$$

Defining a new variable $q' = q - h/\gamma$, along with corresponding $Q' = Q - H/\gamma$ and $F_{q'} = F_q - F_h/\gamma$ we can write this as

$$q'_t + Q'\nabla \cdot \boldsymbol{u} = F_{q'}(q') + \kappa\nabla^2 q' + (\lambda + \nabla^2)h/\gamma. \tag{E2}$$

We have returned to the original form of the equations to be studied with a different interpretation of the moisture variable and some terms due to $h$ on the right hand side of the moisture equation. Physically, it is unclear whether the diffusive forcing





should act on moisture $q$ or on moist static energy $q'$, so the $\nabla^2 h$ term may be unnecessary. If this term is neglected we end up in the situation described by the second question above. The distinction between the two setups can therefore also be interpreted as whether the diffusivity is believed to act upon the moisture or moist static energy.

The situation described by Sugiyama (2009b) is one step further, with $F_h = F_h(q, h)$ and $F_q = F_q(q, h)$. We will now investigate this case, but for simplicity we will continue to take $\epsilon = 0$. In the strict WTG limit, $h = h(t)$, using (2) to eliminate the divergence from (3) gives

$$q_t = F_q(q, h) - \frac{Q}{H}(F_h(q, h) - \overline{F_h(q, h)}) + \kappa \nabla^2 q. \tag{E3}$$

This is still a reaction-diffusion equation for $q$, however now the stable states depend on $h$, $q_\pm = q_\pm(h)$. These vary slowly as $h$ varies slowly with time.

When $L_{\mathrm{dyn}}$ is larger than the domain size, the arguments in section 2 hold and aggregation proceeds as expected. Otherwise, WTG cannot be valid across the entire domain. Moist regions, with their associated heating, will have decreased $h$ and the opposite will apply to dry regions. On a $\beta$-plane the region of reduced $h$ associated with a heating at the equator will extend zonally with the equatorial Rossby and Kelvin wave responses. It is therefore expected that there will be local variation of the stable values of $q$. Of particular interest will be whether the adjustment to the fixed points due to the equatorial wave response to heating will facilitate moistening to the east or west of an equatorial moist region.

We know that the fixed points will depend on $h$, but what do we expect this variation to look like? We will discuss two informative special cases. For the first, we assume that the moisture and thickness coupling terms depend only on the moist static energy, $F_q(q - h/\gamma)$ and $F_h(q - h/\gamma)$. Note that since the spatial mean of $F_h$ must be subtracted when applying the WTG approximation, an additional linear relaxation term in the thickness equation will not affect the WTG moisture equation. In this case the stable fixed points are related to the stable fixed points $q_{\pm 0}$ when $h = 0$ by

$$q_\pm(h) = q_{\pm 0} + h/\gamma. \tag{E4}$$

Thus, the stable values of the moisture increase with $h$.

For the second case we will assume that the effect of $h$ in the moisture equation is weak and therefore may be represented by a linear term. Taking inspiration from te previous case but Taylor expanding the moisture about RCE, the moisture equation (3) is

$$q_t + Q\nabla \cdot \boldsymbol{u} = F_q(q) - F_q'(Q)h/\gamma + \kappa \nabla^2 q. \tag{E5}$$

Note that $F_q' < 0$. Now, rather than the condition that at fixed points the reaction function $G_{hq} = 0$, we need that $G_{hq} = F_q'(0)h/\gamma$. This corresponds to a translation of the curve shown in figure 1 upwards with increasing $h$. Therefore as $h$ increases the stable moisture values will correspondingly increase.

These two situations agree on the expected effect of $h$ dependent forcing in the height equation on the moisture distribution, however the impact on the propagation speed of aggregated regions is unclear. Since the deviation from isotropy in the beta-plane case is due to the equatorial wave response to the heating we expect significant cancellation between the equatorial



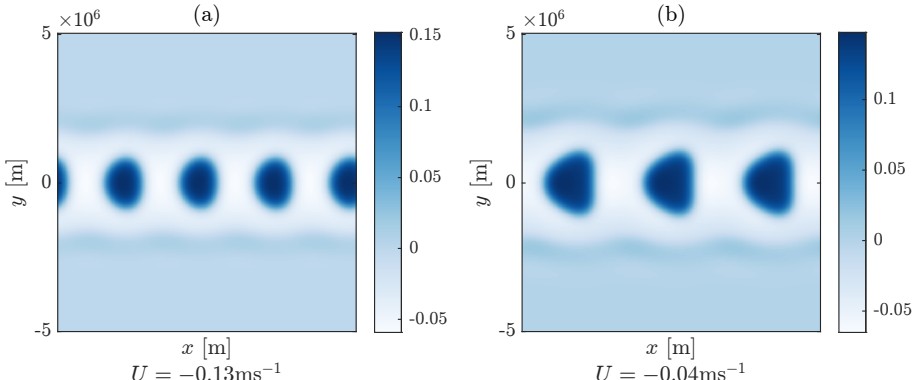

$U = -0.13 \mathrm{ms}^{-1}$ $\qquad\qquad\qquad\qquad$ $U = -0.04 \mathrm{ms}^{-1}$

**Figure E1.** The late time behaviour of a beta-plane simulation with parameters as in panel (f) of figure 14, but with $h$ dependent heating. Panel (a) has $h$ dependence in both $F_h(q - h/\gamma)$ and $F_q(q - h/\gamma)$. Panel (b) has $F_h(q)$ and $F_q(q - h/\gamma)$. The 'specific heat' parameter $\gamma = 22.2$ has been chosen to match the combination of geopotential and temperature in Sugiyama (2009b).

1260  Rossby and Kelvin wave components, as seen in the divergence response discussed in section 5. We will investigate this numerically.

Examples of numerical simulations from both cases is shown in figure E1. The first case, with both coupling terms functions of only moist static energy leads to only very small changes from the corresponding case in figure 14—the spatial distribution and wavenumber are similar, and the speed has increased by $0.1 \mathrm{ms}^{-1}$. The 'specific heat' parameter $\gamma = 22.2$ has been chosen

1265  to match the combination of geopotential and temperature in Sugiyama (2009b). The fact that the effect is small is consistent with the large value of $\gamma$. The height dependence in the second case has had a larger effect, with the wavenumber reduced from 4 to 3, and the speed reduced by a factor of 3.

*Author contributions.* Both authors contributed equally to this work.

*Competing interests.* The authors have no competing interests to declare.

1270 *Acknowledgements.* MD is grateful for funding from the UK EPSRC via a PhD studentship.



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
