# Peer review of "A simple model linking radiative-convective instability, convective aggregation and large-scale dynamics"

_EGUsphere, 2024_

## Referee Comment (RC2)

Review of Manuscript "A simple dynamical model linking radiative-convective instability, convective aggregation and large-scale dynamics" by Davison and Haynes

This is perhaps the most thorough study on convective aggregation and its potential effect on large-scale dynamics. It goes through meticulous discussions, both analytically and numerically, of the role of rotation ($f$ and $\beta$- effects), nonlinear advection, thermal and dynamical damping and mean flow in convective aggregation and in the consequential large-scale zonal propagation. Its systematic treatment of convective aggregation in a variety of parameter space is a huge step forward from previous studies on this subject. This is a massive study with many mathematical details. It's impractical to discuss all these details in this review, although the devil is in them. Most of my comments are on the main objective of this study, which is to link convective aggregation to large-scale dynamics. My general sense is that the discussion on aggregation is fine, but its implication to the MJO is a stretch.

General comments:

1.  There are so many simplifications and assumptions in the formulation of the theory that it is difficult for readers to see which ones are for mathematical convenience to get an analytical solution but bear minimal physical consequences, and which ones are key to the conclusion. It might be helpful to have a table that list all simplifications and assumptions and marks those that is at the center of this theory (e.g., must have to make the conclusion valid).
2.  Sections 3 and 4 are unnecessarily long. The failure of aggregation on an $f$-plane has been documented before (e.g., Carstens and Wing 2022). Even though more detailed discussions on this subject add intellectual values and it might be nice to compare results on a $\beta$-plane to those on an $f$-plane, Sections 3 and 4 do not directly contribute to the main conclusion of this study, which can be made solely based on the $\beta$-plane discussions. Suggest substantially shorten this part and retain only the materials that are directly relevant to compare with those on a $\beta$-plane. This would make the manuscript more focused on tropical dynamics and easier to follow. The removed materials on an $f$-plan can be published elsewhere.
3.  I found the interpretation of large-scale zonal propagation in terms of its dependence on latitudes very interesting. Based on the authors' interpretation, when Rossby (Kelvin) wave response is stronger, the propagation tends to be eastward (westward). This is counter-intuitive but not impossible. There are several issues that the authors may want to consider:
a)  The Rossby wave may lead to eastward propagation without any convective aggregation (Hayashi and Itoh 2017; Rostami and Zeitlin 2019; Yano and Tribbia 2017).
b)  The westward propagation due to stronger Kelvin wave responses in the context of convective aggregation should be discuss in the context of its natural (dynamical) eastward propagation. In other words, what is the net balance between the dynamical eastward propagation of the Kevin wave and its westward propagation due to convective aggregation?
c)  The "<" shape structure is in the opposite direction of the "swallowtail" pattern observed for the MJO (Zhang and Ling 2012). How do the two reconcile with each other if the authors think their results are relevant to the MJO?
d)  With a strength of convective heating associated with moisture anomalies, its Rossby and Kelvin wave responses are pretty much fixed based on the Gill model. Under what

circumstances the relative strengths of Rossby and Kelvin wave responses would vary? By the latitude location of convective heating?

4.  The time it takes for the aggregation to reach a large-scale state (400 days) is much longer than the initiation of the MJO or any tropical large-scale disturbances. How should this inconsistency be reconciled?

5.  In the conclusion section (6), the authors compared their results with those from previous studies of the MJO, all of them relying on the moisture-mode doctrine. It might be further enlightening if the authors walk out of the moisture-mode and aggregation camps and discuss what their results add to other simpler theories. For example, as mentioned above, several studies (Hayashi and Itoh 2017; Rostami and Zeitlin 2019; Yano and Tribbia 2017) proposed that the dry Rossby wave alone may propagate eastward to provide a propeller effect on the MJO. Kim and Zhang (2021) suggested that dry Kelvin waves alone can propagate eastward slowly at the MJO speed in the presence of dynamical damping. They reproduced the observed "swallowtail" pattern without Rossby waves. As the authors know very well, Adames and Kim (2016) provided a perhaps the most comprehensive MJO theory based on the moisture-mode thinking. All these studies reproduced fundamental MJO properties to various degrees without convective aggregation. The big question for this study under review is: What additional physical insights to MJO dynamics can be added to the previous studies or what fundamental MJO properties missing from the previous studies can be produced by including convective aggregation? Especially, the authors mentioned that more physics are needed to make the results more realistic. Is that so? Majda and Stechmann (2009) and Kim and Zhang (2021) included far less physics (assumptions) but produced no less fundamental properties of the MJO than other MJO theories. Sometimes less is better. I challenge the authors of this study to pursue the beauty of simplicity. If convective aggregation plays an important role in the MJO, there must be a way to demonstrate this in a simple, straightforward way without invoking numerous assumptions and massive mathematical procedures.

Detailed comments:

1.  Near line 20: "The key overall properties of the propagating disturbances, the spatial scale and the phase speed, depend on nonlinearity in the coupling between moisture and dynamics and any linear theory for such disturbances therefore has limited usefulness." I beg to differ. See major comment 4 regarding simple linear theories.

2.  Line 28: "Much theoretical and modelling work over the past few decades has focused on the coupling between dynamics and moisture in the tropical atmosphere, which it is clear must be taken into account at leading-order to explain many tropical phenomena." Ditto. Look around outside the moisture-mode camp and you would find theories with equal or greater success in comparison to the moisture-mode ones. This would help to put this theory in a broader context.

3.  The origin of the moisture equation (3) needs to be scrutinized. In the text, it is "assumed", which is not good enough. In most of the moisture-mode studies, such a moisture equation is thrown to the well-established shallow-water equation as a second thought. This has been one of the most severe inconsistencies in the moisture-mode theories that very few have realized. Neelin and Zeng (2000) derived a set of horizontal structure equations for wind, temperature

and humidity (their Eqs 5.1 – 5.4) after applied variable separation and vertical base functions. Can the moisture equation (3) in this study under review be vigorously derived from Neelin and Zeng's humidity equation (5.4)? If not, we have a problem.

4. What is the physical reality of $\frac{d\bar{h}}{dt}$ (Eq 5)? Global mean temperature fluctuation? Is there an example for this? Please forgive my ignorance if this is a common knowledge.

5. Please provide justifications for the values of $\alpha$, $\lambda$, $\kappa$ chosen in Table 1 or point out where in the text such justifications are given if I missed them.

6. The domain integrations (Eqs 42 – 43) need explanations. If it covers the entire longitudinal circle, it mixes the convective part (Indian Ocean and Western Pacific) where zonal speeds tend to be lower than the dry part of the tropics. Otherwise, do they distinguish the faster and slow speeds when the domain covers the moist and dry zones separately?

7. The linear friction coefficient $\alpha$ plays a central role on a $\beta$-plan but not on an $f$-plan. The parameter $c\beta/\alpha^{3/2}\lambda^{\frac{1}{2}}$ suggests that $\alpha$ must not be zero on a $\beta$-plan. All $\alpha = 0$ cases are on an $f$-plan. Linear friction plays critical role in some theories (e.g., Adames and Kim 2016; Wang et al. 2016; Kim and Zhang 2021), but not in others (e.g., Majda and Stechmann 2009; Hayashi and Itoh 2017; Yano and Tribbia 2017; Rostami and Zeitlin 2019; Emanuel 2019). Can the author explain why linear friction must exist in this theory and what exactly it does to make the theory work on a $\beta$-plan? Eqs 36 and 37 clearly demonstrate that $\alpha$ and $f$ (or $\beta y$) play equivalent roles and they cannot be zero at the same time. This deserves a physical explanation: Why must friction exist at the equator but not away from the equator? Go back to the starting point (Eqs 14 – 17), there must be a solution with $\alpha = 0$. What is that solution on a $\beta$-plan?

8. Figs. 14, 15, 16, and 19: Does it take the equal length of integration to reach the states shown in the panels? What is the integration length? 400 days as in Fig. 18? It should be mentioned in the captions.

9. In all the parameters used, which one control the speed at which aggregated perturbations grow? Fig. 9 suggests it's $\kappa$ when $\alpha = \lambda = f = 0$. How about on a $\beta$-plan when $\alpha$ and $\lambda$ are not zero?

References

Carstens and Wing, 2022: A Spectrum of Convective Self-Aggregation Based on Background Rotation, Journal of Advances in Modeling Earth Systems, 14, 1–14.

Emanuel, K. A. (1991). A scheme for representing cumulus convection in large-scale models. Journal of the Atmospheric Sciences, 48, 2313–2329.

Hayashi, Michiya, and Hisanori Itoh, 2017: A new mechanism of the slow eastward propagation of unstable disturbances with convection in the tropics: Implications for the MJO. Journal of the Atmospheric Sciences 74, no. 11, 3749-3769.

Kim, J.-E., and C. Zhang, 2021: Core dynamics of the MJO. *Journal of the Atmospheric Sciences* 78, no. 1, 229-248.

Majda, A. J. and S. N. Stechmann, 2009: The skeleton of tropical intraseasonal oscillations. Proc. Natl. Acad. Sci. USA, 106, 8417–8422.

Neelin, J. D. and Zeng, N., 2000: A Quasi-Equilibrium Tropical Circulation Model - Formulation, Journal of the Atmospheric Sciences, 57, 1741–1766,

Rostami, M., and V. Zeitlin, 2019: Eastward-moving convectionenhanced modons in shallow water in the equatorial tangent plane. Phys. Fluids, 31, 021701

Wang, B., Liu, F., & Chen, G. (2016). A trio-interaction theory for Madden-Julian oscillation. Geoscience Letters, 3, 34.

Yano, J.-I., and J. J. Tribbia, 2017: Tropical atmospheric Madden–Julian oscillation: A strongly nonlinear free solitary Rossby wave? J. Atmos. Sci., 74, 3473–3489,

Zhang, C. and Ling, J., 2012. Potential vorticity of the Madden–Julian oscillation. Journal of the atmospheric sciences, 69(1), pp.65-78.

---

## Author Response (AR1)

We thank the reviewers for their careful reading of the paper and for their helpful comments. We have considered these comments carefully in our revision of the paper and respond below to each in turn. The reviewers comments are shown in black text and our responses in blue.

**Responses to Reviewer 1:**

The manuscript "A simple dynamical model linking radiative-convective instability, convective aggregation and large-scale dynamics" proposed a theoretical model that relates the small-scale convection aggregation and the large-scale dynamics. The authors analyzed in a hierarchical way. The aspects of this study are interesting, and the results have potential implication in the tropical dynamics. However, the writing needs to be improved. The manuscript is too lengthy, and it is unclear what are the major scientific issues the authors want to address. Based on my evaluation, I suggest major revision for this manuscript.

General comments:
        Manuscript is hard to follow, not because of its mathematical content but because of its writing. First, the introduction part needs to be improved. It is unclear what are the specific issues the authors want to address. Do the authors want to study the convection aggregation associated with MJO? Or how small-scale convection aggregation and the large-scale dynamics are connected? Second, the manuscript is too lengthy, and some of the contents appear to be distracting. I suggest the authors greatly shorten the manuscript for conciseness.

RESPONSE: We accept that it is important that the scientific issues to be addressed by the paper are made clear and that in the version of the paper we submitted we attempted to say too much, making it difficult for the reader to identify the intended scope of the paper. In revising we have therefore shortened the manuscript and, in particular restricted the scope of the discussion, along the lines suggested by the Reviewers.

To be clear, our goal here is to set out a simplified theoretical model that addresses the link between the process of convective aggregation, say on the scale of 100km or less, with larger scale dynamics on the scale of 1000km or larger. We see the model as potentially relevant to the MJO, as a manifestation of that larger scale dynamics, but we accept that not all readers will regard the behaviour of the model as presented in this paper as 'MJO-like'. Therefore in revising we have reduced significantly the amount of MJO-related discussion in the paper.

In the revised version of the paper we have amended the abstract to make the goal of the paper clearer. We have reduced the length of Section 1. We have shortened Section 3, in particular to omit material on the 1-D model. We have shortened Section 4, in particular moving most of the discussion of the linear stability problem to an Appendix, and again omitting material on the 1-D model. We have retained most of the material in Section 5, which discusses the equatorial beta-plane behaviour, and in Section 6, but in the latter have again tried to emphasise that key theme of the paper is the link between between convective aggregation and large-scale dynamics with any strong relevance to the MJO requiring further model development.

REVIEWER SPECIFIC COMMENT 1: As I mentioned above, do the authors want to study the relation between convection aggregation and MJO? If so, I'm afraid that this goal was not achieved. Because the results shown in section 5 seems to be not the MJO, which has planetary scales.

RESPONSE: We have focused and shortened the paper as described above, accepting that not all may find the beta-plane behaviour self-evidently MJO-like and therefore that it is best to postpone detailed comment on the MJO to a future paper.

REVIEWER SPECIFIC COMMENT 2: I suggest the authors describe what are the differences between the model used in this study and those in the previous studies in section 2, and focus on the new findings based on these model differences in the following section. This may condense the manuscript.

RESPONSE: We have not followed this recommendation exactly, but we have condensed the manuscript and we have amended Sections 1 and 2 to make it clearer how our model relates to previous studies.

REVIEWER SPECIFIC COMMENT 3: In Page 12, L345, what are the spatial patterns of q for the initial conditions in all figures?

RESPONSE: The spatial pattern for q is taken to be independent random values at each grid point. This is now explicitly stated in the text.

REVIEWER SPECIFIC COMMENT 4: In page 12, L346, I don't understand why "The precise form of the functions Fh and Fq appearing respectively in (2) and (3) is not important to the qualitative behaviour".

RESPONSE: The final part of the quoted sentence "provided that they together lead to a bistable moisture distribution (7)" is important. The discussion in the previous section has argued that if WTG applies and therefore that the reaction-diffusion equation (7) is a good descriptor of the evolution of the moisture distribution then, if there is bistability, locally $q$ will tend one of the two stable values $q_+$ or $q_-$ and that the essence of the subsequent time evolution is the evolution of the boundary between the region with $q = q_+$ and the region with $q = q_-$. Certainly the precise form of the function G_hq appearing on the right-hand side is important to the precise details of the evolution in the reaction-diffusion-like WTG limit and when dynamics is included the fact that the dynamics is forced by the heating term $F_h(q)$ means that the details of the function $F_h(q)$ will determine the details of the time evolution. But the statement is being made about the quantitative time evolution behaviour, not about the detailed behaviour. The text has been amended to emphasise this point. Another way of saying this is that, if there is bistability in the sense explained, then changing the precise forms of the functions $F_h(q)$ and $F_q(q)$ has not been found to lead to any substantially different behaviour, whereas the variation with the parameters alpha, lambda, f and beta, as described in Sections 4 and 5, is more substantial.

REVIEWER SPECIFIC COMMENT 5: In Eq. (13), what are the values of qp and qm?

RESPONSE: Values of parameters used in the numerical simulations are given in the caption Table 1. There was an error in our original submitted version of the paper in that the values given were stated as those of $q_+$ and $q_-$ when the values given were actually those of $q_p$ and $q_m$. The Table 1 caption now (correctly) gives the values of qp and qm.

REVIEWER SPECIFIC COMMENT 6: Can the authors explain the reason for introducing qp and qm in the parameterization of Fh. What are the physical meanings of qp and qm? It seems to me that introducing qp and qm are artificial. And the consequent model performance is a build-in result.

RESPONSE: See previous Response. It is indeed the case that $F_q(q)$ and $F_h(q)$ have been specified artificially, but the rationale here is that we have evidence from previous work, e.g 1-D radiative-convective calculations under the WTG assumption, that radiative-convective instability and thence stable moist and dry states, i.e. bistability, can be attained. The focus of our paper is the implications of this for large-scale dynamics, which of course cannot be addressed within the 1-D calculations. To address this we have simply constructed $F_h(q)$ and $F_q(q)$ that in combination give the required bistability property. Certainly, because the bistability has been imposed, the fact that the system in each location evolves into either the dry state or the moist state is 'built in'. But the focus of the paper, as we have tried to make as clear as possible in the revised version of the paper, is on subsequent aggregation behaviour, how this is affected by the dynamics and on, the beta-plane, how the combination of bistability and dynamics lead to propagation. These consequences are surely not 'built in' -- if they were regarded as 'built in' this description would apply to any consequence of new physics introduced into any model system.

REVIEWER SPECIFIC COMMENT 7: Can the authors explain why nonlinearity could lead to band-like patterns of convection aggregation in Fig. 5? This will probably not happen in real world.

RESPONSE: As noted in Section 3 in the original version of the paper and in the shortened Section 3 in the revised version of the paper, the inclusion of nonlinear advection tends to reduce the size of moist regions, because they coincide with flow convergence. The addition of this effect means that the equilibrium shape of moist and dry regions is now determined by a balance between reaction-diffusion, the divergence-convergence patterns and the geometry constraint of the domain (e.g. the divergence has to integrate to zero over the domain). It is likely that the change in shape as epsilon is increased could be explained via some kind of variational principle but this seems beyond the scope of the paper. We have simply presented the result here to illustrate the range of behaviour that is possible, emphasising that the highly organised band-like patterns seem unlikely to be observed in the real atmosphere -- but leaving open the possibility that some aspects of this behaviour -- e.g. that moist regions are thinner and more 1-dimensional in shape -- might be observed.

REVIEWER SPECIFIC COMMENT 8: Did the authors calculate the zonal propagation speed in Fig. 14, 16 and 18 based on Eq. (41-43)? Could the disturbances in these figures have both eastward and westward components? If so, using Eq. (41-43) may not be the best choice for measuring the zonal propagation speed.

RESPONSE: The values of zonal propagation speed given in Figs. 14, 16 and 18 are estimated from the time evolution of the q field in the simulation, not from the expressions (41-43) -- (38-40) in the revised version. The captions to these Figures have been modified to emphasise this point. We describe at the beginning of Section 5.2 how the zonal propagation speed has been estimated, noting that in some cases there is no single phase speed because different parts of the pattern propagate at different phase speeds. We agree that formulae (41-43) are unlikely in such cases to be helpful -- they would be capturing some kind of 'average' phase speed.

**Responses to Reviewer 2**

REVIEWER OPENING COMMENT: This is perhaps the most thorough study on convective aggregation and its potential effect on large-scale dynamics. It goes through meticulous discussions, both analytically and numerically, of the role of rotation (f and beta-effects), nonlinear advection, thermal and dynamical damping and mean flow in convective aggregation and in the consequential large-scale zonal propagation. Its systematic treatment of convective aggregation in a variety of parameter space is a huge step forward from previous studies on this subject. This is a massive study with many mathematical details. It's impractical to discuss all these details in this review, although the devil is in them. Most of my comments are on the main objective of this study, which is to link convective aggregation to large-scale dynamics. My general sense is that the discussion on aggregation is fine, but its implication to the MJO is a stretch.

RESPONSE: We interpret the above comments as broadly positive, but expressing reservations about the length of the paper and about whether the relevance of the link from convective aggregation to large-scale dynamics, which is indeed the intended focus of the paper, can be extended to the MJO.

In revising the paper we have attempted to address both these reservations. In the revised version of the paper we have reduced the length of Section 1, omitting most of the MJO discussion. We have shortened Section 3, in particular to omit material on the 1-D model. We have shortened Section 4 on the f-plane, in particular moving most of the discussion of the linear stability problem to an Appendix, and again omitting material on the 1-D model, with the intention of focusing on those f-plane results that provide a framework for interpreting the equatorial beta-plane behaviour. We have retained most of the material in Section 5, on the equatorial beta-plane behaviour, and in Section 6, but in the latter have again tried to focus the content on the key theme of the paper, the link between between convective aggregation and large-scale dynamics, rather than any specific focus on the MJO.

REVIEWER GENERAL COMMENT 1: There are so many simplifications and assumptions in the formulation of the theory that it is difficult for readers to see which ones are for mathematical convenience to get an analytical solution but bear minimal physical consequences, and which ones are key to the conclusion. It might be helpful to have a table that list all simplifications and assumptions and marks those that is at the center of this theory (e.g., must have to make the conclusion valid).

RESPONSE: We have considered the idea of a table setting out assumptions but feel that it is better to focus on setting out the assumptions as clearly as possible in the text. To this end we have revised some of the ordering of Section 2 to make clearer the logic of the model formulation, including how it builds on previous work. In brief, the reduction to single-level equations, shallow-water-like in the dynamics and accompanied by a moisture evolution equation follows Neelin and Zeng (2000) and successor papers. Then the key assumption regarding the moisture-dependent heating and precipitation terms that appear in these equations is that there is a bistability property. This follows work by Sobel et al (2007), Emanuel et al (2014) and others who have demonstrated such bistability in 1-D radiative-convective calculations under the WTG assumption. This was explained in the submitted version of the paper but we have attempted to make it even clearer in the revised version.

REVIEWER GENERAL COMMENT 2: Sections 3 and 4 are unnecessarily long. The failure of aggregation on an f-plane has been documented before (e.g., Carstens and Wing 2022). Even

though more detailed discussions on this subject add intellectual values and it might be nice to compare results on a beta-plane to those on an f-plane, Sections 3 and 4 do not directly contribute to the main conclusion of this study, which can be made solely based on the beta-plane discussions. Suggest substantially shorten this part and retain only the materials that are directly relevant to compare with those on a beta-plane. This would make the manuscript more focused on tropical dynamics and easier to follow. The removed materials on an f-plane can be published elsewhere.

RESPONSE: We see the inclusion of Section 3 -- establishing that our model does indeed in the WTG regime exhibit aggregation of the same type as the CMWC model -- and Section 4 -- going beyond WTG to investigate the role of rotation, friction and thermal damping -- as part of the systematic progression towards the beta-plane case that the Reviewer has noted in their opening comments. Nonetheless we accept that too much detail can be counterproductive and we have shorted both of these Sections -- omitting discussion of the 1-d case and removing significant detail on the linear instability problem. We hope that this shortening addresses the Reviewer's concerns whilst providing important supporting evidence and interpretation that underpins the discussion of the beta-plane case.

REVIEWER GENERAL COMMENT 3: I found the interpretation of large-scale zonal propagation in terms of its dependence on latitudes very interesting. Based on the authors' interpretation, when Rossby (Kelvin) wave response is stronger, the propagation tends to be eastward (westward). This is counter- intuitive but not impossible. There are several issues that the authors may want to consider:

1. a)  The Rossby wave may lead to eastward propagation without any convective aggregation (Hayashi and Itoh 2017; Rostami and Zeitlin 2019; Yano and Tribbia 2017).

RESPONSE: In two of the papers above (Rostami and Zeitlin 2019; Yano and Tribbia 2017) the eastward propagation is as vortex-pair effect, with cyclones on each side of the equator, and this can be described as a strongly nonlinear Rossby wave. In the Hayashi and Itoh paper there is also a vortex pair, but the mechanism for propagation is not vortex-pair-self-advection, but the fact that the vorticity field organises convective heating, which in turn leads to a vorticity tendency that shifts the vorticity structure to the east. This again could be described as a Rossby wave -- but perhaps a 'diabatic Rossby wave' to emphasise the fact that diabatic processes play an essential role. These mechanisms for eastward propagation are quite different  from that in our model, where it is the relation between the moisture field and the associated heating, and the corresponding pattern of convergence/divergence implying a change in the moisture field, that leads to the propagation. For the Rossby wave response to the heating, the convergence is shifted to the east of the heating and the divergence is to the west, therefore the effect is propagation to the east. We have cited the above three papers in our revised version and noted that the propagation mechanisms are very different.

2. b)  The westward propagation due to stronger Kelvin wave responses in the context of convective aggregation should be discuss in the context of its natural (dynamical) eastward propagation. In other words, what is the net balance between the dynamical eastward propagation of the Kevin wave and its westward propagation due to convective aggregation?

RESPONSE: As in the explanation above for the Rossby wave case, it is the relation of the convergence/divergence pattern to the heating, and the implications for the moisture, that determines the propagation. (The dynamics is in quasi-steady state with the heating. The propagation of dynamical information is of course important for determining the quasi-steady

dynamical field, but it is the effect on the moisture that determines the propagation.) For the Kelvin wave response the convergence is shifted to the west of the heating and the divergence to the east, therefore the propagation is to the west.

3. c) The "<" shape structure is in the opposite direction of the "swallowtail" pattern observed for the MJO (Zhang and Ling 2012). How do the two reconcile with each other if the authors think their results are relevant to the MJO?

RESPONSE: We acknowledged in the first version of the paper that this aspect of the propagating disturbances was not consistent with the observed MJO and that is why we were cautious in describing the disturbances as MJO-like. This caution has been retained in the revised version of the paper, in particular in the concluding statements in Section 6.

4. d) With a strength of convective heating associated with moisture anomalies, its Rossby and Kelvin wave responses are pretty much fixed based on the Gill model. Under what circumstances the relative strengths of Rossby and Kelvin wave responses would vary? By the latitude location of convective heating?

RESPONSE: Our analysis and results suggest that it is the shape of the moisture anomaly that determines the relative strength of the Kelvin wave and Rossby wave responses. Broadly speaking a moisture and hence heating anomaly that is wider in latitude will tend to produce a stronger Rossby wave response. We discuss this point in Section 5.2 of the paper. But as we note, because there is always both a Kelvin wave response and a Rossby wave response, with opposing effects on propagation, whether or not there is net westward or eastward propagation can depend on quite subtle aspects of the relative geometry of the moisture field and the heating-induced patterns of convergence/divergence.

REVIEWER GENERAL COMMENT 4: The time it takes for the aggregation to reach a large-scale state (400 days) is much longer than the initiation of the MJO or any tropical large-scale disturbances. How should this inconsistency be reconciled?

RESPONSE: Certainly any proposed mechanism for development of tropical large-scale disturbances must provide a route for such development on time scales much less than 400 days. We are proposing aggregation as a potentially relevant mechanism but not necessarily as a mechanism operating in isolation. Our simulations are initialised by grid-scale small-amplitude noise. First the radiative-convective instability has to develop to finite-amplitude, giving small-scale moist and dry regions, then those regions have to increase in size through aggregation. In the real tropical atmosphere there will be finite-amplitude external perturbations on the system, such as equatorial waves forced by remote mesoscale convective events, perhaps with some moderate scale spatial organisation, meaning that the starting scale for the aggregation process may be, say, a few hundred kilometres. The key point that we wish to make is that there is an aggregation process that will tend to increase scales, and therefore to explain a phenomenon such as the MJO one does not have to find a 'scale-selection' process that initialises finite-amplitude disturbances at the scale of the MJO itself. Therefore aggregation should be considered as relevant alongside proposed mechanisms for scale selection, such as Adames and Kim (2016) proposal of a scale-dependent radiative destabilisation. (There are probably several proposed mechanisms.)

We have amended the revised text to make it clear that time scale of O(100 days) needed for aggregation to the scales of O(1000 km) suggests that aggregation, as described in our paper, cannot be the the sole mechanism acting.

REVIEWER GENERAL COMMENTS 5: In the conclusion section (6), the authors compared their results with those from previous studies of the MJO, all of them relying on the moisture-mode doctrine. It might be further enlightening if the authors walk out of the moisture-mode and aggregation camps and discuss what their results add to other simpler theories. For example, as mentioned above, several studies (Hayashi and Itoh 2017; Rostami and Zeitlin 2019; Yano and Tribbia 2017) proposed that the dry Rossby wave alone may propagate eastward to provide a propeller effect on the MJO. Kim and Zhang (2021) suggested that dry Kelvin waves alone can propagate eastward slowly at the MJO speed in the presence of dynamical damping. They reproduced the observed "swallowtail" pattern without Rossby waves. As the authors know very well, Adames and Kim (2016) provided a perhaps the most comprehensive MJO theory based on the moisture-mode thinking. All these studies reproduced fundamental MJO properties to various degrees without convective aggregation. The big question for this study under review is: What additional physical insights to MJO dynamics can be added to the previous studies or what fundamental MJO properties missing from the previous studies can be produced by including convective aggregation? Especially, the authors mentioned that more physics are needed to make the results more realistic. Is that so? Majda and Stechmann (2009) and Kim and Zhang (2021) included far less physics (assumptions) but produced no less fundamental properties of the MJO than other MJO theories. Sometimes less is better. I challenge the authors of this study to pursue the beauty of simplicity. If convective aggregation plays an important role in the MJO, there must be a way to demonstrate this in a simple, straightforward way without invoking numerous assumptions and massive mathematical procedures.

RESPONSE: This is a wide-ranging comment. A first part of the response is to say that, since both Reviewers have encouraged us to de-emphasize the MJO in this paper, we are reluctant to include a larger scale discussion of MJO theories. A second is to say that whilst aggregation might seem complicated, what is being described is a behaviour that emerges robustly from what are simple equations -- to a large extent similar equations to those that have been used in many of the papers cited above. The primary difference is that in our study the important moisture-dependent terms, the heating and the precipitation, are nonlinear functions of moisture that in combination lead to a bistable behaviour. But, as we have noted, that choice is clearly motivated by detailed calculations in 1-D radiative convective models, for example, and it does not rely on delicate properties of the appropriate functions -- there are many choices of these functions that will give similar behaviour. There is a certainly an assumption here -- but it is in our view a well-motivated assumption. If in future work we need to introduce further physical assumptions then we will of course have to justify those. But that is surely for a future review process, not for this one. What we do accept an important is not to give the impression that moisture-mode theories are the only viable theories for the MJO and we have amended our discussion in an effort to do that.

REVIEWER DETAILED COMMENT 1: Near line 20: "The key overall properties of the propagating disturbances, the spatial scale and the phase speed, depend on nonlinearity in the coupling between moisture and dynamics and any linear theory for such disturbances therefore has limited usefulness." I beg to differ. See major comment 4 regarding simple linear theories.

RESPONSE: Rephrased as 'Within this model, the key overall properties of the propagating disturbances, the spatial scale and the phase speed, depend on nonlinearity in the coupling between moisture and dynamics and any linear theory for such disturbances therefore has limited usefulness.' to make it clear that comment applies to this particular model (where it is justifiable), rather than being a comment that applies to linear models in general.

REVIEWER DETAILED COMMENT 2. Line 28: "Much theoretical and modelling work over the past few decades has focused on the coupling between dynamics and moisture in the tropical atmosphere, which it is clear must be taken into account at leading-order to explain many tropical phenomena." Ditto. Look around outside the moisture-mode camp and you would find theories with equal or greater success in comparison to the moisture-mode ones. This would help to put this theory in a broader context.

RESPONSE: Rephrased as: 'Much theoretical and modelling work over the past few decades has focused on the coupling between dynamics and moisture in the tropical atmosphere, which it has been argued should be taken into account at leading-order to explain many tropical phenomena.' This is a statement of fact. It does not claim that ALL work on this topic has focused on the coupling between dynamics and moisture, nor does it claim that alternative approaches are invalid.

REVIEWER DETAILED COMMENT 3: The origin of the moisture equation (3) needs to be scrutinized. In the text, it is "assumed", which is not good enough. In most of the moisture-mode studies, such a moisture equation is thrown to the well-established shallow-water equation as a second thought. This has been one of the most severe inconsistencies in the moisture-mode theories that very few have realized. Neelin and Zeng (2000) derived a set of horizontal structure equations for wind, temperature and humidity (their Eqs 5.1 – 5.4) after applied variable separation and vertical base functions. Can the moisture equation (3) in this study under review be vigorously derived
from Neelin and Zeng's humidity equation (5.4)? If not, we have a problem.

RESPONSE: In the submitted version of the paper we noted that our governing equations could be obtained from the full primitive equations by the procedure set out out by Neelin and Zeng (2000) -- we have re-ordered the text in the revised version to make that clearer.

REVIEWER DETAILED COMMENT 4: What is the physical reality of d hbar/dt (Eq 5)? Global mean temperature fluctuation? Is there an example for this? Please forgive my ignorance if this is a common knowledge.

RESPONSE: The key point here is that the divergence term appearing on the left-hand side of (2) integrates to zero over the domain, therefore if the change in $q$ implies a non-zero domain average of $F_h(q)$ this cannot be balanced by the divergence term, it must result in a change in the domain average h, i.e. domain average temperature. This does not seem to be physically unrealistic -- global mean temperature would indeed change if there was an increase in global mean heating due to water vapour -- e.g. more latent heating. The same would apply to the mean taken over more restricted domain, e.g. the tropics, if there was no transport into or out of that domain.

REVIEWER DETAILED COMMENTS 5: Please provide justifications for the values of alpha, lambda and kappa chosen in Table 1 or point out where in the text such justifications are given if I missed them.

RESPONSE: We have chosen values that are representative of those by other recent papers on similar topics. Details are given when Table 1 is introduced, except that values for alpha, lambda and f are discussed later, in Section 4, where these parameters are taken to be non-zero. Note that we take the view that there is no 'correct' for alpha and lambda in particular and we present results for different values of these parameters in order to highlight the different behaviour that may occur as a result.

REVIEWER DETAILED COMMENT 6: The domain integrations (Eqs 42 – 43) need explanations. If it covers the entire longitudinal
circle, it mixes the convective part (Indian Ocean and Western Pacific) where zonal speeds tend to be lower than the dry part of the tropics. Otherwise, do they distinguish the faster and slow speeds when the domain covers the moist and dry zones separately?

RESPONSE: We have amended the text (beginning of Section 5) to make this clearer. What is being considered here is a hypothetical tropics in which there is no imposed longitudinal variation. Furthermore, the focus in this paper is on the case where the equilibrium gross moist stability is negative. This is intended as a representation of the 'warm pool' Indian Ocean and West Pacific region. It is similar in spirit to large-scale CRM simulations where conditions are such that convective aggregation occurs over the whole domain. The extension to a longitudinally varying state, where for example, equilibrium gross moist stability is negative in part of the domain (the 'warm pool' region) and positive elsewhere, will be investigated in future.

REVIEWER DETAILED COMMENT 7: The linear friction coefficient alpha plays a central role on a beta-plane but not on an f-plane. The parameter c beta /?alpha^3/2 lambda^1/2 suggests that alpha must not be zero on a beta-plane. All alpha = 0 cases are on an f-plane. Linear friction plays critical role in some theories (e.g., Adames and Kim 2016; Wang et al. 2016; Kim and Zhang 2021), but not in others (e.g., Majda and Stechmann 2009; Hayashi and Itoh 2017; Yano and Tribbia 2017; Rostami and Zeitlin 2019; Emanuel 2019). Can the author explain why linear friction must exist in this theory and what exactly it does to make the theory work on a f-plane? Eqs 36 and 37 clearly demonstrate that alpha and f (or beta y) play equivalent roles and they cannot be zero at the same time. This deserves a physical explanation: Why must friction exist at the equator but not away from the equator? Go back to the starting point (Eqs 14 – 17), there must be a solution with alpha = 0. What is that solution on a beta-plane?

RESPONSE: Actually the only alpha=0 cases we show are in Section 3, with f=0 and lambda=0, when aggregation proceeds to the domain scale. So we have not intended to demonstrate that friction is required at the equator and not required away from the equator. Furthermore, whilst both alpha and f certainly appear in the formulae (36) and (37), and in other formulae, they do not appear in the same way. So they cannot be considered to play the same roles. The role of friction in both cases (f-plane and beta-plane) is to allow a quasi-steady dynamical response to the heating provided by a slowly-varying moisture distribution. If the frictional damping is absent then the dynamics will not be quasi-steady. We have discussed briefly what might happen in this case in Section 4 and we have also shown that when the frictional damping becomes small the aggregated state becomes more time dependent and its structure changes. A more detailed examination of the small-friction case could be undertaken but, bearing in mind that both Reviewers find the paper as currently written is long, we feel that would be best included in a separate paper. To respond to this reviewer comment we have noted, where we justify the values assumed for alpha on the basis of those chosen in previous calculations, that there is continuing uncertainty on the inclusion of simple friction into models of equatorial dynamics.

REVIEWER DETAILED COMMENT 8: Figs. 14, 15, 16, and 19: Does it take the equal length of integration to reach the states shown in the panels? What is the integration length? 400 days as in Fig. 18? It should be mentioned in the captions.

RESPONSE: The states shown in these Figures are indeed all at 400 days. This is mentioned in the text at the beginning of section 5.2.1 as applying to all cases shown, but we have now added to the caption to Figure 14 (Figure 11 in the revised paper) to emphasise the point.

REVIEWER DETAILED COMMENT 9: In all the parameters used, which one controls the speed at which aggregated perturbations grow? Fig. 9 suggests it's kappa when alpha = lambda = f = 0. How about on a beta-plane when alpha and lambda are not zero?

RESPONSE: We see the appropriate description as that when the scale of moist/dry regions is small compared with L_dyn then the rate of aggregation is determined by kappa. (In other words the non-WTG dynamics can be ignored at this stage.) Then when the scale becomes comparable with L_dyn the non-WTG dynamics becomes important and the aggregation ceases. This is consistent with the behaviour shown in Figure 9 (Figure 8 in new numbering). We think that it is more helpful to think about the dynamics halting the aggregation at a given scale than slowing it. On the beta-plane the relevant L_dyn is simply obtained by setting f=0 and becomes $c/(\alpha \lambda)^{1/2}$. This is the longitudinal scale at which aggregation halts. We have looked carefully at the text of Section 5 and have made small changes to make this point clearer. As we note the scale $c/(\alpha \lambda)^{1/2}$ is the longitudinal damping-controlled length scale for equatorial waves noted by Wu et al (2001) for the case where thermal and frictional damping rates are different, generalising the 'textbook' case considered by Gill.

---

## Referee Report (RR1)

The authors present a comprehensive study of convective aggregation in the relatively simple dynamical framework of the shallow water equations. At the core of this simple model of convective aggregation is an assumed bistability of the RCE state which tends to be driven towards two different stable equilibrium states, one moist and one dry. Having been brought on as a reviewer in this second round, I feel that the authors have adequately addressed the concerns over framing (and in particular the relevance of the study to the MJO) raised in the first round, and the main body of the manuscript now presents a coherent picture of convective aggregation within this idealized model. As such, I feel the manuscript is nearly ready for publication, and have only minor suggestions that I feel will bring further clarity to the presented results.

**Specific Comments**

**Equation 11**: The quantity $A_+$ is defined in line 267 as a fractional area, but then the LHS of this equations has dimensions $1/T$, and the RHS dimensions $L^2/T$. Presumably multiplication of the LHS by the square of the domain scale will rectify this inconsistency.

**Figure 3**: The information presented in panel (a) and panels (b)-(d) seems redundant. I find panel (a) to be a poor visualization and hard to glean information from, and so would recommend its removal. The caption also does not indicate the significance of the red curves in panel (a). The main points, i.e. the bimodal character of the moisture distribution and its achievement of a steady state, are adequately shown just with panels (b)-(d).

**Figure 4**: The fact that the aggregated regions in the nonlinear simulations organize along the directions of the underlying discretized grid seems like a numerical artifact (i.e. why wouldn't the aggregated line have some arbitrary orientation in the horizontal plane?). This should be briefly commented on in Section 3.2.

**Line 425**: With the chosen parameters $Q/H = 0.5$, $\mu_2/\mu_1 = 3$, the normalized gross moist stability (GMS) is $M = -0.5$. While I have no issue with the requirement of negative GMS, this gives quite a large magnitude relative to what is assumed in linear models for convectively-coupled waves. The authors have noted above that their model is insensitive to specific parameter choices, but I still feel this strongly negative value should be noted explicitly when introducing the GMS as an important parameter for the model.

**Line 520**: This wording in this sentence is poor, I would recommend re-wording to make it more clear.

**Line 588**: Should this read $\alpha = \lambda = f = 0$, rather than simply $\alpha = \lambda = f$?

**Figure 10**: The figure and corresponding caption are hard to follow. I would recommend colour-coding the lines or labelling them on the figure itself.

**Line 899**: Do Sugiyama's physically derived forms for $F_q$ and $F_h$ exhibit the kind of bistable behavior that is central to the results of this study? This seems like an important point in connecting the authors' ansatz to reality.

---

## Author Response (AR2)

We thank the reviewers and co-editor for further careful reading of the paper and for their helpful comments. These comments have been considered carefully during the revision of the paper, and we will respond to each in turn below. The reviewers comments are shown in black and our responses in blue.

**Response to Report #1**

The long lead time (>100 days) needed for a large-scale convective system to grow out of aggregation cannot by any means be related to the MJO. The initiation timescale of the MJO is about 30 days. By that time, a large-scale circulation pattern would have already formed and interacted with convection. Any convective growth from then on cannot be explained by aggregation.

We accept that this comment is fair however we feel that it has been sufficiently covered in the current manuscript. In the previous revision of the paper we included additional notes at the end of the introduction (lines 107-112) that the model does not reach the stage of direct relevance to the MJO. We discussed this further in the conclusions (lines 957-975), emphasising the difference between our model and the MJO in both time-scale for the aggregation to form and in the spatial structure and propagation characteristics of the aggregated moist and dry regions on the beta plane. In particular, in the paragraph at lines 957-966, we specifically acknowledge that aggregation alone is unlikely to account for the observed behaviour of the MJO where a well-organised disturbance appears on a time scale of a few days at a spatial scale of 10^4 km.

**Response to Report #2**

The authors present a comprehensive study of convective aggregation in the relatively simple dynamical framework of the shallow water equations. At the core of this simple model of convective aggregation is an assumed bistability of the RCE state which tends to be driven towards two different stable equilibrium states, one moist and one dry. Having been brought on as a reviewer in this second round, I feel that the authors have adequately addressed the concerns over framing (and in particular the relevance of the study to the MJO) raised in the first round, and the main body of the manuscript now presents a coherent picture of convective aggregation within this idealized model. As such, I feel the manuscript is nearly ready for publication, and have only minor suggestions that I feel will bring further clarity to the presented results.

Specific Comments

Equation 11: The quantity A+ is defined in line 267 as a fractional area, but then the LHS of this equations has dimensions 1/T, and the RHS dimensions L^2/T. Presumably multiplication of the LHS by the square of the domain scale will rectify this inconsistency.

The text and one equation have been altered so that A_- and A_+ are now actual areas, not fractional areas.

Figure 3: The information presented in panel (a) and panels (b)-(d) seems redundant. I find panel (a) to be a poor visualization and hard to glean information from, and so would recommend its removal. The caption also does not indicate the significance of the red curves in panel (a). The main points, i.e. the bimodal character of the moisture distribution and its achievement of a steady state, are adequately shown just with panels (b)-(d).

We feel that panel (a) does provide useful extra information, in particular on the evolution of the q_+ and q_- values with time and the agreement of the observed moisture distribution with the predicted values of q_+ and q_-, but accept that the caption did not make this clear. We have added detail to the caption to make it clearer what can be seen in panel (a).

Figure 4: The fact that the aggregated regions in the nonlinear simulations organize along the directions of the underlying discretized grid seems like a numerical artifact (i.e. why wouldn't the aggregated line have some arbitrary orientation in the horizontal plane?). This should be briefly commented on in Section 3.2.

The referee raises an interesting point regarding the possible role of the numerical grid in determining the steady-state solutions shown in Figures 4(b) and 4(c). It is certainly the case the geometry of the domain, as distinct from the geometry of the numerical grid, is likely to be playing a very important role. We have changed the text to emphasise this point further. The role of the domain shape in nonlinear reaction-diffusion type problems has been established in the relevant literature through analytical work which does not depend on any particular grid. Therefore we think that it is more likely that it is the shape of the domain rather the shape of the numerical grid that is relevant here and we have not mentioned the latter in the revised text. (But we accept that there cannot be absolute certainty on this point.)

Line 425: With the chosen parameters Q/H = 0.5, μ2/μ1 = 3, the normalized gross moist stability (GMS) is M = −0.5. While I have no issue with the requirement of negative GMS, this gives quite a large magnitude relative to what is assumed in linear models for convectively-coupled waves. The authors have noted above that their model is insensitive to specific parameter choices, but I still feel this strongly negative value should be noted explicitly when introducing the GMS as an important parameter for the model.

We have now noted the value of M in the text immediately after the specification of parameter values in Section 3.1, line 362, so that readers are made aware of it.

Line 520: This wording in this sentence is poor, I would recommend re-wording to make it more clear.

We have re-ordered the sentence to make the meaning more clear.

Line 588: Should this read α = λ = f = 0, rather than simply α = λ = f?

This error has been corrected in the text.

Figure 10: The figure and corresponding caption are hard to follow. I would recommend colourcoding the lines or labelling them on the figure itself.

The figure has been updated by colour coding the lines and giving their equations in the legend, and the caption has been simplified correspondingly.

Line 899: Do Sugiyama's physically derived forms for Fq and Fh exhibit the kind of bistable behavior that is central to the results of this study? This seems like an important point in connecting the authors' ansatz to reality. 1

Sugiyama (2009a) indeed notes bistable behaviour in  a certain limit of the model considered and we have added a comment this effect. However we have also reminded the reader that the bistability is motivated more generally by other work cited earlier in the paper and have repeated those citations.